# Damaged brain accelerates bone healing by releasing small extracellular vesicles that target osteoprogenitors

Wei Xia [1,7], Jing Xie [1,7], Zhiqing Cai [1,7], Xuhua Liu [2], Jing Wen [3], Zhong-Kai Cui [1], Run Zhao [1], Xiaomei Zhou [1], Jiahui Chen [2], Xinru Mao [4], Zhengtao Gu[5], Zhimin Zou [5], Zhipeng Zou [1], Yue Zhang [1], Ming Zhao[5], Maegele Mac[6], Qiancheng Song [1✉] & Xiaochun Bai [1✉]

Clinical evidence has established that concomitant traumatic brain injury (TBI) accelerates bone healing, but the underlying mechanism is unclear. This study shows that after TBI, injured neurons, mainly those in the hippocampus, release osteogenic microRNA (miRNA)-enriched small extracellular vesicles (sEVs), which targeted osteoprogenitors in bone to stimulate bone formation. We show that miR-328a-3p and miR-150-5p, enriched in the sEVs after TBI, promote osteogenesis by directly targeting the 3′UTR of *FOXO4* or *CBL*, respectively, and hydrogel carrying miR-328a-3p-containing sEVs efficiently repaires bone defects in rats. Importantly, increased fibronectin expression on sEVs surface contributes to targeting of osteoprogenitors in bone by TBI sEVs, thereby implying that modification of the sEVs surface fibronectin could be used in bone-targeted drug delivery. Together, our work unveils a role of central regulation in bone formation and a clear link between injured neurons and osteogenitors, both in animals and clinical settings.

[1] Guangdong Provincial Key Laboratory of Bone and Joint Degeneration Diseases, Department of Cell Biology, School of Basic Medical Sciences, Southern Medical University, Guangzhou 510515, China. [2] State Key Laboratory of Organ Failure Research, Academy of Orthopedics, Guangdong Province, The Third Affiliated Hospital of Southern Medical University, Guangzhou, Guangdong 510630, China. [3] Department of Radiology, Nanfang Hospital, Southern Medical University, Guangzhou, Guangdong 510515, China. [4] Department of Clinical laboratory, Nanfang Hospital, Southern Medical University, Guangzhou, Guangdong 510515, China. [5] Department of Pathophysiology, Guangdong Provincial Key Laboratory of Shock and Microcirculation Research, Southern Medical University, Guangzhou, Guangdong 510515, China. [6] Institute for Research in Operative Medicine, Private University of Witten-Herdecke, Cologne Merheim Medical Center, Ostmerheimerstr 200, D-51109 Cologne, Germany. [7] These authors contributed equally: Wei Xia, Jing Xie, Zhi-Qing Cai. ✉email: songqc@smu.edu.cn; baixc15@smu.edu.cn

The brain governs most physiological functions of the body, from complexities of cognition, learning, and memory, to regulation of basal body temperature, heart rate, and breathing. The brain is well established as a master regulator of homeostasis in peripheral tissues; central regulation of bone remodeling represents a novel and expanding field of study[1,2]. A variety of neurogenic factors, such as leptin, serotonin, semaphorins, neuropeptide Y, proopiomelanocortin, and amphetamine-regulated transcript have been shown to act on the hypothalamus and profoundly alter bone reconstruction[1–5]. More recently, it was shown that bone might return the favor to the brain through the secretion of the osteoblast-derived molecule, osteocalcin (OCN)[6,7]. The skeleton influences development and cognitive functions of the central nervous system suggesting intimate crosstalk between bone and brain.

Traumatic brain injury (TBI) is one of the most severe trauma-induced injuries and is one of the leading causes of death and disability[8]. In patients with long bone fractures, concomitant TBI is commonly observed in trauma settings. Interestingly, clinical evidence over the last five decades has shown that concomitant TBI accelerates bone healing[9,10]. Multiple studies have shown an exacerbated callus formation and increased mineral density in patients and rats with concomitant TBI. Reports suggest that some osteogenic cytokines and hormones, such as nerve growth factor, vascular endothelial growth factor, bone morphogenetic proteins, thrombin, and leptin are elevated in the serum or the brain during TBI[9,11–14]; however, compelling evidence for a correlation between damaged brain and exacerbated bone formation is lacking. Some studies suggest that immune factors are involved; however, the observation that injuries to other organs, which also cause changes in immune factors, do not promote fracture healing, is a conundrum. Thus, elucidating mechanisms responsible for TBI-stimulated osteogenesis will not only provide cogent evidence for neuronal control of bone but also prompt potential therapy for the treatment of delayed bone healing or bone nonunion.

In this study, we showed that plasma small extracellular vesicles (sEVs) from patients and rats with TBI had much greater potential to promote osteogenesis than soluble factors. Injured neurons produce sEVs that carry osteogenic microRNAs (miRNAs), which target osteoprogenitors. Hydrogel carrying sEVs containing these miRNAs efficiently accelerated bone healing and repaired critical-sized rat calvarial defects in rats. Our findings have established a clear link between the damaged brain and bone formation both in animals and clinical settings.

## Results

**TBI accelerates bone healing in patients and rats.** To confirm the impact of TBI on fracture healing, we compared the recovery rate of 12 patients with femoral fracture and concomitant TBI and 12 patients with femoral fracture only (Supplementary Table 1). X-ray images at 4 and 6 weeks after surgery showed that fracture recovery rate was much faster in patients with fracture and TBI than in patients with fracture only (Fig. 1a and Supplementary Fig. 1). Volumes of fracture callus, which is an important index of bone fracture healing, were much larger in patients with fracture and concomitant TBI than in fracture-only patients at 4 and 6 weeks after fracture (Fig. 1a). These data confirmed that TBI promoted the healing of bone fractures in patients.

To duplicate the effect of the damaged brain on bone formation in an animal model, we used lateral fluid percussion to establish a TBI model in rats (Fig. 1b and Supplementary Fig. 2). Thirty Sprague–Dawley (SD) rats were subsequently randomized into the proximal tibia defect (PTD) and PTD combined-TBI (PTD + TBI) groups. Micro-computed tomography (micro-CT) analysis

showed a marked increase in callus volume in PTD + TBI rats compared with that of PTD rats on days 7 and 14 after surgery (Fig. 1c). Bone volume/tissue volumes (BV/TV%), hematoxylin and eosin (H&E) staining, and the number of osteoblasts per bone perimeter also showed that osteogenesis was elevated dramatically in PTD + TBI rats (Fig. 1d, e and Supplementary Fig. 3a). Accordingly, in vivo bone regeneration rate increased significantly after TBI (Supplementary Fig. 3b–d). Furthermore, OCN staining confirmed an increase in osteogenesis in PTD + TBI rats (Fig. 1f, g), whereas no significant change in the number of osteoclasts tartrate-resistant acid phosphatase (TRAP) positive cells and osteoclast surface per bone surface between the two groups (Supplementary Fig. 3e–g).

On the contrary, the liver injury could not promote bone healing in the PTD or PTD combined liver injury rat model (Supplementary Fig. 4). In summary, these results indicated that TBI but not liver injury stimulated bone formation and accelerated bone healing in patients and rats.

**Plasma sEVs from TBI patients and rats contribute to TBI-stimulated osteogenesis.** To explore the mechanism by which TBI promotes bone formation, we first compared the osteogenic effects of plasma from patients or rats with TBI and controls. We found that plasma from TBI patients or rats promoted osteoblastic differentiation (Fig. 2a, b and Supplementary Fig. 5a) without affecting its proliferation (Supplementary Fig. 5b).

Next, TBI plasma was separated into insoluble and soluble components to screen for factors responsible for TBI-stimulated osteogenesis. Unexpectedly, plasma sEVs from patients and rats with TBI had much greater potential to promote osteogenesis in vitro than soluble factors (Fig. 2c, d). The purified sEVs were verified by transmission electron microscopy (TEM), NanoSight analysis, and immunoblotting for protein markers (CD9, Alix, and TSG101), according to the proposal of the International Society of Extracellular Vesicles (Supplementary Fig. 6). Notably, the concentration of plasma sEVs in TBI rats was much higher than that in shams (Fig. 2e).

To confirm the role of sEVs in TBI-stimulated osteogenesis and bone healing in vivo, GW4869, an inhibitor of sEVs secretion, was intravenously injected into PTD rats with concomitant TBI. The significant reduction of sEVs isolated from the plasma of rats with GW4869 treatment was confirmed by NanoSight Analysis and western blotting (Supplementary Fig. 7a, b). Micro-CT analysis showed a significantly decreased callus volume, bone volume/tissue volumes, BV/TV%, and trabecular number (Tb.N), after GW4869 treatment on days 7 and 14 post-injury ($P < 0.05$) (Fig. 2f–h). Trabecular bone and OCN expression were also reduced dramatically after inhibition of sEV release by GW4869 (Fig. 2i–k). These results suggested that sEV release was required for TBI-stimulated osteogenesis and bone healing in rats.

Taken together, these results suggested that plasma sEVs from patients and rats with TBI had much greater potential to promote osteogenesis than soluble factors. SEVs contributed to TBI-stimulated bone formation in vitro and bone healing in rats.

**Plasma sEVs released by TBI patients and rats carry potent osteogenic miRNAs, such as miR-328a-3p and miR-150-5p.** Next, we performed miRNA sequencing and proteomics analysis to explore how TBI-derived sEVs promoted osteogenesis and bone healing. The miRNAs, which can be abundantly encapsulated in sEVs, are key regulators of bone remodeling in health and disease[15,16]. We, therefore, evaluated the miRNA expression profiles in sham-sEV and TBI-sEV groups using miRNA sequencing. To identify the specific miRNAs involved, we compared miRNA

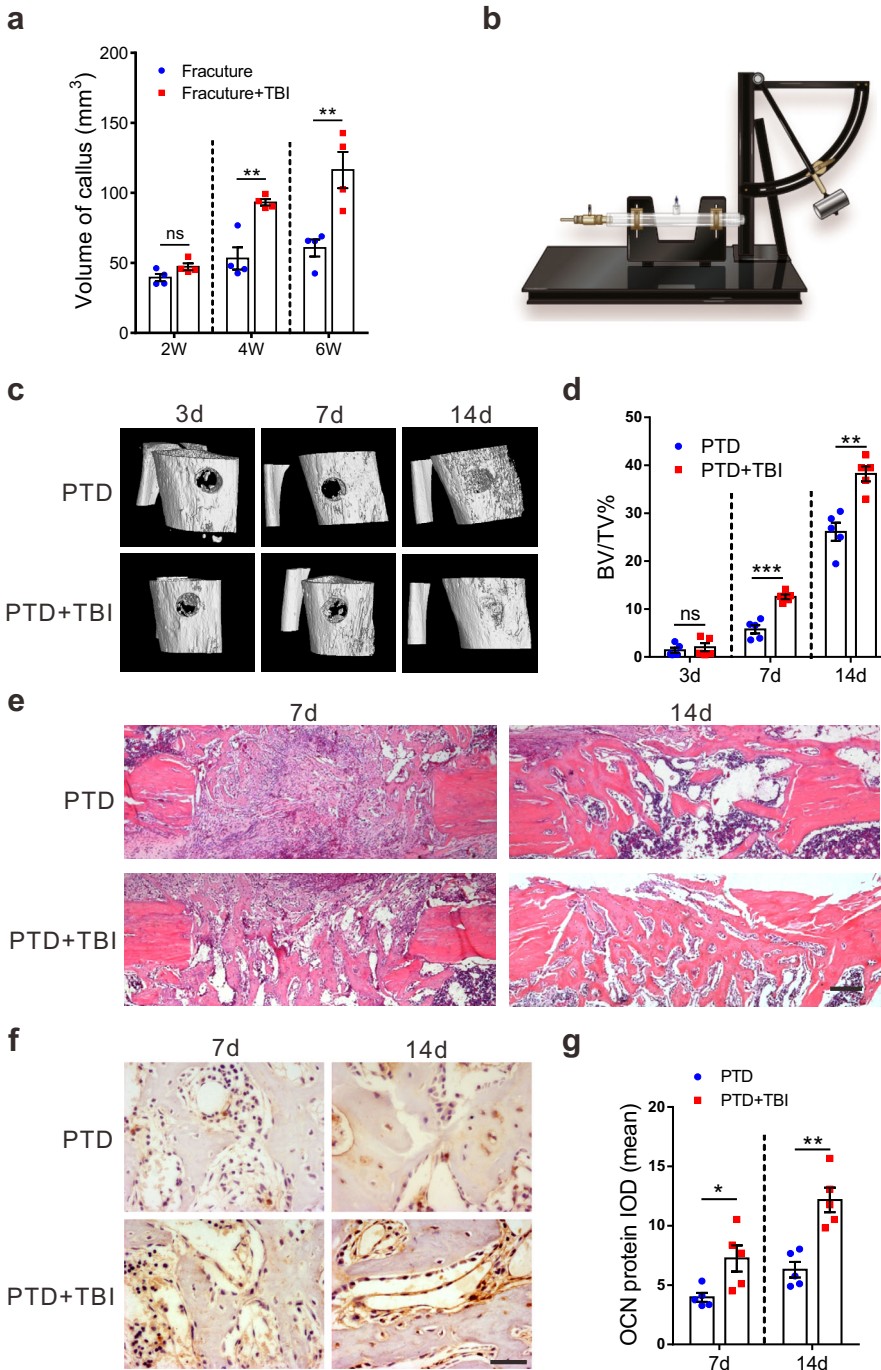

**Fig. 1 TBI (traumatic brain injury) accelerates bone healing in patients and rats. a** Fracture callus volume in fracture alone patients and TBI patients with concomitant fracture ($n = 4$ person per group per time point, Student's two-sided unpaired $t$ test). $P_{(2w\ TBI+fracture\ vs.\ 2w\ Fracture)} = 0.0787$, $P_{(4w\ TBI+fracture\ vs.\ 4w\ Fracture)} = 0.0028$, $P_{(6w\ TBI+fracture\ vs.\ 6w\ Fracture)} = 0.0082$. **b** Scheme illustrating the fluid-percussion device for experimentally induced TBI. Representative micro-computed tomography (CT) 3D image (**c**) and BV/TV% analysis (**d**) of defect sites of PTD (proximal tibia defect) and PTD + TBI (PTD combined-TBI) groups on day 3, 7, and 14 after operation ($n = 5$ animals per group per time point; Student's two-sided unpaired $t$ test). $P_{(3d\ PTD+TBI\ vs.\ PTD)} = 0.5424$, $P_{(7d\ PTD+TBI\ vs.\ PTD)} = 0.0001$, $P_{(14d\ PTD+TBI\ vs.\ PTD)} = 0.0011$. **e** Representative H&E staining of rat PTD sites show newly formed woven bone in the defect sites. Scale bars, 200 μm ($n = 5$ animals per group per time point). **f**, **g** Representative immunohistochemical staining and quantitation of OCN-positive cells in rat PTD sites at 7 and 14 days after surgery ($n = 5$; Student's two-sided unpaired $t$ test). $P_{(7d\ PTD+TBI\ vs.\ PTD)} = 0.0225$, $P_{(14d\ PTD+TBI\ vs.\ PTD)} = 0.0014$. Scale bars, 40 μm. The quantitation result were plotted as dot plots, showing the mean ± SEM of three independent experiments. $^{*}P < 0.05$, $^{**}P < 0.01$; ns = nonsignificant.

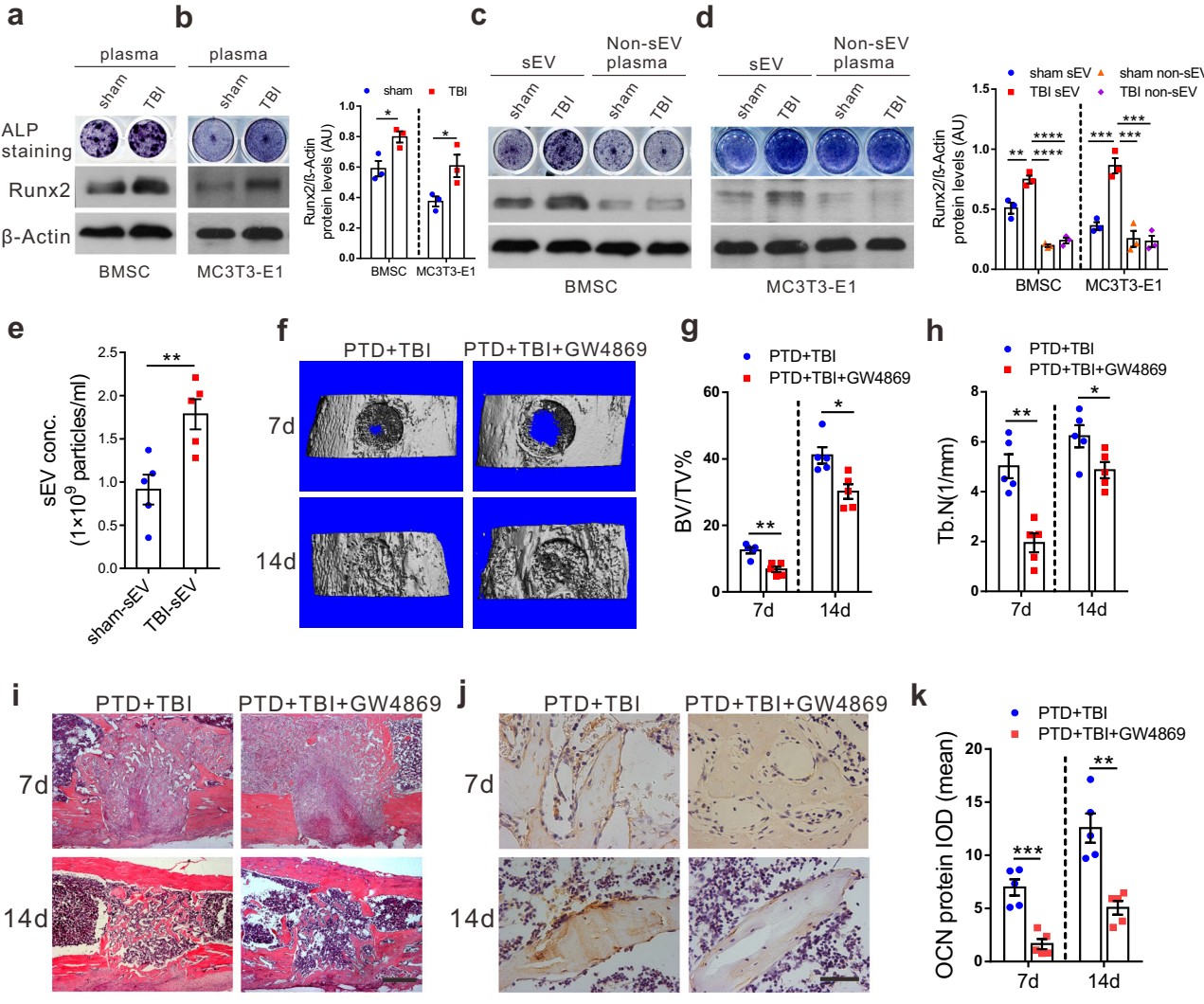

**Fig. 2 Plasma sEVs from TBI rats contribute to TBI-stimulated osteogenesis. a, b** In vitro, freshly isolated rat BMSCs and MC3T3-E1 cells ($2 \times 10^5$/well) were treated with plasma of rats (20 µL). Total proteins were extracted and subjected to Western blot analysis showing the expression of RUNX2, β-actin, and ALP staining of BMSCs and MC3T3-E1 cells treated with plasma isolated from sham and TBI groups ($n = 3$; Student's two-sided unpaired $t$ test). $P_{(BMSC\ TBI\ vs.\ sham)} = 0.029$, $P_{(MC3T3-E1\ TBI\ vs.\ sham)} = 0.0444$. **c, d** sEVs and non-sEV component were isolated from the plasma of sham or TBI rats. BMSCs and MC3T3-E1 cells ($2 \times 10^5$/well) were treated with the sEVs or the non-sEV component isolated from equal amount of plasma. Western blot analysis showing the expression of RUNX2, β-actin, and ALP staining of BMSCs and MC3T3-E1 cells treated with sham sEV, TBI sEV, sham non-sEV, or TBI-non-sEV ($n = 3$; one-way analysis of variance with Turkey's multiple comparisons test was performed). $P_{(BMSC\ sham\ sEV\ vs.\ TBI\ sEV)} = 0.0028$, $P_{(BMSC\ TBI\ sEV\ vs.\ sham\ non-sEV)} < 0.0001$, $P_{(BMSC\ TBI\ sEV\ vs.\ TBI\ non-sEV)} < 0.0001$, $P_{(MC3T3-E1\ sham\ sEV\ vs.\ TBI\ sEV)} = 0.0008$, $P_{(MC3T3-E1\ TBI\ sEV\ vs.\ sham\ non-sEV)} = 0.0002$, $P_{MC3T3-E1\ TBI\ sEV\ vs.\ TBI\ non-sEV} = 0.0002$. **e** Concentration of sEVs isolated from the plasma of rats ($n = 5$, Student's two-sided unpaired $t$ test). $P_{(TBI-sEV\ vs.\ sham-sEV)} = 0.0076$. **f** Micro-CT 3D structure image of proximal tibial defect healing at 7 and 14 days postsurgery with intraperitoneal injection of GW4869 or DMSO saline. BV/TV (**g**) and Tb.N (**h**) of region with proximal tibial defect were obtained from micro-CT analysis ($n = 5$; Student's two-sided unpaired $t$ test). $P_{(7d\ BV/TV\ PTD+TBI+GW4869\ vs.\ PTD+TBI)} = 0.0015$, $P_{(14d\ BV/TV\ PTD+TBI+GW4869\ vs.\ PTD+TBI)} = 0.011$, $P_{(7d\ Tb.N\ PTD+TBI+GW4869\ vs.\ PTD+TBI)} = 0.001$, $P_{(14d\ Tb.N\ PTD+TBI+GW4869\ vs.\ PTD+TBI)} = 0.0392$. **i** Representative H&E staining images of proximal tibial defect region at 7 days and 14 days post-impact. Scale bars, 200 µm, ($n = 5$ animals per group per time point). **j, k** Representative immunohistochemical staining and quantitation of expression of OCN in rat PTD sites ($n = 5$; Student's two-sided unpaired $t$ test). $P_{(7d\ PTD+TBI+GW4869\ vs.\ PTD+TBI)} = 0.0004$, $P_{(14d\ PTD+TBI+GW4869\ vs.\ PTD+TBI)} = 0.0012$. Scale bars, 40 µm. The quantitation result were plotted as dot plots, showing the mean ± SEM of three independent experiments. $^*P < 0.05$, $^{**}P < 0.01$, $^{***}P < 0.001$, $^{****}P < 0.0001$.

profiles of plasma sEVs derived from the TBI and the sham rats; the results are shown by heatmaps (Fig. 3a and Supplementary Table 2). Four of the maximally upregulated miRNAs and two of the downregulated miRNAs were validated by real-time quantitative polymerase chain reaction (qRT-PCR) (Fig. 3b).

To determine the effects of these miRNAs on osteoblast formation, MC3T3-E1 cells and primary rat BMSCs were transfected with miRNA mimics or negative controls. We found that miR-328a-3p and miR-150-5p significantly promoted

osteoblast differentiation (Fig. 3c, d). Inhibitors of miR-328a-3p or miR-150-5p, on the contrary, prevented osteoblastic differentiation of osteoprogenitors (Fig. 3e, f). Interestingly, miR-328a-3p and miR-150-5p were also markedly increased in plasma sEVs from patients with TBI, as compared to healthy controls (Fig. 3g).

These data suggested that sEVs released by patients and rats with TBI carried potent osteogenic miRNAs, such as miR-328a-3p and miR-150-5p.

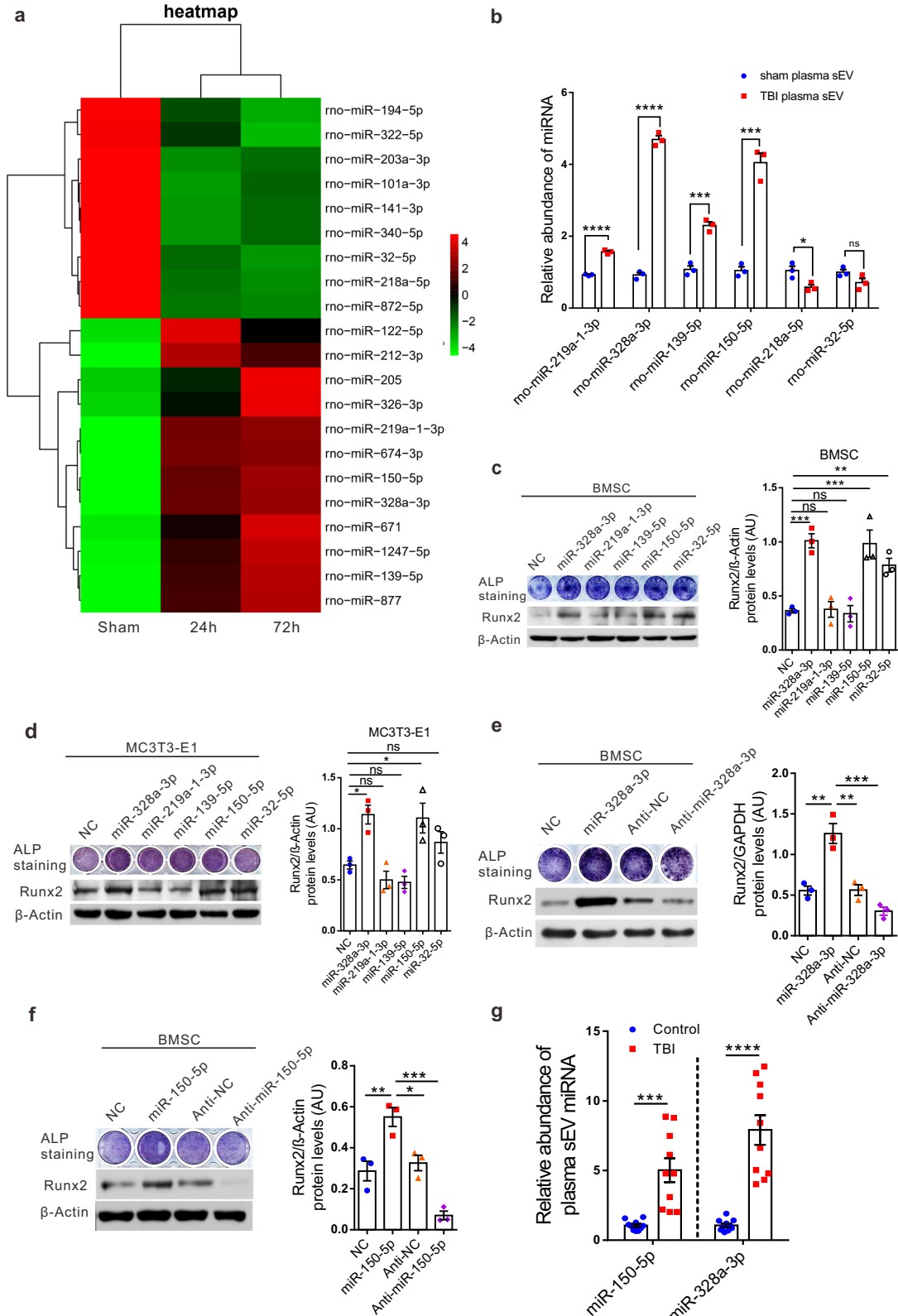

**sEV miR-328a-3p and miR-150-5p promote osteogenesis by directly targeting *FOXO4* and *CBL* 3′-UTR, respectively**. To identify the targets of miR-328a-3p and miR-150-5p, first, we applied PicTar and TargetScan to search for putative miRNA targets. Based on the representation of miRNA sites in their 3′ UTRs, >100 mRNAs were predicted to be regulated by miRNAs. Among these candidates, eight genes were involved in the suppression of osteogenesis.

To determine whether miRNAs target these genes directly, we cloned the 3′UTRs of the putative targets into a dual luciferase assay system. Reporter assays using MC3T3-E1 cells revealed that miR-328a-3p repressed the *FOXO4* 3′UTRs and miR-150-5p repressed the *CBL* 3′UTRs, significantly (Fig. 4a–c). Mutations of the putative miRNA sites in these two 3′UTRs abrogated responsiveness to miRNA (Fig. 4c–e).

**Fig. 3 Plasma sEVs released by TBI patients and rats carry potent osteogenic miRNAs, such as MiR-328a-3p and MiR-150-5p. a** Heat map presentation of relative miRNAs with ≥1.5-fold difference between rat plasma sham-sEV and TBI-sEV at 24 and 72 h post-impact. **b** Comparison of relative miR-219a-1-3p, miR-328a-3p, miR-139-5p, miR-150-5p, miR-218a-5p, miR-32-5p content between sham-sEV and TBI-sEV of rat plasma at 24 h post-impact by qRT-PCR ($n = 3$; Student's two-sided unpaired $t$ test). $P_{(miR-219a-1-3p\ TBI\ plasma\ sEV\ vs.\ sham\ plasma\ sEV)} < 0.0001$, $P_{(miR-328a-3p\ TBI\ plasma\ sEV\ vs.\ sham\ plasma\ sEV)} < 0.0001$, $P_{(miR-139-5p\ TBI\ plasma\ sEV\ vs.\ sham\ plasma\ sEV)} = 0.0009$, $P_{(miR-150-5p\ TBI\ plasma\ sEV\ vs.\ sham\ plasma\ sEV)} = 0.0004$, $P_{(miR-218a-5p\ TBI\ plasma\ sEV\ vs.\ sham\ plasma\ sEV)} = 0.0278$, $P_{(miR-32-5p\ TBI\ plasma\ sEV\ vs.\ sham\ plasma\ sEV)} = 0.1044$. **c, d** BMSCs and MC3T3-E1 cells were transfected with miR-328a-3p, miR-219a-1-3p, miR-139-5p, miR-150-5p, miR-32-5p mimics, or NC (negative control) in osteogenic medium. ALP staining and western blots were performed on day 7. β-actin served as the loading control ($n = 3$; one-way analysis of variance with Dunnett's multiple comparisons test was performed). $P_{(BMSC\ NC\ vs.\ miR-328a-3p)} = 0.0007$, $P_{(BMSC\ NC\ vs.\ miR-219a1-3p)} = 0.9998$, $P_{(BMSC\ NC\ vs.\ miR-139-5p)} = 0.9987$, $P_{(BMSC\ NC\ vs.\ miR-150-5p)} = 0.0004$, $P_{(BMSC\ NC\ vs.\ miR-32-5p)} = 0.009$, $P_{(MC3T3-E1\ NC\ vs.\ miR-328a-3p)} = 0.0278$, $P_{(MC3T3-E1\ NC\ vs.\ miR-219a-1-3p)} = 0.8747$, $P_{(MC3T3-E1\ NC\ vs.\ miR-139-5p)} = 0.7938$, $P_{(MC3T3-E1\ NC\ vs.\ miR-150-5p)} = 0.0430$, $P_{(MC3T3-E1\ NC\ vs.\ miR-32-5p)} = 0.5791$. **e, f** Human BMSC were transfected with miRNAs mimics, NC and their anti-miRNAs on day 7 for ALP staining. Western blot analysis of RUNX2 expression in human BMSCs. β-actin and GAPDH serve as loading control ($n = 3$; One-way analysis of variance with Turkey's multiple comparisons test was performed). $P_{(NC\ vs.\ miR-328a-3p)} = 0.001$, $P_{(miR-328a-3p\ vs.\ Anti-NC)} = 0.0011$, $P_{(miR-328a-3p\ vs.\ Anti-miR-328a-3p)} = 0.0001$. $P_{(NC\ vs.\ miR-150-5p)} = 0.0067$, $P_{(miR-150-5p\ vs.\ Anti-NC)} = 0.0166$, $P_{(miR-150-5p\ vs.\ Anti-miR-150-5p)} = 0.0001$. **g** RT-PCR identification of sEV miR-328a-3p and miR-150-5p in TBI patients versus healthy controls ($n = 10$; Student's two-sided unpaired $t$ test). $P_{(miR-150-5p\ TBI\ vs.\ Control)} = 0.0002$, $P_{(miR-328a-3p\ TBI\ vs.\ Control)} < 0.0001$. The quantitation result were plotted as dot plots, showing the mean ± SEM of three independent experiments. $^{*}P < 0.05$, $^{**}P < 0.01$, $^{***}P < 0.001$, $^{****}P < 0.0001$; ns not significant.

We next determined whether there was an association between expression levels of target genes in bone defect sites of TBI-TPD rats. We found that levels of FOXO4 and CBL were decreased at the bone defect site of TBI-PTD rats as compared with that of PTD rats, which suggests that FOXO4 and CBL may be involved in TBI-stimulated bone formation (Fig. 4f, g). Accordingly, transfection of miR-328a-3p or miR-150-5p mimics reduced endogenous FOXO4 or CBL protein levels in MC3T3-E1 cells. On the contrary, inhibition of miR-328a-3p or miR-150-5p significantly increased FOXO4 or CBL protein levels (Fig. 4h, i). We further found that knockdown of FOXO4 or CBL produced similar changes in osteogenesis to that of miR-328a-3p or miR-150-5p overexpression, respectively (Fig. 4j, k). Importantly, the osteogenic effect of miR-328a-3p or miR-150-5p could be partially rescued by overexpression of FOXO4 or CBL, indicating that FOXO4 and CBL were downstream targets of miR-328a-3p and miR-150-5p respectively, which promoted bone formation.

Taken together, these results suggested that sEV miR-328a-3p and miR-150-5p downregulated FOXO4 and CBL expression, respectively, by directly targeting their 3′UTR to promote bone formation.

**Injured neurons produce sEVs that target bone and osteoprogenitors**. We next explored where these osteogenic sEVs originated from during TBI. We speculated that these sEVs might originate from injured brain tissue. In situ hybridization showed that expression of miR-328a-3p in brain tissues of rats increased significantly after TBI, especially in the CA1 and DG regions of the hippocampus (Fig. 5a and Supplementary Fig. 8). Importantly, immunofluorescence staining showed that expression of CD63 in rat brains was intensively induced by TBI (Fig. 5b). Interestingly, most of the CD63 was expressed in microtubule-associated protein 2 (MAP2)-positive, especially in the CA1 region of the hippocampus, but not glial fibrillary acidic protein (GFAP)-positive cells, indicating that damaged neuronal cells, but not astrocytes secreted sEVs vigorously (Fig. 5b, c and Supplementary Figs. 9 and 10).

Primary traumatic injury generally initiates a cascade of secondary injury mechanisms such as edema, hemorrhage, and decreased cerebral blood flow resulting in ischemic injury to brain tissue[17]. While the ischemic injury is a secondary consequence of TBI, ischemia is the primary mechanism of injury in cases that disrupt arterial cerebral blood flow leading to a lack of oxygen and glucose in brain tissue and oxygen–glucose deprivation (OGD) is an in vitro model commonly used for TBI[18,19]. The results of the western blot of RUNX2 and ALP staining showed that the osteoblastic differentiation of BMSC and MC3T3-E1 cells increased after cocultured with OGD treated primary hippocampal neurons (Supplementary Fig. 11d, e). Moreover, we found that the sEVs release and sEV miR-328a-3p and miR-150-5p levels were significantly elevated in cultured primary hippocampal neurons after OGD treatment (Fig. 5d and Supplementary Fig. 11b, c). And the miR-328a-3p content did not change significantly while the content of miR-150-5p was increased in sEVs of murine microglial cell line BV2 after OGD treatment (Supplementary Fig. 11a).

To determine whether these damaged neurons produced sEVs and their miRNA cargo could target bone and osteoprogenitors, rats were intravenously injected with PKH67-labeled sEVs (800 μg per rat) isolated from plasma of TBI or sham rats, and the distribution of PKH67-sEV was evaluated by biophotonic imaging after intracerebroventricular (ICV) injection or intravenous injection. We found that PKH67-labeled sEVs purified from rats with TBI, but not from rats with liver injury or sham rats, accumulated in bone, but not in visceral organs or brain (Fig. 5e and Supplementary Figs. 12 and 13c). Moreover, we compared the intake of PKH67-labeled sEVs by cultured BMSC, and MC3T3-E1 and MLO-Y4 cell lines (osteocyte cell lines) and found that BMSC and MC3T3-E1 cell lines absorbed more plasma sEVs from TBI rats than from sham rats, whereas no difference was observed in MLO-Y4 cells (Fig. 5f). Notably, in situ hybridization analysis of miR-328a-3p expression in rat PTD sites demonstrated that the abundance of miR-328a-3p was higher in PTD sites in TBI rats than the sham rats at 7 days after surgery (Fig. 5g).

Taken together, these data implied that sEVs and their miRNA cargo derived from injured neurons could target bone and osteoprogenitors.

**Protein fibronectin 1 (FN1) directs sEVs to target osteoprogenitors and bone**. It is interesting to note that TBI-induced sEVs accumulated in bone. We speculated that proteins expressed on sEVs surfaces or membranes may contribute to increased affinity of these sEVs to the bone; hence, we performed proteomic analysis and compared protein profiles of plasma sEVs from TBI and sham rats (Fig. 6a and Supplementary Data 1). It was of interest that the quantity of FN1, an extracellular matrix and surface glycoprotein, increased 1.32 times in amount in TBI sEVs as compared with control sEVs. Notably, A2M, a marker for neuronal injury[20,21], also increased 3.099 times in amount, implying that these sEVs might originate from injured neurons. Further experiments results confirmed the increase of FN1 and

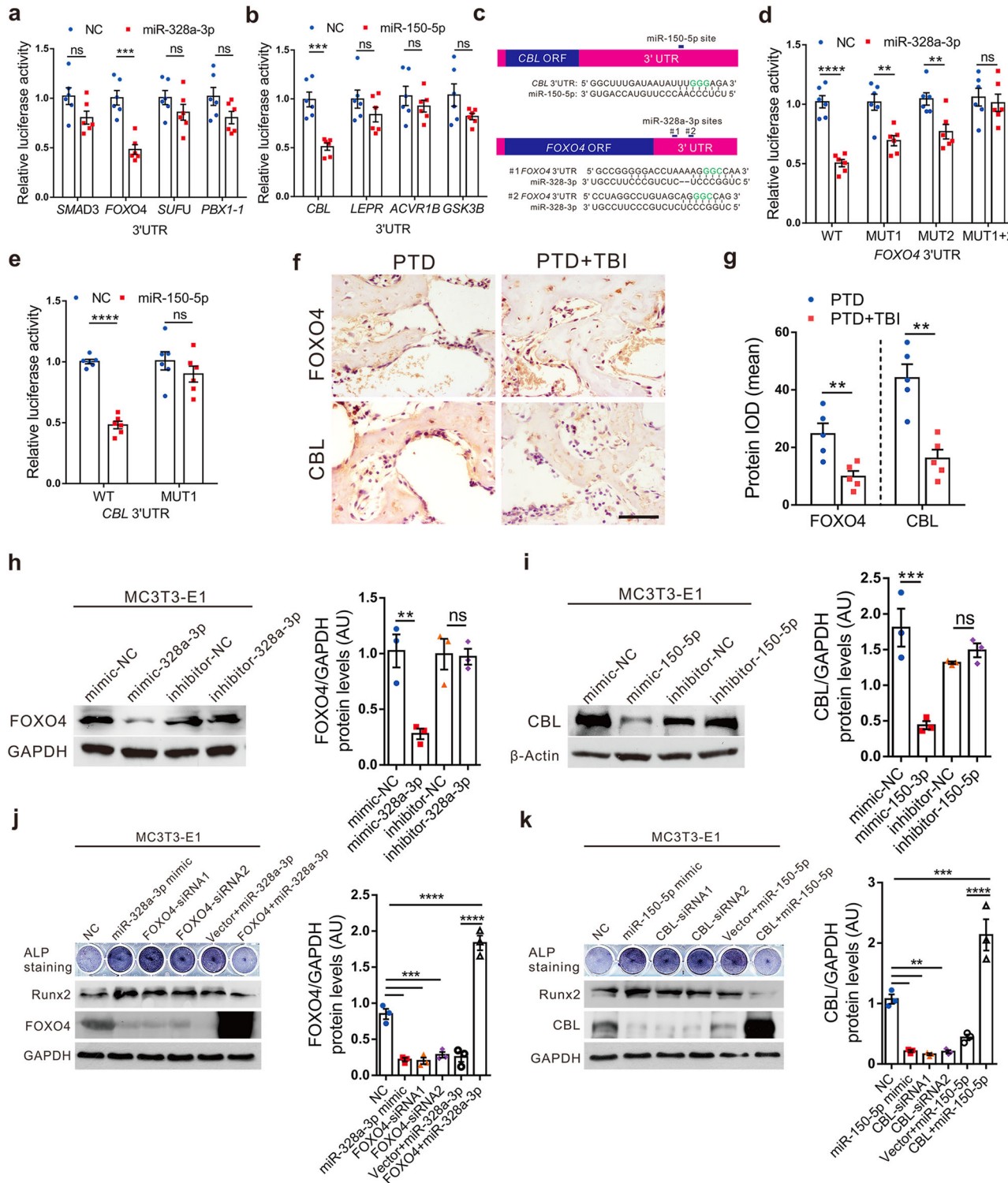

A2M protein in sEVs from TBI rats (Fig. 6b, c and Supplementary Fig. 14a, b). In addition, western blot results showed that MAP2 and UCHL1, two makers of neuron cells, were increased in sEVs isolated from TBI rat plasma (Supplementary Fig. 14a). Immunofluorescence staining showed that expressions of both FN1 and A2M in the brain were intensively induced by TBI (Fig. 6d, e). Furthermore, co-staining of A2M and OCN could be clearly observed at the PTD site of PTD-TBI rats but not PTD

alone rats (Fig. 6f), thereby implying that uptake of sEVs might concurrently increase osteogenesis in PTD-TBI rats.

We further confirmed the FN1-directed targeting of sEVs to osteoprogenitors in vitro. As expected, FN1 expression was dramatically enhanced in OGD treated primary hippocampal neurons, HT22 cell lines and in its sEVs, as compared with controls (Fig. 6g–i and Supplementary Fig. 14c–g). In addition, the uptake sEVs isolated from primary hippocampal neurons and

**Fig. 4 sEV miR-328a-3p and miR-150-5p promote osteogenesis by directly targeting *FOXO4* and *CBL* 3′-UTRs, respectively. a** The psiCHECK-2-*SMAD3/FOXO4/SUFU/PBX1-1* 3′-UTRs of wild-type genes were co-transfected with the NC or miR-328a-3p mimic into MC3T3-E1 cells. Cells were harvested for quantitation of dual luciferase activities at 24 h post transfection (*n* = 6, Student's two-sided unpaired *t* test). $P_{(SMAD3\ miR-328a-3p\ vs.\ NC)}$ = 0.0684, $P_{(FOXO4\ miR-328a-3p\ vs.\ NC)}$ = 0.0001, $P_{(SUFU\ miR-328a-3p\ vs.\ NC)}$ = 0.1952, $P_{(PBX1-1\ miR-328a-3p\ vs.\ NC)}$ = 0.0795. **b** The psiCHECK-2-*CBL/LEPR/ACVR1B/GSK3B* 3′-UTR of wild-type genes were co-transfected with the NC or miR-150-5p mimic into MC3T3-E1 cells. Cells were harvested for quantitation of dual luciferase activities at 24 h post-transfection (*n* = 6; Student's two-sided unpaired *t* test). $P_{(CBL\ miR-150-5p\ vs.\ NC)}$ = 0.0002, $P_{(LEPR\ miR-150-5p\ vs.\ NC)}$ = 0.2135, $P_{(ACVR1B\ miR-150-5p\ vs.\ NC)}$ = 0.3862, $P_{(GSK3B\ miR-150-5p\ vs.\ NC)}$ = 0.0946. **c** Schematic diagram of putative miR-328a-3p and miR-150-5p binding sites in the *Foxo4* and *CBL* 3′-UTR. Green letters denote mutation sites. Luciferase activity in the indicated MC3T3-E1 cells upon transfection of miR-328a-3p (**d**) and miR-150-5p (**e**) binding site mutant (Mut) of 3′-UTR-driven reporter constructs (*n* = 6; Student's two-sided unpaired *t*-test). $P_{(WT\ miR-328a-3p\ vs.\ NC)}$ < 0.0001, $P_{(MUT1\ miR-328a-3p\ vs.\ NC)}$ = 0.0023, $P_{(MUT2\ miR-328a-3p\ vs.\ NC)}$ = 0.0063, $P_{(MUT1+2\ miR-328a-3p\ vs.\ NC)}$ = 0.6581, $P_{(WT\ miR-150-5p\ vs.\ NC)}$ < 0.0001, $P_{(MUT1\ miR-150-5p\ vs.\ NC)}$ = 0.303. Representative immunohistochemical staining (**f**) and quantitation of the expression (**g**) of FOXO4 and CBL in rat PTD sites on day 7 after surgery (*n* = 5; Student's two-sided unpaired *t* test). $P_{(FOXO4\ PTD+TBI\ vs.\ PTD)}$ = 0.0073, $P_{(CBL\ PTD+TBI\ vs.\ PTD)}$ = 0.0011. Scale bars, 40 μm. Western blot analysis of the change in FOXO4 (**h**) and CBL (**i**) protein in MC3T3-E1 cells after transfection with NC, miR-328a-3p mimics, miR-150-5p mimics (50 nm), inhibitor-NC, inhibitor-miR-328a-3p, inhibitor-miR-150-5p (100 nm) for 72 h (*n* = 3; one-way analysis of variance with Turkey's multiple comparisons test was performed). $P_{(mimic-NC\ vs.\ mimic-328a-3p)}$ = 0.006, $P_{(inhibitor-NC\ vs.\ inhibitor-328a-3p)}$ = 0.9988, $P_{(mimic-NC\ vs.\ mimic-150-3p)}$ = 0.0007, $P_{(inhibitor-NC\ vs.\ inhibitor-150-5p)}$ = 0.8309. **j**, **k** The MC3T3-E1 cells were transfected with NC, miRNA mimic, siRNA against *FOXO4/CBL*, co-transfected with miRNA mimics and *FOXO4/CBL* or vector plasmid. ALP staining and western blotting were performed to analysis of RUNX2, FOXO4, and CBL expression in MC3T3-E1 cells after transfection. GAPDH served as loading control (*n* = 3; one-way analysis of variance with Turkey's multiple comparisons test was performed). $P_{(NC\ vs.\ miR-328a-3p\ mimic)}$ = 0.0002, $P_{(NC\ vs.\ FOXO4-siRNA1)}$ = 0.0001, $P_{(NC\ vs.\ FOXO4-siRNA2)}$ = 0.0005, $P_{(Vector+miR-328a-3p\ vs.\ FOXO4+miR-328a-3p)}$ < 0.0001, $P_{(NC\ vs.\ FOXO4+miR-328a-3p)}$ < 0.0001, $P_{(NC\ vs.\ miR-150-5p\ mimic)}$ = 0.0017, $P_{(NC\ vs.\ CBL-siRNA1)}$ = 0.001, $P_{(NC\ vs.\ CBL-siRNA2)}$ = 0.0016, $P_{(Vector+miR-150-5p\ vs.\ CBL+miR-150-5p)}$ < 0.0001, $P_{(NC\ vs.\ CBL+miR-150-5p)}$ = 0.0003. The quantitation result were plotted as dot plots, showing the mean ± SEM of three independent experiments. *$P$ < 0.05, **$P$ < 0.01, ***$P$ < 0.001, ****$P$ < 0.0001; ns not significant.

HT22 cells with OGD treatment by BMSC and MC3T3-E1 cells were found in larger amounts than those of the controls (Fig. 6j and Supplementary Fig. 14g). Importantly, these effects were completely reversed by a synthetic peptide, GRGDNP (Gly-Arg-Gly-Asp-Asn-Pro), which inhibited fibronectin-integrin binding (Fig. 6k).

Furthermore, the FN1 inhibitory peptide also inhibited the accumulation of TBI sEVs in bone (Fig. 6l). Taken together, these results suggested that FN1 directed sEV targeting of osteoprogenitors and bone.

**Hydrogels carrying miR-328a-3p-containing sEVs efficiently promote bone formation, and repair bone defects**. To identify the therapeutic potential of osteogenic sEV and miRNAs released from the damaged brain to enable bone healing, miR-328a-3p/negative control mimics were electrotransferred into sEVs and embedded into the hydrogel. To load the sEVs with miR-328a-3p/negative control mimics, 100 μg of purified sEVs and nanoparticle-miRNA complexes (final concentration of 100 nM for miR-328a-3p/negative control mimics) were gently mixed in 100 μL of phosphate-buffered saline (PBS) at 4 °C. After electroporation at 200 V for 5 ms in electroporation cuvettes, the mixture was incubated at 37 °C for 30 min. We found that hydrogel carrying sEVs containing miR-328a-3p efficiently promoted osteoblast differentiation and mineralization in vitro, as manifested by a marked increase in ALP and Alizarin red staining in a hydrogel-embedded MC3T3-E1 3D model (Fig. 7a–d).

The hydrogel carrying sEV miR-328a-3p was then applied to the rat PTD model. Bone healing was detected at day 7 and day 14 post-surgery. The results showed that BV/TV%, Tb.N, OCN and FOXO4 protein levels were enhanced by sEV miR-328a-3p (Fig. 7e–g and Supplementary Fig. 15b–e). PTD treated with sEV miR-328a-3p hydrogel was replaced with newly formed bone, 14 days post-surgery, which was much faster than that of controls (Fig. 7e, h). No significant change in the number of osteoclasts TRAP-positive cells (Supplementary Fig. 15f, g) and osteoclast surface per bone surface (Supplementary Fig. 15h) was observed. These results suggested that sEV miR-328a-3p efficiently promoted bone healing in rats.

To determine the effects of hydrogel carrying sEV miR-328a-3p on the repair of calvarial defects, one of the most challenging orthopedic problems, a critical-sized rat calvarial defect model was generated and treated with the hydrogel. sEV release kinetics from hydrogel was showed (Supplementary Fig. 16a). Micro-CT analysis showed that the size of the remaining defect in rats treated with sEV-miR-328a-3p-embedded hydrogel was much smaller than that of controls, 1 (Supplementary Fig. 16b) and 3 (Fig. 7i) month post-surgery, and the BV/TV% and Tb.N increased up significantly (Fig. 7j, k and Supplementary Fig. 16c, d). HE staining showed that the calvarial defect was almost repaired by the hydrogel at 3 months post-surgery (Fig. 7l).

Taken together, these findings indicated that sEV miR-328a-3p containing hydrogel had potent potential to repair bone defects, without the involvement of any additional exogenous factors or cells.

## Discussion

Although it has long been known in clinical medicine that TBI accelerates bone healing, the underlying mechanisms remain largely unknown. The present work showed that after TBI, injured neurons, especially those in the hippocampus, released osteogenic miRNAs-enriching sEVs, which are transferred to bone through the circulatory system, and target osteoprogenitors to promote osteogenesis and accelerate bone healing, suggesting a paradigm of brain-to-bone interaction during bone homeostasis. We found that miR-328a-3p and miR-150-5p enriched in plasma sEVs after TBI were strong osteogenic miRNAs, and hydrogel carrying miR-328a-3p-containing sEVs had potent potential to repair bone and treat delayed union and nonunion fractures. Moreover, we found that increased FN1 expression on sEVs surfaces contributed to the direction of TBI sEVs to the bone. Hence, it is plausible that modification of sEVs surface FN1 could be used in bone-targeted drug delivery, providing an avenue for bone-targeted therapy. Taken together, our findings have established a clear link between the damaged brain and bone formation both in animals and clinical settings (Fig. 8).

Clinical management of the bone fracture is oriented to attain bone healing with the best possible functional recovery in the shortest possible time frame. However, in about 10% of cases, fracture healing can go awry in the form of delayed or nonunion fractures, which has a significant effect on the quality of life of patients[22,23]. There are currently no approved pharmacological

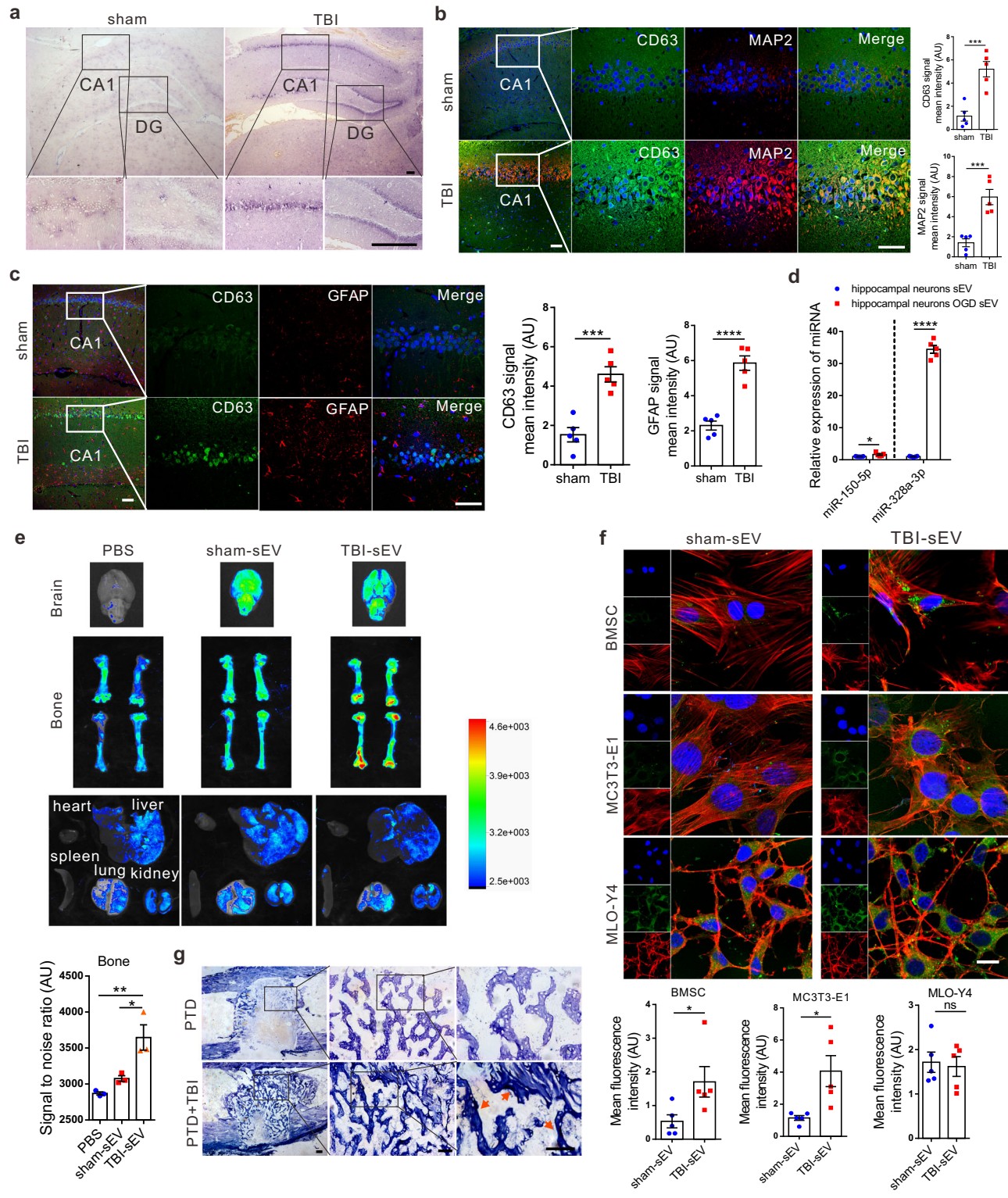

agents for the treatment of established bone nonunion, or for the acceleration of fracture healing[23]. A better understanding of molecular mechanisms underpinning fracture healing and seeking innovative strategies that efficiently accelerate bone regeneration is essential for the treatment of delayed bone healing or bone non-union. The process of fracture healing is a complex process that requires many kinds of cells and factors. We studied bone repair with a monocortical defect model which has been used in many previous research[24–29]. Although this repair model recapitulates

most steps of fracture healing, however, the majority of clinical fractures heal through secondary bone formation (endochondral ossification). Further studies using stabilized fracture model should be performed to verify its potential application. This study identified osteogenic miRNA-containing sEVs, which strongly accelerated bone healing, representing a potential strategy for the treatment of delayed or nonunion fractures.

Bone homeostasis relies on the transfer of active molecules between cells, and compelling evidence has emerged to show the

**Fig. 5 Injured neurons produce sEVs that target bone and osteoprogenitors. a** In situ hybridization with a digoxigenin-labeled probe of miR-328a-3p in rat hippocampus after TBI. Scale bar, 200 µm, ($n = 5$ animals per group). **b** Representative confocal images and quantitation of immunocytofluorescence staining of CD63 (green), dendritic marker MAP2 (red), and DAPI (blue) in CA1 region of the rat hippocampus ($n = 5$; Student's two-sided unpaired $t$ test). $P_{(CD63\ TBI\ vs.\ sham)} = 0.0009$, $P_{(MAP2\ TBI\ vs.\ sham)} = 0.0007$. Scale bars: 50 µm. **c** Representative confocal images and quantitation of immunocytofluorescence staining of CD63 (green), astrocyte marker GFAP (red), and DAPI (blue) in CA1 region of the rat hippocampus ($n = 5$; Student's two-sided unpaired $t$ test). $P_{(CD63\ TBI\ vs.\ sham)} = 0.0004$, $P_{(GFAP\ TBI\ vs.\ sham)} < 0.0001$. Scale bars, 50 µm. **d** The profile of miR-328a-3p and miR-150-5p in rat hippocampal neuron sEVs 24 h after oxygen glucose deprivation(OGD) treatment ($n = 5$; Student's two-sided unpaired $t$ test). $P_{(miR-150-5p\ hippocampal\ neurons\ OGD\ sEV\ vs.\ hippocampal\ neurons\ sEV)} = 0.0381$. $P_{(miR-328a-3p\ hippocampal\ neurons\ OGD\ sEV\ vs.\ hippocampal\ neurons\ sEV)} < 0.0001$. **e** Representative biophotonic images of the organ distribution of fluorescence signal in rats 24 h after ICV brain infusion with purified PKH67-labeled sEVs isolated from rat plasma of sham, TBI groups or PBS only ($n = 3$; one-way analysis with Turkey's multiple comparisons test was performed). $P_{(PBS\ vs.\ TBI-sEV)} = 0.0047$, $P_{(sham-sEV\ vs.\ TBI-sEV)} = 0.0199$. **f** Representative confocal images of BMSCs, MC3T3-E1, and MLO-Y4 cells incubated with PKH67-labeled rat plasma sEVs (green) for 12 h. Cells were fixed and stained with phalloidin for F-actin (red) and DAPI (blue) staining for nuclei ($n = 5$, Student's two-sided unpaired $t$ test). $P_{(BMSCs\ TBI-sEV\ vs.\ sham-sEV)} = 0.0448$, $P_{(MC3T3-E1\ TBI-sEV\ vs.\ sham-sEV)} = 0.0175$, $P_{(MLO-Y4\ TBI-sEV\ vs.\ sham-sEV)} = 0.7698$. Scale bars, 20 µm. **g** Representative in situ hybridization images showing the expression of miR-328a-3p in rat PTD sites, ($n = 5$ animals per group per time point). Scale bars: 200 µm. The quantitation result were plotted as dot plots, showing the mean ± SEM of three independent experiments. $^{*}P < 0.05$, $^{**}P < 0.01$, $^{***}P < 0.001$, $^{****}P < 0.0001$.

regulatory activities of sEVs in bone remodeling by transfer-specific sEV miRNA and proteins[30,31]. sEVs with their cargo miRNAs are key components of the fracture repair process, which is just beginning to be understood. During fracture healing, BMSC-derived sEVs contribute to bone remodeling in a mouse model[32]. However, the network of regulatory activities of sEVs in bone homeostasis and their therapeutic potential in bone injury remain largely unknown. Furthermore, published reports have shown that sEVs are exchanged mainly among osteoblasts, osteoclasts, and osteocytes and their precursors in bone remodeling; the role of sEVs released by distant cells, such as neurons in this process is unclear[33–37]. Here we demonstrated that miR-328a-3p and miR-150-5p in plasma sEVs secreted by damaged neurons promote osteogenesis and bone repair by targeting osteoprogenitors. Interestingly, previous studies have found that patients' serum miR-328-3p level increased significantly in response to fracture[38], and upregulated miR-328-3p level was also detected in serum and cerebrospinal fluid samples of patients with TBI[39], supporting our findings that miR-328a-3p is an osteogenic miRNA related to both bone fracture and TBI. *FoxO4* is ubiquitously expressed in mammalian tissues, including bone[40]. Mice with deletion of *FoxO4* in osteoblast progenitors have high bone mass caused by increased osteoblast number and bone formation[40]. The anti-osteogenic actions of *FoxO4* result from the binding of FoxO4 to β-catenin and the prevention of the association between β-catenin and TCF/Lef transcription factors and, thereby Wnt/β-catenin transcriptional activity[41]. Recent studies indicate that E3 ubiquitin ligases Cbl proteins regulate osteoprogenitor cell proliferation, differentiation, and survival through ubiquitination and degradation of receptor tyrosine kinases and other molecules[42,43]. Our findings identified that *FOXO4* and *CBL* are direct and functional target genes of miR-328a-3p and miR-150-5p, respectively, emphasizing the crucial role of these miRNAs in bone formation. Together, these observations are compelling, as circulating, miRNAs are not only biomarkers but also functional constituents in the intricate crosstalk between the injured brain and the remote bone healing processes.

Previous studies have found that extracellular matrix protein FN plays an important role in the targeting of sEVs[44,45]. One of the unique features of FN is its ability to bind a large number of cell adhesion receptors, growth factors, and extracellular matrix proteins. Liver endothelial cells release SK1-containing sEVs, which engage with hepatic stellate cells via FN-integrin-dependent adhesion[45]. Microvesicles released by human cancer cells can be taken up by fibroblasts in an FN-dependent manner[44]. All these studies, however, were carried out in vitro, and, therefore, which tissues or organs the sEVs with FN will eventually reach in vivo is unknown. In this study, we found that

following TBI, circulating sEVs and their cargo miRNAs were transferred selectively to peripheral organs and preferentially to the bone. Furthermore, the amount of FN1 on sEVs increased after TBI, which contributed to their osteogenic function. These could be, thus, used in bone-targeted drug delivery, providing an avenue for bone-targeted therapy.

## Methods

**Patients.** From January 2016 to January 2019, 24 patients were admitted to the Third Affiliated Hospital of Southern Medical University. All patients were fully informed of the study, and they provided written consent prior to participation. Among the 24 patients, there were 12 patients with simple shaft fractures and 12 patients with shaft fractures and concomitant TBIs who were assigned to three groups (week 2, week 4, and week 6, four patients per group). On admission, patient inclusion criteria were severe cranial trauma with a Glasgow Coma Scale score ≤ 8 and head abbreviated injury score > 2. Exclusion criteria included all forms of prior neurological pathology or bone-related diseases, immunosuppression, rheumatoid arthritis, and diabetes, as well as steroid or bisphosphonate therapy. Plasma sEVs were collected from TBI patients (10 males) and healthy control (10 males) between the ages of 17 and 72 admitted to the Third Affiliated Hospital of Southern Medical University. Venous blood samples were collected from all patients immediately after hospital admission (always within 8 h after trauma). Healthy matched individuals were selected among standard plasma samples as control. They were trauma-free and matched for age and sex status. No subjects had histories of neurological disease, psychiatric illness, head injury, stroke, nor learning disabilities. We confirm that our study is compliant with the "Guidance of the Ministry of Science and Technology (MOST) for the Review and Approval of Human Genetic Resources". Ethics approval was granted by the Human Research Ethics Committee of the Third Affiliated Hospital of Southern Medical University, and all patients or next of kin gave informed consent prior to the study. The study was conducted in accordance with the principles and guidelines of The Declaration of Helsinki.

All patients sustained an isolated shaft fracture of the femur. The long bone fractures were imaged by conventional radiographs in anteroposterior and lateral projections. The fractures in all patients were treated. All femoral fractures were stabilized with an intramedullary nail with the use of a reamed technique. The two groups were not significantly different in terms of the time interval of surgical treatment for the bone fracture.

**Rat models.** All animal experimental protocols were approved by the Animal Care and Use Committee of the Southern Medical University, Guangzhou, China (SYXK (Guangdong) 2016-0167). None of the authors are members of this committee. The care of animals was in accordance with the guidelines of the US National Institutes of Health (NIH) and the Chinese National Institute of Health. Male SD rats (body weight, 220–250 g) were housed individually under controlled environmental conditions with a 12 h light/dark cycle and were given unrestricted access to pellet food and water throughout the study. The animals were purchased from the Experimental Animal Center of the Southern Medical University in Guangzhou, China. All surgical interventions were performed under anesthesia with a mixture of 13.3% urethane and 0.5% chloralose (0.65 mL/100 g body weight, ip), using a standardized protocol established in our laboratory. All efforts were made to reduce the number of animals used and to minimize animal discomfort.

**Cell lines.** The BMSC were harvested from the femora and tibiae of SD rats. Femurs and tibiae were aseptically removed, and bone marrow was flushed with PBS using a sterile syringe. Bone marrow samples were centrifuged for 5 min at

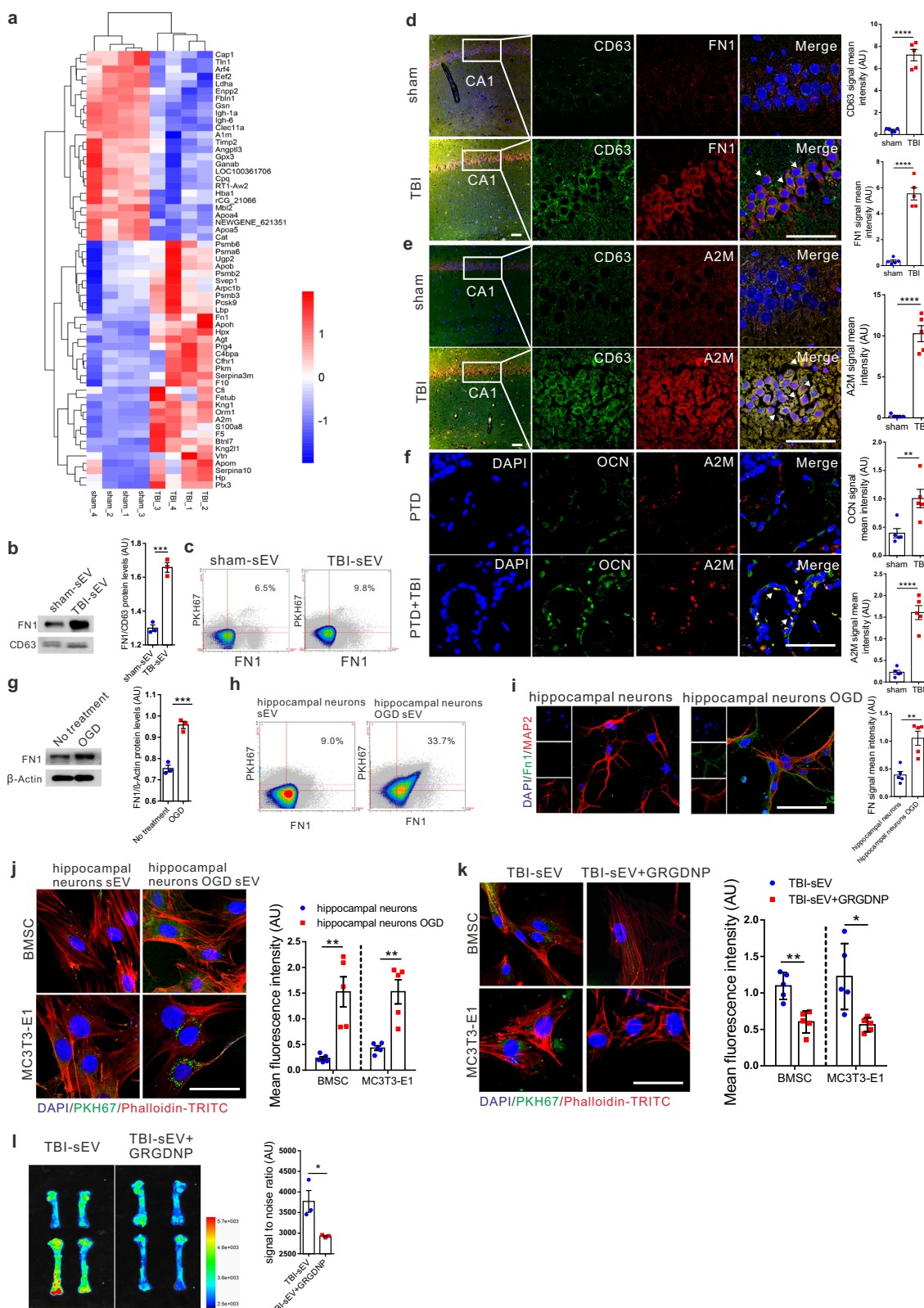

1000 × g and then placed in minimum essential medium alpha (α-MEM; Gibco, Gaithersburg, MD, USA) and replenished with 10% fetal bovine serum (FBS, Gibco, Carlsbad, CA, USA) and 1% penicillin and streptomycin (Thermo), then incubated at 37 °C in an incubator containing 5% CO$_2$ for 48 h. Subsequently, the nonadherent cells were discarded, and the adherent cells were allowed to grow to 80% confluence. These cells were defined as passage one cells (P1). P3 cells were used for all experiments. The MC3T3-E1 cells were purchased from American Type Culture Collection and grown in α-MEM (Invitrogen, Waltham, MA, USA)

containing 10% FBS and 1% penicillin and streptomycin. For induction of osteoblastic differentiation, MC3T3-E1 cells were incubated in an osteogenic medium (α-MEM, 10% FBS, 10 mM β-glycerophosphate, and 50 μg/mL ascorbic acid) and treated with 1 μM dexamethasone (DEX) (Sigma-Aldrich, St. Louis, MO, USA) or vehicle control. The HT22 cells were purchased from American Type Culture Collection and were cultured in DMEM (Invitrogen, Waltham, MA, USA) with 10% FBS. The cell lines were not authenticated after purchase but routinely tested negative for mycoplasma contamination.

**Fig. 6 Protein FN1 directs sEVs to target osteoprogenitors and bone. a** The heat map shows protein abundance in sEVs isolated from TBI rat plasma and the sham group by TMT analyses. The decrease and increase in proteins are indicated by range of green and red intensities, respectively. **b**, **g** Western blot analysis of FN1 in sEVs isolated from rat plasma or from primary culture of rat hippocampal neurons after OGD (oxygen–glucose deprivation) treatment. $P_{(TBI-sEV \ vs. \ sham-sEV)} = 0.0005$, $P_{(OGD \ vs. \ No \ treatment)} = 0.0008$. **c**, **h** sEVs isolated from plasma and primary culture of rat hippocampal neurons supernatant 24 h after OGD treatment were labeled with PKH67, 10 μg/mL of monoclonal antibody FN1-PE, and analyzed by nanoscale Apogee flow cytometer (FC). The MV gate was determined using size-calibrated green fluorescent silica beads ($n = 3$; Student's two-sided unpaired $t$ test). **d**, **e** Immunofluorescent images and quantitation of CA1 region of rat hippocampus stained with CD63 (green), FN1 or A2M (red), together with DAPI (blue) staining for nuclei ($n = 5$; Student's two-sided unpaired $t$ test). Scale bar, 50 μm. $P_{(CD63 \ TBI \ vs. \ sham)} < 0.0001$, $P_{(FN \ TBI \ vs. \ sham)} < 0.0001$, $P_{(A2M \ TBI \ vs. \ sham)} < 0.0001$. **f** Representative immunofluorescent images and quantitation of OCN (green), A2M (red) of new bone formation at defect sites in PTD + TBI and PTD only group, together with DAPI (blue) for nuclei ($n = 5$, Student's two-sided unpaired $t$ test). $P_{(OCN \ TBI \ vs. \ sham)} = 0.0099$, $P_{(A2M \ TBI \ vs. \ sham)} < 0.0001$. Scale bar, 50 μm. **i** Representative confocal images of immunocytofluorescence staining of MAP2 (red) and FN1 (green), together with DAPI (blue) for nuclei in primary culture of rat hippocampal neurons 24 h after OGD treatment ($n = 5$; Student's two-sided unpaired $t$ test). $P_{(hippocampal \ neurons \ OGD \ vs. \ hippocampal \ neurons)} = 0.0017$. Scale bar, 50 μm. **j** BMSC and MC3T3-E1 cells were incubated with PKH67-labeled sEVs released by primary hippocampal neurons after OGD treatment and subjected to immunofluorescence of PKH-67-labeled sEVs (green), F-actin (red), and DAPI (blue) for nuclei ($n = 5$; Student's two-sided unpaired $t$ test). $P_{(BMSCs \ hippocampal \ neurons \ OGD \ vs. \ hippocampal \ neurons)} = 0.0024$, $P_{(MC3T3-E1 \ hippocampal \ neurons \ OGD \ vs. \ hippocampal \ neurons)} = 0.0018$. Scale bar, 50 μm. **k** BMSC and MC3T3-E1 cells were incubated with PKH67-labeled TBI-sEV or PKH67-labeled TBI-sEV with peptide Gly-Arg-Gly-Asp-Asn-Pro (GRGDNP) for 6 h and subjected to immunofluorescence for PKH-67-labeled sEVs (green) and phalloidin for F-actin (red), DAPI (blue) for nuclei ($n = 5$; Student's two-sided unpaired $t$ test). $P_{(BMSCs \ TBI-sEV+GRGDNP \ vs. \ TBI-sEV)} = 0.0017$, $P_{(MC3T3-E1 \ TBI-sEV+GRGDNP \ vs. \ TBI-sEV)} = 0.0127$. Scale bar, 50 μm. **l** Representative biophotonic images of fluorescence signal in rat femurs and tibiae at 6 h after intracerebroventricular injection with purified PKH67-labeled TBI-sEV with or without GRGDNP ($n = 3$; Student's two-sided unpaired $t$ test). $P_{(TBI-sEV+GRGDNP \ vs. \ TBI-sEV)} = 0.0334$. The quantitation result were plotted as dot plots, showing the mean ± SEM of three independent experiments. $^*P < 0.05$, $^{**}P < 0.01$, $^{***}P < 0.001$, $^{****}P < 0.0001$.

**Oxygen–glucose deprivation (OGD)**. Primary cultured hippocampal neurons and mouse neuron HT22 cells were subjected to OGD in vitro to simulate ischemia-like conditions. Briefly, the culture medium was replaced by glucose-free EBSS, and then cells were placed into an oxygen-deprived (94% $N_2$, 1% $O_2$, and 5% $CO_2$) incubator for 2 h at 37 °C. The EBSS was then replaced with a regular medium and placed back into an incubator at normal conditions to undergo reperfusion for 24 h.

**Lateral fluid-percussion (LFP) brain injury and GW4869 administration**. LFP was performed as previously described[46]. Briefly, anesthetized rats were placed in a stereotaxic frame. After incision of the scalp, the temporal muscles were reflected, and a 5 mm craniotomy was drilled (2.5 mm lateral to the sagittal sinus and centered between the bregma and lambda), keeping the dura mater intact. A hollow female Luer-Lok fitting was placed directly over the dura and rigidly fixed using dental cement. Before the induction of trauma, the female Luer-Lok was connected to the fluid percussion injury device via a transducer (Biomedical Engineering Facility, Medical College of Virginia, USA). For the infliction of TBI, a metal pendulum was released from a pre-selected height, thus leading to a rapid injection of normal saline into the closed cranial cavity. A pulse of increased intracranial pressure of 21–23 ms duration was elicited, controlled, and recorded by an oscilloscope (Agilent 54622D, MEGA Zoom; Germany). The severity of the injury could be altered by adjusting the amount of force generated by the pendulum. For the present experiment, a moderate severity injury was induced (2.5 ± 0.2 atmospheres). Sham-treated animals underwent identical preparatory procedures, including craniotomy, but were not injured. We used GW4869, a compound used to inhibit sEV secretion in CNS of rats. GW4869 was dissolved in dimethylsulfoxide (DMSO) at 8 mg/mL. The working solution was prepared in 0.9% normal saline, freshly made before use with a final concentration of 0.3 mg/mL (2.5 μg/g body weight), with 3.75% DMSO saline as a control. Rats were injected with the working solution twice a week.

**Rat tibial bone defect and calvarial defect model for bone generation**. The male SD rats weighing approximately 250 g were used in the experiments. Each rat was anesthetized with an inhalation of isoflurane followed by an intraperitoneal injection of sodium pentobarbital (50 mg/kg). The proximal site of the right tibia was exposed after skin incision, and the periosteum was retracted to expose the cortical bone surface. A circular unicortical defect, 2 mm in diameter, was created on the anteromedial surface of the proximal metaphysis using a burr drill. The overlying muscle and skin were closed in layers. A 5 mm full-thickness calvarial defect was created in male SD rats on the right side of the parietal bone with a trephine drill. Each defect was syringed with sterile saline solution to remove bone debris and then implanted with methacrylated glycol chitosan (MeGC) hydrogels containing various nanoparticles (200 μg/mL) or left empty. The hydrogels were formed by exposing 40 μL of the suspension under visible blue light (400–500 nm, 500–600 mW/cm$^{-2}$; Bisco, Schaumburg, IL, USA) in the presence of a photo-initiator, riboflavin (final concentration 6 μM). After surgery, all rats received analgesics and were allowed to move freely in their cages with access to standard rat chow and tap water. To get operative treatment, all animals received analgesia with subcutaneous injections of buprenorphine with a concentration of 0.1 mg/kg for 3 days. Based on previous work with rat tibial bone defects, the time periods chosen were 7 and 14 days. For cranial defects, the time periods chosen were 4 and

12 weeks. At both time points, the samples were evaluated with microCT and histology after harvest. To investigate in vivo bone regeneration, rats were intraperitoneally injected with 15 mg/kg of calcein to mark the new bone 1 and 3 weeks after the surgery. When the rats were sacrificed by an overdose of pentobarbitone sodium at week 4, the defected tibia was harvested. Then, the tibia was embedded in polymethylmethacrylate, and undecalcified sections were acquired using a Leica diamond saw (Leica SP1600). A confocal laser scanning microscope (Leica CLSM) was used to observe calcein-labeled bone.

**Cell counting kit-8 assay**. Cell Counting Kit-8 assay (CCK-8) was used to determine cell viability. Totally $4 \times 10^3$ cells were seeded in 96-well plates and allowed to adhere overnight. After incubation with compounds under evaluation for 72 h, 10 μL CCK-8 dye was added to each well, and cells were incubated for 1 h at 37 °C. Subsequently, the absorbance was determined at 450 nm. The cell viability was calculated by Eq

$$\text{Cell viability} (\%) = (A_s - A_b)/(A_c - A_b) \times 100\%$$

where $A_s$ is the absorbance of the well-containing cell, culture medium, CCK-8 solution, and sample; $A_c$ is the absorbance of the well-containing cell, culture medium, and CCK-8 solution; $A_b$ is the absorbance of well without cell, culture medium, CCK-8 solution, and sample. The results were assayed in triplicate experiments.

**Primary culture of rat hippocampal neurons**. SD rats (purchased from Southern Medical University (Guangzhou, China)) younger than 12 h old were sterilized with 75% ethanol, and the hippocampus tissues were separated under aseptic conditions after they were anesthetized by intraperitoneal injection with 10% chloral hydrate (dose: 5 mL/kg). The study was approved by the Animal Care and Use Committee of the Southern Medical University (Guangzhou, China). Briefly, the hippocampus was removed from the brain under sterile conditions and minced into 0.2–0.5 cm size. Then the tissue was digested by 0.125% (w/v) trypsin (Invitrogen, Carlsbad, CA, USA) for 20 min at 37 °C. Disassociated neurons were collected by centrifugation at 1000×g for 10 min. Primary hippocampal neurons were plated onto poly-D-lysine-coated, 12-mm diameter glass coverslips at 30 × 10$^3$ cells/dish, cultured in Neurobasal A media (Invitrogen) supplemented with 2% B27 supplement and 0.5 mM L-glutamine (all from Invitrogen). Cells were incubated at 37 °C, 5% $CO_2$ in a humidified cell culture incubator. The media was changed every 3–4 days. Primary neurons were selected for experiments after maintaining in culture for 7 days.

**Plasma samples**. Twenty-four-hour post TBI surgery, all rats were anesthetized and peripheral blood was collected in a tube containing EDTA as an anticoagulant. The samples were stored at 4 °C for a short time and immediately centrifuged at 4 °C and 3000 rpm for 10 min. The supernatant was carefully collected and transferred to a new tube without disturbing the intermediate buffy coat layer. In total, 20 samples (10 from the TBI group and 10 from the sham group) were used to isolate plasma. All samples were analyzed in triplicate under all conditions.

**sEV labeling and cellular uptake**. sEVs were labeled with PKH67 Green Fluorescent Cell Linker Kit (Sigma-Aldrich) by following the manufacturer's protocol.

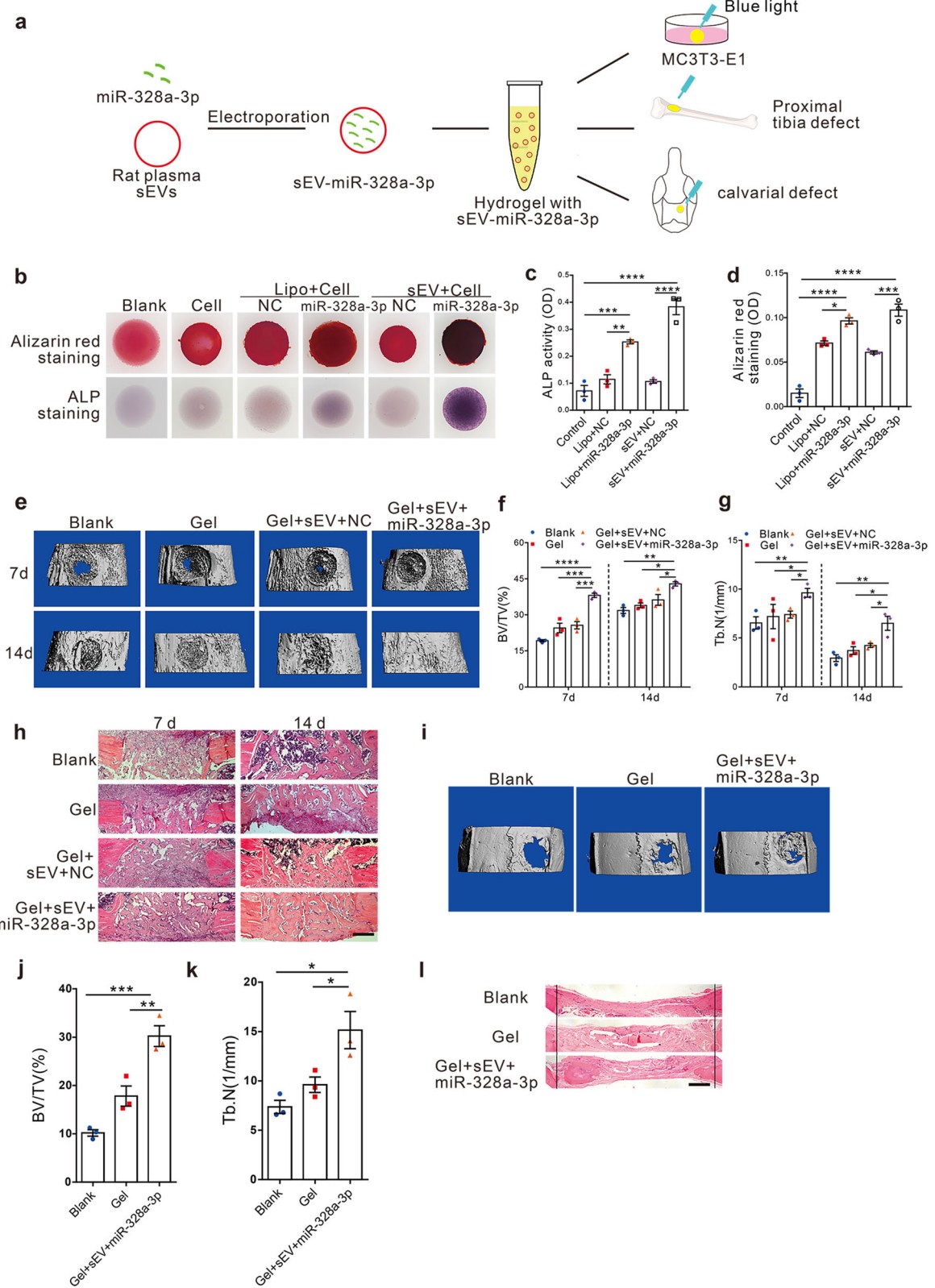

The isolated sEVs diluted in PBS was added to 0.5 mL of Diluent C. Two μL of PKH67 dye was added and incubated for 4 min at room temperature. Two mL of 1% BSA/PBS was added to bind excess dye. The synthetic peptide, GRGDNP (300 μg/mL) was incubated with labeled sEVs (50 μg/mL) for 30 min. The labeled sEVs were washed at 100,000 × g for 70 min, and the sEV pellet was suspended in PBS and used for uptake experiments. We then cocultured these PKH67 sEVs with BMSC and MC3T3-E1. After the indicated time of co-culture, we stained BMSC, MC3T3-E1 with DAPI (Sigma-Aldrich) and observed them with confocal microscopy.

**RNA isolation**. RNA was extracted from cells using the QIAzol Lysis Reagent (Qiagen, Hilden, Germany) according to the user guidelines. Briefly, the cells were collected in a reaction tube, lysed with QIAzol, and mixed with chloroform. After being centrifuged at 12,000 × g for 15 min at 4 ℃, the upper aqueous phase was transferred to an RNeasy Mini spin column in a 2 mL collection tube and mixed with 100% ethanol. After being washed at 7500 × g for 5 min at 4 ℃, the total RNA was collected for qRT-PCR analysis. RNA was extracted from plasma and sEVs

**Fig. 7 Hydrogel carrying miR-328a-3p-containing sEVs efficiently promotes bone formation and repairs bone defects. a** Schematic representation of sEV loading and treatment model. Purified sEVs were loaded with miR-328a-3p through electroporation and encapsulated into hydrogels. Then, the hydrogels were injected into bone defect sites for treatment. **b** Alizarin Red staining of MC3T3-E1 in hydrogel culture with blank (no cell group), cell (MC3T3-E1 cell group), Lipo + cell NC (transfected with negative control in MC3T3-E1 cell group by liposome), Lipo + cell miR-328a-3p (transfected with miR-328a-3p mimics in MC3T3-E1 cell group by liposome), sEV + cell NC (electroporated with negative control in MC3T3-E1 cellgroup by sEVs), sEV+cell miR-328a-3p (electroporated with miR-328a-3p mimics in MC3T3-E1 cell group by sEVs) were carried out on day 28 for mineralization analysis and ALP staining on day 7. The OD values of ALP activity (**c**) and mineralization (**d**) were evaluated for quantitation ($n = 3$; one-way analysis with Turkey's multiple comparisons test was performed). $P_{(ALP\ Lipo+NC\ vs.\ Lipo+miR-328a-3p)} = 0.0023$, $P_{(ALP\ Control\ vs.\ Lipo+miR-328a-3p)} = 0.0003$, $P_{(ALP\ sEV+NC\ vs.\ sEV+miR-328a-3p)} < 0.0001$, $P_{(ALP\ Control\ vs.\ sEV+miR-328a-3p)} < 0.0001$, $P_{(Alizarin\ Red\ Control\ vs.\ Lipo+miR-328a-3p)} < 0.0001$, $P_{(Alizarin\ Red\ Lipo+NC\ vs.\ Lipo+miR-328a-3p)} = 0.015$, $P_{(Alizarin\ Red\ sEV+NC\ vs.\ sEV+miR-328a-3p)} = 0.0001$, $P_{(Alizarin\ Red\ Control\ vs.\ sEV+miR-328a-3p)} < 0.0001$. **e** 3D Micro-CT images of the proximal tibial defect regions of rats treated with hydrogels (Gel group), encapsulated with sEV negative control (Gel + sEV + NC group), encapsulated with sEV miR-328a-3p (Gel + sEV + miR-328a-3p group) or left empty (Blank group) 7 and 14 days postsurgery. Quantitative data of BV/TV (**f**) and Tb.n (**g**) were obtained from micro-CT analysis at day 7 and day 14 post-treatment ($n = 3$; one-way analysis with a Turkey's multiple comparisons test was performed). $P_{(7d\ BV/TV\ Blank\ vs.\ Gel+sEV+miR-328a-3p)} < 0.0001$, $P_{(7d\ BV/TV\ Gel\ vs.\ Gel+sEV+miR-328a-3p)} = 0.0005$, $P_{(7d\ BV/TV\ Gel+sEV+NC\ vs.\ Gel+sEV+miR-328a-3p)} = 0.0009$, $P_{(14d\ BV/TV\ Blank\ vs.\ Gel+sEV+miR-328a-3p)} = 0.0029$, $P_{(14d\ BV/TV\ Gel\ vs.\ Gel+sEV+miR-328a-3p)} = 0.0106$, $P_{(14d\ BV/TV\ Gel+sEV+NC\ vs.\ Gel+sEV+miR-328a-3p)} = 0.0474$, $P_{(7d\ Tb.n\ Gel+sEV+miR-328a-3p\ vs.\ Blank)} = 0.0079$, $P_{(7d\ Tb.n\ Gel+sEV+miR-328a-3p\ vs.\ Gel)} = 0.0285$, $P_{(7d\ Tb.n\ Gel+sEV+NC\ vs.\ Gel+sEV+miR-328a-3p)} = 0.0434$, $P_{(14d\ Tb.n\ Gel+sEV+miR-328a-3p\ vs.\ Blank)} = 0.0026$, $P_{(14d\ Tb.n\ Gel+sEV+miR-328a-3p\ vs.\ Gel)} = 0.0114$, $P_{(14d\ Tb.n\ Gel+sEV+NC\ vs.\ Gel+sEV+miR-328a-3p)} = 0.0335$. **h** Representative H&E staining of proximal tibial defect regions at day 7 and day 14 post-impact, ($n = 5$ animals per group per time point). Scale bars: 200 μm. 3D Micro-CT image (**i**) and quantitative data of BV/TV % (**j**) and Tb.N (**k**) were obtained from micro-CT analysis of calvarial defects of rats treated with hydrogels (Gel group), encapsulated with sEV miR-328a-3p (Gel + SEV + miR-328a-3p) or left empty (Blank group) 3 months postsurgery. ($n = 3$; one-way analysis of variance with Turkey's multiple comparisons test was performed). $P_{(BV/TV\ Blank\ vs.\ Gel+sEV+miR-328a-3p)} = 0.0005$, $P_{(BV/TV\ Gel\ vs.\ Gel+sEV+miR-328a-3p)} = 0.0062$, $P_{(Tb.N\ Blank\ vs.\ Gel+sEV+miR-328a-3p)} = 0.0104$, $P_{(Tb.N\ Blank\ vs.\ Gel+sEV+miR-328a-3p)} = 0.0448$. **l** Representative H&E staining to analyze bone regeneration in calvarial defects, ($n = 3$ animals per group). Scale bar: 200 μm. Two black vertical lines were drawn for the ease of observation. All quantitation results were plotted as dot plots, showing the mean ± SEM of three independent experiments. $^*P < 0.05$, $^{**}P < 0.01$, $^{***}P < 0.001$. $^{****}P < 0.0001$.

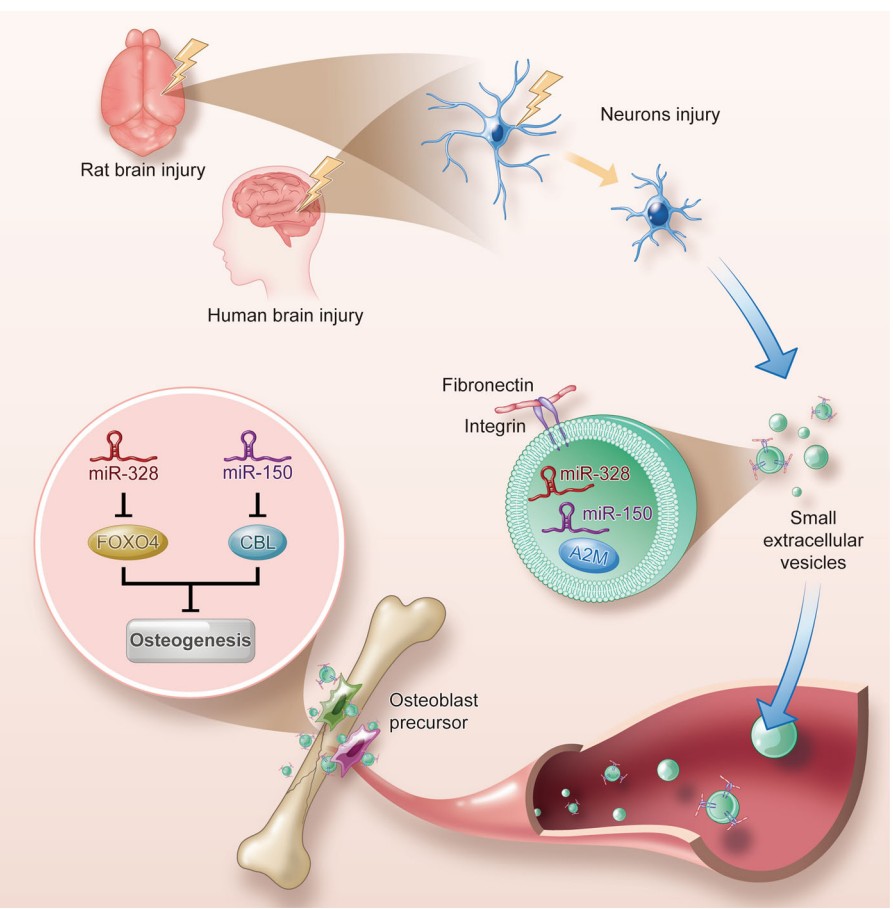

**Fig. 8 Scheme of the mode of damaged brain accelerates bone healing by releasing sEVs that target osteoprogenitors.** After TBI, the injured neurons released osteogenic miRNA-enriching sEVs, which were transferred to bone through the circulatory system, and target osteoprogenitors to promote osteogenesis and accelerate bone healing. The increased FN1 expression on sEVs surface contributed to direct sEVs to target osteoprogenitors. miR-328a-3p and miR-150-5p enriched in plasma sEVs after TBI were strong osteogenic miRNAs, which had potent potential to repair bone by directly targeting *FOXO4* and *CBL* 3'-UTR, respectively.

fractions using Qiagen miRNeasy Mini kit (Qiagen), according to the manufacturer's instructions, with a final elution volume of 50 µL.

**RT-PCR and qRT-PCR**. RNA samples were quantitated and qualified using NanoDrop analysis (ThermoFisher Scientific). Equal quantities (5 ng) of total RNAs from each sample were used for cDNA synthesis using the PrimeScript® RT reagent kit (TaKaRa, Tokyo, Japan). The reverse transcriptions of miRNAs were performed by looped miRNA-specific RT primers for miRNAs. qRT-PCR was performed an Applied Biosystem StepOnePlus, using an SYBR Green I Real-Time PCR Kit (GenePharma) for miR-150-5p and miR-328a-3p. Dissociation curves were generated to ensure the specificity of each qRT-PCR reaction. The relative expression levels of miRNAs in each sample were calculated and quantified using the $2^{-\triangle\triangle Ct}$ method after normalization for expression of the positive control.

**TEM and nanosight tracking analysis**. The morphology of the sEVs was observed by TEM. Briefly, the sEVs suspension was mixed with an equal amount of 4% paraformaldehyde. After washing with PBS, 4% uranyl acetate was added for chemical staining of sEVs, and images were captured using a Hitachi H-7650 TEM (Hitachi, Tokyo, Japan). Nanosight tracking analysis (NTA) was performed to measure the size and the concentration of the isolated sEVs using NanoSight NS500 (Malvern, Westborough, MA, USA), according to the operating instructions, without any changes. The sEVs were diluted to be within the recommended concentration range. Five 60 s videos were captured for each sample during flow mode (camera settings: slider shutter 890, slider gain 146). Plasma-derived sEVs were diluted and loaded onto the NS500 instrument by a syringe at a constant flow of 20. The videos were analyzed with the NTA 3.2 software (Malvern). All measurements were performed at room temperature.

**MiRNA sequencing**. Total RNA extraction of the samples was conducted using the miRNeasy Serum/Plasma Kit (Qiagen), according to standard operating procedures provided by the manufacturer. The quality and integrity of the total RNA were examined using the Agilent 2100 Bioanalyzer and the RNA 6000 Nano LabChip Kit (Agilent, Santa Clara, CA, USA) with an RNA integrity number > 7.0. A sequencing RNA library was constructed by performing a 3′-end linker, a 5′-end linker, reverse transcription, amplification, cDNA library size selection, and purification steps using total RNA. Cluster generation and first stage sequencing primer hybridization were performed on the cBot of the Illumina HiSeq sequencer from Shanghai Biotechnology (Shanghai, China), according to the corresponding procedure in the cBot User Guide. Then, the sequencing reagent was prepared following the Illumina User Guide and the flow cell carrying the cluster was loaded into the machine. Single-end sequencing was performed using the single-read program. The sequencing process was controlled by the data collection software provided by Illumina that performed real-time data analyses.

**MiRNA transfection**. MC3T3-E1 and BMSC cells were cultured to 70–80% confluence and transfected with miR-328a-3p mimic (sense, 5′-CUGGCCCUCU-CUGCCCUUCCGU-3′ and antisense, 5′-GGAAGGGCAGAGAGGGCCAGUU-3′), miR-328a-3p inhibitor (sense, 5′-ACGGAAGGGCAGAGAGGGCCAG-3′), miR-150-5p mimic (sense, 5′-UCUCCCAACCCUUGUACCAGUG-3′ and antisense, 5′-CUGGUACAAGGGUUGGGAUAUU-3′), miR-150-5p inhibitor (sense, 5′-CACUGGUACAAGGGUUGGGAGA-3′), miRNA mimic negative control (sense, 5′-UUCUCCGAACGUGUCACGUUTT-3′ and antisense, 5′-ACGUGA-CACGUUCGGAGAATT-3′) miRNA inhibitor negative control (sense, 5′-CAGUACUUUUGUGUAGUACAA-3′) (GenePharma Inc, Shanghai, China) by Lipofectamine 3000 (Invitrogen), according to the manufacturer's instructions. Mimics of miR-328a-3p, miR-150-5p, and miRNA mimic negative control were used at a final concentration of 50 nm, inhibitors of miR-328a-3p and miR-150-5p were transfected at 100 nM concentration and incubated for 6 h. Subsequently, the medium was changed to normal culture medium to terminate transfection.

**Coculture experiments**. The well inserts with a 0.4 mm pore size filter (BD Falcon, Corning, NY, USA) for 6-well plates were used following the manufacturer's instructions. Primary hippocampal neurons were seeded into the well inserts with DMEM/F12 medium or after OGD treatment. BMSC and MC3T3-E1 cells were seeded into 6-well plates and induced into osteoblasts using complete α-MEM. After differentiation, osteoblasts were washed with PBS, and then co-cultured for 7 d according to the experimental protocol. All co-culture experiments were conducted in α-MEM with sEV-free FBS (Life Technologies, Carlsbad, CA, USA).

**Luciferase assays**. MC3T3-E1 and BMSC cells were cultured at $1 \times 10^5$ cells/well in 12-well plates. The cells were co-transfected with miR-328a-3p and miR-150-5p mimic (50 nM) or miRNA mimic negative control (50 nM) and 0.2 µg of psi-CHECK™-1-UTR. Transfection was performed using Lipofectamine 3000. After 48 h, cells were collected and luciferase activity was determined using the Dual-Luciferase reporter assay system (Promega, Madison, WI, USA) with the dual luciferase assay reporter-ready luminometer (Promega). The assays were performed in triplicate.

**In situ hybridization**. In situ hybridization was performed to determine the expression of miR-328a-3p using miRCURY LNA miRNA Detection, Optimization Kit 2 (miR-328a-3p) (Exiqon, Vedbæk, Denmark), according to the manufacturer's protocol. Sections were serially cut to 4 µm thicknesses, and after deparaffinization of coronal sections of rat brain from the TBI and sham group, the tissues were subjected to pepsin (1.3 mg/mL, Sigma-Aldrich) for 30 min. After washing in PBS, the slides were submerged in 99.7% ethanol and air-dried. Slides were hybridized with 40 nM miR-328a-3p in an incubation chamber at 37 °C for 16–18 h, followed by stringent washes with saline-sodium citrate buffer at 37 °C. For immunodetection, slides were blocked with digoxigenin blocking reagent (Roche, Mannheim, Germany) in maleic acid buffer containing 2% sheep serum at ambient temperature for 60 min and then incubated with sheep anti-digoxigenin conjugated to ALP (diluted 1:800 in blocking reagent) at ambient temperature for 60 min, resulting in dark-blue staining. The NBT/BCIP reaction mixture[1] was incubated within a humidified chamber at 37 °C for 2 h and the reaction was stopped with KTBT buffer, and then washed in PBS three times, dehydrated, and mounted for microscopy. Five random microscopic fields per section were evaluated at 20× original magnification and the integral optical density of each visual field was calculated.

**Protein extraction, digestion, and TMT labeling**. The sEVs proteins of samples were suspended in lysis buffer (8 M urea, 1% SDS with a protease inhibitor cocktail) and treated by ultrasound at 40 kHz, 40 W for 2 min, incubated on ice for 60 min. After centrifugation at $12,000 \times g$ for 30 min at 4 °C, the protein concentration was determined using a BCA Protein Assay Kit (Thermo Fisher Scientific).

*Proteins were digested as follows*. A hundred microgram of protein sample was taken and the volume was replenished to 90 µl with lysate. Tris(2–carboxyethyl) phosphine (TCEP) (final concentration of 10 mM) was added and the mixture was incubated at 37 °C for 60 min. Then, 40 mM iodoacetamide was added to the mixture, and the mixture was incubated at room temperature in the dark for 40 min. Prechilled acetone was added and incubated for 4 h at −20 °C before centrifugation at $1000 \times g$ for 20 min. All collected proteins were transferred to a new tube, followed by trypsin digestion with a substrate ratio of 1:50 (w/w) at 37 °C overnight according to the manufacturer's protocols. The digested peptides were labeled with a TMT Reagent kit according to the manufacturer's protocol (Thermo Fisher Scientific). After tagging for 2 h at room temperature, hydroxylamine was added for 15 min. Finally, all samples were pooled, desalted, and vacuum-dried.

**Reversed-phase chromatography with high pH separation**. The TMT-tagged peptides were mixed and reconstituted in UPLC loading buffer, followed by loading on an Acquity UPLC BEH C18 column (1.7 µm, 2.1 mm × 150 mm; Waters, Milford, MA, USA). The peptides were eluted at a flow rate of 200 µL/min with a linear gradient of 0–5% solvent B (80% acetonitrile (ACN) 20 mM NH₄HCO₂, pH 10) for 2 min; 5% solvent B for 15 min, 5–30% solvent B for 8 min; 30–36% solvent B for 3 min; 36–42% solvent B for 1 min; 42–100% solvent B for 1 min and hold for 8 min. A total of 20 fractions were collected based on peak types and times, which were combined into 10 fractions per sample. Each of them was dried by vacuum centrifugation.

**LC–MS/MS analysis**. The labeled peptides were dissolved in solvent A (2% ACN with 0.1% formic acid), and then centrifuged (4 °C, $12,000 \times g$, 20 min) and the supernatant was transferred to a sample tube. Nano LC–MS/MS was carried out using a Q-Exactive MS (Thermo Fisher Scientific) coupled online to the UPLC system (Thermo Fisher Scientific). The peptide was loaded onto a C18 column (75 µm × 25 cm, Thermo Fisher Scientific) at a flow rate of 300 nL/min. Peptides were eluted from the column by a linear gradient from 5% solvent B (80% ACN with 0.1% formic acid) to 23% solvent B for 40 min; 29% solvent B for 10 min; 48% solvent B for 7 min; 100% solvent B for 1 min and hold for 8 min, after which the mobile phase was returned to 0% solvent B for 5 min. The 20 precursors with the most intense signals were selected for higher-energy collisional dissociation (HCD) fragmentation with normalized collision energy, dynamically choosing the most abundant precursor ions from a scan (m/z 350–1300). HCD spectra were acquired in the Orbitrap with 60,000 resolution at an m/z 200 and the fixed first mass m/z 100.

**Database search and quantitative proteomics analysis**. The RAW data files were analyzed using Proteome Discoverer (Thermo Fisher Scientific, Version 2.4) against Uniprot Rattus norvegicus database (version 20200617; 35,779 sequences). The MS/MS search criteria were as follows: mass tolerance of 20 ppm for MS and 0.02 Da for MS/MS Tolerance, trypsin as the enzyme with two missed cleavage allowed, carbamidomethylation of cysteine, and the TMT of N-terminus and lysine side chains of peptides as fixed modification, and methionine oxidation as dynamic modifications. High confidence peptides were used for protein identifications by setting a target false discovery rate threshold of 1% at the peptide level. Identified proteins that had at least one unique peptide were used for protein identifications. The thresholds of fold changes (>1.2 or <0.83) and P-value < 0.05 were used to identify differentially expressed proteins. A total of 273 expressed proteins were

identified as belonging to the proteome of Solanum tuberosum. Then we found 36 upregulated and 68 downregulated proteins in group TBI compared with the sham group.

**Micro-CT scanning**. After the animals were euthanized, the tibiae of the animals in the experimental groups were fixed in 4% paraformaldehyde and analyzed at a resolution of 12 μm on a micro-CT Scanner (Viva CT40; Scanco Medical AG, Bassersdorf, Switzerland). The area used for analysis began at the defect site close to the femoral head and extended through the whole defect area proximally for 500 slices. Using two-dimensional data from scanned slices, three-dimensional structure and morphometry were conducted to calculate morphometric parameters defining microarchitecture, including bone volume/tissue volume (BV/TV) and trabecular number (Tb.N 1/mm). 3D images of the ROI were created using Skyscan CTvol software (Bruker microCT, Kontich, Belgium).

**Electroporation**. To load plasma sEVs with miR-328a-3p, a total protein concentration of purified sEVs (50 μg) and miR-328a-3p mimic (400 nanomoles) were gently mixed in 400 μL PBS at 4 °C. The mixtures were transferred into ice-cold electroporation cuvettes and electroporated at 200 V for 5 ms, and then the mixture was incubated at 37 °C for 30 min. Non-electroporated samples were used as negative controls. All electroporations procedures were performed on a NEPA21 Type II Electroporator (NEPA GENE, Chiba, Japan).

**Nanoscale flow cytometry analysis of sEVs**. Samples were analyzed using an A-50 Micro-PLUS flow cytometer (Apogee Flow Systems, Hertfordshire, UK). The sample flow rate was 1.5 μL/min and the time of acquisition was 60 s for all measurements. An illumination wavelength of 405 nm (70 mW) was used to detect scattered light by microparticles. Before sample analysis, calibration of flow cytometer was performed using a reference bead mix (ApogeeMix, Apogee Flow Systems), composed of a mixture of silica nanoparticles with diameters of 110, 180, 240, 300, 590, 880, and 1300 nm with a refractive index (RI) of 1.42, and 110 and 500 nm green fluorescent (excited by blue laser) polystyrene nanoparticles with an RI of 1.59 (latex) were used. The upper limit of the microvesicles gate was determined using size-calibrated green fluorescent silica beads of size 200 nm, and then applied to all samples. In order to exclude excessive background noise, a combined threshold on light scatter and green fluorescence was set above the background using PBS and an un-labeled sEV sample, respectively. Briefly, 50 μg of sEVs were incubated at 37 °C for 30 min with PKH67 Green Fluorescent Cell Linker Kit (Sigma-Aldrich), in the presence or absence of 10 μg/mL of PE-labeled FN1 (Novus Biologicals, Littleton, CO, USA). Titration of all detecting antibodies was performed with preconjugated clones and dilutions were determined from the original concentration (μg/mL) as provided by the manufacturer. Dilutions of preconjugated antibodies were also performed using 0.20 μm filtered PBS. Ten microlitres of sEV was incubated with 100 ng of FN1-PE in each experiment. The mixture was then transferred to 1 mL of PBS. After centrifugation, the pellet was blocked by incubation with 110 μL of 1 M glycine (100 mM final) for 30 min. sEVs were washed two times in PBS and resuspended in 100 μL of PBS. The data were analyzed using FlowJo, version 10.4 software (FlowJo, Ashland, OR, USA).

**Plasma sEV and cell supernatant sEV isolation**. sEVs were isolated from rat and human plasma by following recommended protocols[47,48]. The plasma was centrifuged at 3000×g for 15 min to remove cell debris and platelets. Then, 250 μL of platelet-free plasma was added to 63 μL of ExoQuick Exosome Precipitation Solution (SBI System Bioscience, Mountain View, CA, USA) and carefully mixed. After refrigeration at 4 °C for 30 min, the mixture was centrifuged at 1500×g for 30 min at 4 °C to sediment sEVs. The supernatant was collected as non-sEV component. For comparison, sEVs and the non-sEV component from equal amounts of plasma were used.

A multi-step centrifugation procedure was used to isolate culture supernatant sEVs, as described previously[49]. Briefly, the collected culture supernatant was pre-purified by centrifugation at 300 × g for 10 min at 4 °C to remove floating cells and debris, and then at 2000 × g for 20 min to obtain apoptotic bodies and at 20,000×g for 30 min at 4 °C to obtain microvesicles. At each step, the supernatants were transferred to new tubes and the pellets were immediately resuspended in either PBS or rinsed with PBS. The resulting supernatant was filtered through a 0.22-μm membrane to remove particles larger than 200 nm. sEVs were pelleted via ultracentrifugation at 100,000 × g for 70 min at 4 °C. Pelleted sEVs were resuspended in PBS and ultracentrifuged again. sEVs were used immediately or stored at −80 °C. TSG101, CD9, and Alix were detected by western blot as sEV markers. The protein concentration in sEVs was determined using a BCA protein assay kit (ThermoFisher Scientific, Paisley, UK).

**Staining and quantitation of ALP and Alizarin Red S**. BMSC and MC3T3-E1 cells were treated and induced for 14 days. After 14 days of induction, the medium was removed and rinsed with PBS followed by fixation in 4% paraformaldehyde for 30 min at room temperature. BMSC and MC3T3-E1 cells were washed with PBS twice and then incubated with ALP staining buffer (Beyotime Institute of Biotechnology, Beijing, China) at 37 °C in the dark for 30 min. Then, the reaction was

stopped by using distilled water, and the plate was dried before taking photographic images.

At predetermined time point, hydrogels were incubated in 10% formalin buffered in phosphate for 30 min, washed with PBS, and fixed in a solution consisting of NBT and BCIP stock solutions in ALP buffer (100 mM Tris, 50 mM MgCl₂, 100 mM NaCl, pH 8.5) for 2 h. ALP activity was quantified with an alkaline phosphatase colorimetric assay.

Hydrogels were fixed in 10% formalin buffered in phosphate for 20 min, washed with PBS, and incubated in 2% Alizarin Red S solution for 5 min. Then, the gels were washed with PBS with gentle shaking for 16 h, and the PBS was changed at least three times. The images were acquired using an Olympus SZX16 stereomicroscope (Tokyo, Japan).

**Histological evaluation**. The fixed tissues were decalcified using 10% EDTA solution with gentle shaking for 1 month. The EDTA solution was changed every day. Decalcified samples were embedded in paraffin and cut into sections of 5 μm thickness. The defect area around the grafted bone was measured and the percentage of the bone matrix within the callus was calculated by Image Pro Plus 6.0 software (Media Cybernetics, MD, USA) and compared with the control group. Under high power magnification, the new matrix formation was measured and compared among all samples and the control.

**Immunofluorescence staining and immunohistochemistry**. The injured-side cortices of the tibiae from rats were harvested at different times and were fixed using 4% paraformaldehyde at 4 °C for 48 h and decalcified in 0.5 M EDTA, pH 7.4, on a shaker for 1 month. The bone tissues were embedded in paraffin and 2–5 μm sagittal sections were prepared for histological analyses. After deparaffinization and rehydration, sections were incubated in citrate buffer (10 mM citric acid, pH 6.0) for 30 min at 90 °C or treated with 200 mg/mL proteinase K (Sigma-Aldrich) for 10 min at 37 °C to unmask antigen. Sections for immunohistochemistry were treated with 3% hydrogen peroxide for 15 min. After that, the sections were permeabilized with 0.1% Triton X-100 in PBS for 5 min at room temperature and then blocked with 1% sheep serum at room temperature for 1 h. For immunohistochemistry, we incubated primary antibodies that recognized OCN (Abcam, Cambridge, UK, #ab13420, 1:200), FoxO4 (Proteintech, Wuhan, China, #A21535-1-AP, 1:100), and Cbl (ABclonal, Wuhan, China, #A7881, 1:100) overnight at 4 °C. Subsequently, we used HRP-labeled secondary antibodies (1:200 in 1% bovine serum albumin [BSA], 1 h) at 37 °C. Anti-rabbit IgG (cat# 7074, dilution 1:200) from Cell Signaling Technology (MA, USA), anti-mouse IgG (cat# A9044, dilution 1:200) from Sigma Aldrich (MO, USA). DAB was used as chromogen; hematoxylin was used as counterstain. Brains were fixed in situ by intra-aortic perfusion with 350 mL of 4% paraformaldehyde in PBS, and post-fixed in the same fixative overnight at 4 °C. Vibratome coronal brain sections, 50 μm in thickness, were stored at −80 °C in the Optimal Cutting Temperature Compound. The free-floating section procedure was used.

For immunofluorescence, we incubated primary antibodies that recognized CD63 (Abcam, Cambridge, UK, cat. #: ab193349, 1:50), FN1 (Proteintech, cat. #: 15613-1-AP, 1:100), GFAP (Merck, Darmstadt, Germany, cat #: MAB360, 1:100), A2M (ABclonal, cat. #: A1573, 1:100), and MAP2 (Cell Signaling Technology, Danvers, MA, USA, cat. #: 4542 S, 1:100). For secondary reactions, species-matched Alexa Fluor 488- and Alexa Fluor 594-secondary antibodies were used (1:500 in 1% BSA, 1 h) at 37 °C in the dark. Secondary antibodies for immunofluorescence staining, Goat anti-rabbit Alexa Fluor 488 (cat#A-11008, dilution 1:500) from Invitrogen (MA, USA), Goat anti-mouse Alexa Fluor 594 (cat#A-11032, dilution 1:500) from Invitrogen (MA, USA). The sections were mounted with DAPI (Thermo) before imaging and we examined more than five different microscopic images under a confocal laser scanning microscope.

**ICV injection**. For ICV injection of sEVs, rats were anesthetized with 2.5% isoflurane and placed in a stereotaxic apparatus. A portion of the parietal skull was carefully removed, and a guide cannula was then inserted into the right lateral ventricle (coordinates from the bregma, −1.5 mm; medial/lateral, ±1.0 mm; dorsal/ventral, −3.2 mm) and fixed with dental cement. To verify the injected location, the guide cannula was connected to a polyethylene (PE) pipe that was filled with artificial cerebrospinal fluid. Then, the PE pipe was vertically inserted into the right lateral ventricle through the guide cannula. When the liquid level in the PE pipe was dropped down, it implied that the PE pipe was successfully inserted into the lateral ventricle. Then, after the rat recovered from surgery, 10 μL of either PKH-67-labeled sEVs (20 μg/μL) or PBS (0.01 M) was micro-infused into the lateral ventricle using a 33-gauge infusion cannula connected to a microsyringe (Gaoge, Shanghai). The sEVs were injected with the use of an electric microinjection pump at a flow rate of 0.5 μL/min. The cannula remained in situ for at least 5 min after infusion and was then slowly withdrawn.

**Fluorescence imaging analysis of organ distribution of sEVs**. To facilitate tracking in vivo, we injected PKH67-labeled sEVs into SD rats. A total of 200 μg (ICV) and 800 μg (intravenous) sEVs were injected into each rat. Thereafter, the rats were killed at 24 h (ICV) and 8 h (intravenous) after the injection and subjected to biophotonic imaging. Fluorescence imaging for PKH67-labeled sEVs in

these organs was performed using the optical imaging system, Fx Pro (Bruker BioSpin MRI GmbH, Ettlingen, Germany). For anatomical orientation, a white light/grayscale picture was performed and used with the fluorescent signal (Cy3.5: excitation = 570 nm; emission = 620 nm). For data analysis, molecular imaging software 7.1.3 was used. Regions of interest (ROI) were defined around the bone defect parts and analyzed compared to the background of injection of PBS for calculation of the signal to noise ratio.

**Western blot analysis**. We lysed cells and sEV with 2% SDS, 2 M urea, 10% glycerol, 10 mM Tris-HCl (pH 6.8), 10 mM dithiothreitol, and 1 mM phenylmethylsulfonyl fluoride. The lysates were centrifuged and the supernatants were separated by SDS-polyacrylamide gel electrophoresis and blocked with BSA. After blotting onto a nitrocellulose membrane (Bio-Rad Laboratories, UK), the membrane was incubated with TSG101 (Abcam, cat. #: ab83, 1:1,000), CD63 (Abcam, cat. #: ab193349, 1:1,000), CD9 (Abcam, cat. #: ab223052, 1:1,000), Alix (Abcam, cat. #: ab76608, 1:1,000), Runx2 (Cell Signaling, cat. #: 12556, 1:1,000), FN1 (Proteintech, cat. #: 15613-1-AP, 1:1,000), FOXO4 (Proteintech, cat. #: A21535-1-AP, 1:1,000), and CBL (ABclonal, cat. #: A7881, 1:1,000), β-actin (Cell Signaling Technology, cat. #: 4970S, 1:4000), GAPDH (Cell Signaling Technology, cat. #: 2118, 1:5000), UCHL1 (Cell Signaling Technology, cat#13179, 1:1000), MAP2 (Cell Signaling Technology, cat#: 4542 S, 1:1000). Secondary antibodies for western blot anti-rabbit IgG, (cat# 7074, dilution 1:2000) from Cell Signaling Technology (MA, USA), anti-mouse IgG, (cat# A9044, dilution 1:2000) from Sigma Aldrich (MO, USA). The membrane was then analyzed using specific antibodies and visualized by an enhanced chemiluminescence[50] kit (Amersham Biosciences, Piscataway, NJ, USA).

**The 3D cell cultures**. Preparation of MeGC was done according to a previous protocol[51]. A total of $2 \times 10^6$ MC3T3-E1 cells mL$^{-1}$ and various nanoparticles (final concentration of 100 nM for miRNA mimic) were mixed in MeGC solution (final concentration of 2% w/v). Forty microliters of the suspension were gelled under visible blue light in the presence of a photoinitiator, riboflavin (final concentration 6 μM).

**sEV releases kinetics from hydrogels**. To examine sEV release from hydrogels, PKH67-labeled sEVs were loaded in hydrogels at a concentration of 50 μg/hydrogel and incubated at 37 °C in 1 mL PBS in a 12-well plate. At each time point, PBS was collected and replenished with an equal amount of PBS. The amount of sEVs was calculated using an A-50 Micro-PLUS flow cytometer.

**Statistical analysis**. All data were analyzed for statistical significance using GraphPad Prism 6.0 software (GraphPad Software, San Diego, CA, USA). P value was determined by the Student's $t$ test for two-group or one-way ANOVA test for multiple group comparisons. $P < 0.05$ was considered to be statistically significant. All experiments were repeated at least three times. Quantitative data were expressed as mean ± standard error.

**Reporting summary**. Further information on research design is available in the Nature Research Reporting Summary linked to this article.

## Data availability

The miRNA-seq Illumina reads for all samples data generated in this study have been deposited into the Sequence Read Archive at the National Center for Biotechnology Information database under accession code PRJNA670580. The mass spectrometry proteomics data used in this study are available in the PRIDE database under accession code PXD022126. All other relevant data are available from the authors upon reasonable request. We use WPS 11.1 software to collect data. Source data are provided with this paper.

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

## Acknowledgements

We thank Central Laboratory, Southern Medical University for providing facilities and technical support. This work was supported by grants from the National Natural Science Foundation of China (Grant No. 81991510, 81991511, 81625015), Key Research & Development Program of Guangzhou Regenerative Medicine and Health Guangdong Laboratory (2018GZR110104002), Natural Science Foundation of Guangdong Province, China (2018A030310434, 2017A030313696), China Postdoctoral Science Foundation funded project (2016M602487), Medical Research Foundation of Guangdong Province, China (A2017117).

## Author contributions

Q.-C.S. and X.-C.B. conceptualized the research. W.X., Q.-C.S. and X.-C.B. designed the experiments; W.X. conducted the experiments. Z.-Q.C., J.X. and J.W. assisted in rat experiments. Z.-K.C assisted in 3D gel experiments. W.X., R.Z., X.-M.Z. and Z.-P.Z. performed cell experiments; J.-H.C. and X.-R.M. provided clinical samples. Z.-T.G., Z.-M.Z. and Mac Maegele contributed to developing the fluid percussion injury model. Funding Acquisition, W.X., Q.-C.S., X.-C.B.; Q.-C.S. and X.-C.B. supervised the study, interpreted the data and wrote the manuscript. Q.-C.S. and X.-C.B. conceptualized the research. W.X., Q.-C.S. and X.-C.B. designed the experiments; W.X. conducted the experiments. J.X, Z.-Q.C., X.-H.L. and J.W. assisted in rat experiments. Z.-K.C. and Y.Z. assisted in 3D gel experiments. W.X., R.Z., X.-M.Z. and Z.-P.Z. performed cell experiments; J.-H.C. and X.-R.M. provided clinical samples. Z.-T.G., Z.-M.Z., M.Z. and Mac Maegele contributed to developing the fluid percussion injury model. Funding Acquisition, W.X., Q.-C.S., X.-C.B.; Q.-C.S. and X.-C.B. supervised the study, interpreted the data and wrote the manuscript.

## Competing interests

The authors declare no competing interests.
