## [Peer Review File · Nature Communications]

Reviewers' Comments:

Reviewer #1:

Remarks to the Author:

Wei and Colleagues propose here to investigate the mechanism by which traumatic brain injury (TBI) accelerates bone healing. The authors describe that after TBI, neurons can produce exosomes that carry osteogenic miRNA, which are vehiculated to bone via the blood circulation, and target osteoprogenitors to promote bone formation and healing. Performing genomic and proteomic analyses, the authors found two miRNAs (miR-328a-3p and miR-150-5p) are markedly increased in plasma exosomes after TBI. These exosomes modulate FOXO4 and CBL, facilitating bone healing. They next propose that fibronectin expression on exosomal surface contributes to targeting osteoprogenitors in bone by TBI induced exosomes. These findings suggest a potential new strategy of bone-targeted drug therapy to treat delayed fractures and favor bone repair. The overall findings presented in the manuscript are interesting and novel for the field. The authors propose an interesting model and their results may have potential important therapeutic applications. However, the authors provide a number of claims to support these data that are not well substantiated without revision. In conclusion, several major concerns need to be addressed to make this study more compelling and ultimately worthy of publication in Nature Communications.

Major comments:

1- Critical measurements of bone analysis for their interpretations are missing such as the number of osteoblasts per bone perimeter, bone formation rate, osteoclast surface per bone surface and mineral apposition rate.

2- The authors claim that TBI-derived plasma promotes osteoblastic differentiation. However, based on the data presented in this manuscript, we cannot exclude that TBI-derived plasma could enhance osteoblast proliferation. This is an important question that the authors should address.

3- There is a striking absence of WB quantifications in this manuscript. The authors need to include these quantifications and clearly state the number of experiments and samples used for each figure panel.

4- The authors need to perform more convincing experiments for brain histological analyses. i) In Figure 5a, higher resolution images with clear markers of the hippocampal region of the brain are needed. It is unclear the audience is looking at the hippocampus ii) In Figure 5b, 6b and 6c, the pictures are not annotated. The authors need to precise what parts of the hippocampus are analyzed, and their differences within these different hippocampal area (DG, CA1, CA2, CA3). iii) The authors analyze the expression of miR-328a-3p and miR-150-5p in neurons and astrocytes. Microglia is an important source of exosome in the brain, so miRNAs expression should also be analyzed in this specific cell population. iv) The authors provided sufficient data that justify an exclusive focus on the hippocampal region of the brain as a major source of miRNA-enriched exosomes after TBI. However, the authors should provide additional data for miR-328a-3p and miR-150-5p expression in other brain regions after TBI and compare these information to the hippocampus.

5- In Figure 5, the authors claim that exosomes and exosomal miRNAs derived from injured neurons after TBI could target bone and osteoprogenitors. Based on the experiment proposed here, relying on intravenous injections of PKH67-labelled exosomes, this is an overstated conclusion. To support this notion, we believe that the authors should perform additional experiments, such as confronting their results by repeating these analyses after ICV brain infusion of PKH67-labelled exosomes or primary neurons derived exosomes, and not after of intravenous injections.

6- The authors need to provide the quantification of the exosome accumulation in Figure 6j and i.

7- The authors choose to use HT22 cells as an in vitro model of hippocampal neurons. However, it

is not clear whether these cells were differentiated into adult neurons. It would have been preferable to use primary hippocampal neurons for more physiological relevance. There are no technical difficulties explaining this choice, it is relatively easy to produce primary hippocampal neuronal cultures with standard protocols. The authors should clearly argue on their choice of cell culture and soften their conclusions.

8- The authors only tested the therapeutic potential of miR-328a-3p. They should consider analyzing the possible effect of miR-150-5p or explain the rationale for focusing only on miR-328a-3p.

9- The authors should state more clearly how many independent experiments were performed and how many samples they used for each figure panel presented (for instance, in the figure legends) in the manuscript.

10- Statistical outcomes are given in various figures only as p values but it's not always clear what test is being used. The authors perform ANOVAs for several of their experiments, but they did not mention if they followed this up with post hoc tests that correct for multiple comparisons (such as Tukey's).

Minor comments:

- The authors used GW4869 as a specific inhibitor of exosome secretion. The authors should provide the quantification of the number of exosomes after injection of this inhibitor.
- Figure 7: Authors should provide description of each treatment group.
- Supplemental Figure 5: The authors should provide quantifications of ALP staining and indicate how many experiments were performed.
- The following sentence is not correct: « In addition, the uptake of injured HT22 cell-derived exosomes by BMSC and MC3T3-E1 cells was far more than that by undamaged HT22 cells (Fig. 6i, j). ». The HT22 cells are not injured or undamaged. This sentence should be edited for readability.
- The authors need to define the concept of "Hyperactive miRNA" in the manuscript.
- The authors should fully explain the rationale of focusing on miRNA sequencing.
- The authors need to provide the rationale for using Oxygen-glucose deprivation as a model in vitro.
- A few typographical errors are present in the manuscript. Please correct.

Reviewer #2:

Remarks to the Author:

What are the major claims of the paper?

Overall this is a high-quality research study that investigates whether exosomes collected from the circulation of patients following traumatic brain injury (TBI) contribute mechanistically to the observed clinical phenomenon of accelerated fracture repair with concomitant TBI. The paper identified miR-328 and miR-150 in the exosomes isolated from TBI patient plasma and demonstrate that they inhibit FOXO4 and CBL, the negative regulators of osteogenesis, respectively. The paper also found that exosomes secreted after TBI have more fibronectin (FN1) on their membrane surface and suggest that FN1 selectively directs exosomes to bone and osteoprogenitors.

Are the claims novel? If not, please identify the major papers that compromise novelty

Yes, to my knowledge this is the first paper to investigate the role of exosomes within circulating plasma as a mechanism for brain to bone cross-talk and identify and underlying mechanism for regulation of enhanced bone regeneration from these exosomes. Importantly, this mechanism was successfully engineered into a therapy that shows translational promise for recapitulating TBI-induced accelerated fracture repair.

Will the paper be of interest to others in the field?

Yes, for bone researchers this study identifies novel regulators for bone homeostasis after injury and sheds light on brain-to-bone crosstalk. For researchers in other fields hoping to study exosomes this study and its methods will be of interest and a valuable reference.

Will the paper influence thinking in the field?

- Yes, this study should influence further research looking at other miRNAs that regulate bone homeostasis and provide new ideas supporting brain-to-bone crosstalk.

Are the claims convincing? If not, what further evidence is needed?

- Yes

Are there other experiments that would strengthen the paper further? How much would they improve it, and how difficult are they likely to be?

To improve the quality of the paper additional editing for English should be performed.

In addition, the following listed concerns related to the experiments/figures/text should be addressed in a revised manuscript. These modifications are important to the paper but should require only minimal experimentation.

- One limitation of this study is that the clinical scenario was a femoral shaft injury with IM rod stabilization that heals predominantly by endochondral ossification, where the preclinical study was a unicortical drill hole injury in the tibia that heals by intramembranous ossification. Since the majority of clinical fractures heal through secondary bone formation this discrepancy and potential translational consideration should be addressed in the discussion.
- All data figures should be re-designed to show individual data points as it is often unclear the biological replicate number (<https://journals.plos.org/plosbiology/article?id=10.1371/journal.pbio.1002128>). This can be done easily with Prism Software (<https://www.graphpad.com/support/faq/graph-tip-how-can-i-make-a-barcolumn-graph-that-also-shows-the-individual-data-points/>) or other graphing packages.
- Statistics – Did wo-group analyses qualify for t-test? Mann-Whitney may be more appropriate given small sample size. Please justify or correct statistical testing. Please provide exact p- or F-values for statistical tests.
- Sup Fig/Table 1 – Radiographic images show simple (not “severe” as stated in the methods) femoral shaft FX, please correct terminology or detail in supplemental patient data. Can the supplemental patient data table 1 specify additional information regarding the relationship between the location of TBI and FX (ipsilateral and contralateral), the severity of TBI, and FX (AO classification)? This additional information would be beneficial since the phenomenon of TBI promoting FX healing is not always developed.

Fig 2 -

- The authors write, “We found that plasma from patients or rats with TBI but not controls promoted osteoblastic differentiation of primary cultured rat bone marrow stem cell (BMSC) and osteoprogenitor cell line” in result section, but there is no result of whole patient plasma treatment (Figure 2a-d only show rat plasma and exosomes, and Suppl. 5 shows only patient exosomes).
- Please expand methods with additional details related to the collection of the rat plasma/exosomes to include (a) collection date post-TBI, (b) were plasma/exosomes collected from a single rat or multiple pooled rats, (c) what was duration of plasma/exosome treatment on the cells.
- How does 20uL plasma correlate to 10ug exosomes? Could a BCA be performed on non-exosomal component in figure 2? How was the concentration of exosomes (10ug) chosen?
- Was efficacy of the GW4869 drug treatment and dosing confirmed by measuring exosomes in the

blood of groups in 2f? Please show this data or provide a reference to confirm reduction/efficacy of the drug and there isn't some other cytotoxic effect?

- Why do PTD + TBI OCN levels not correlate when comparing figures 1g and 2k?
- Figure 3g please include additional methods and details related to collection of TBI patient plasma exosomes (collecting location - peripheral blood?, collection date post-TBI; how many patients concentrations and what was their severity & type of TBI, age/sex match with healthy subject?).
- Figure 4f – why would FOXO4 and CBL proteins be upregulated in PTD+TBI bone tissue (histology) if they are negative regulators of osteogenesis, more bone is forming and the exosomes from TBI are supposed to be downregulating these pathways – this data seems contradictory, please clarify.

Fig. 5

- Please correct the orientation of Fig. 5a brain images according to the standard: <http://atlas.brain-map.org/atlas?atlas=1&plate=100960260#atlas=1&plate=100960088&resolution=10.47&x=5247.9998779296875&y=4032.0000648498535&zoom=-3>
- Fig. 5d - exosomal miRNA levels in damaged neuronal cells using OGD assay. Oxygen deprivation may be a better in vitro model of ischemic brain injury model (stroke) than TBI – can the authors please justify. and there is no in vivo expression data of miRNAs in the hippocampus after TBI?
- Fig 5e – it is not totally clear if there are increased exosomes following liver injury. Is there any semi-quantification or replicate information that can be included?
- Fig5f – The authors should provide some quantification of PKH67-labeled exosome uptake, it is difficult from a 3 cell image to know how ubiquitous this is. Why did MLO-Y4 cells not stain with Phalloidin-TRITC
- Figure 5g – Please provide a low magnitude image to show where in the fracture these images are taken. Please resolve the apparent increased miR-328a-3p (Fig 5g) and the increased FOXO4 IHC (4f) they seem to conflict.

Fig. 7

- Details regarding the hydrogel are very slim. What is the concentration of exosomes loaded into the hydrogel? How was the hydrogel placed in the fracture defect and what are the hydrogel degradation kinetics? What are the release kinetics of the exosomes from the hydrogel? (IMPORTANT!)
- Fig 7e-f – Better statistical data should be given for 7f (ANOVA results followed by Tukey HSD). It appears (7e) that Gel+Exo+NC increases bone formation by microCT, this does not come through in the stats for 7f, but these stats do not appear complete.
- Can IHC or gene expression analysis be performed to confirm the delivery of the Gel+Exo+MiR-328 lead to effective decrease in FOXO4. This would help confirm results of Sup Fig 7A-B. Better statistical description (ANOVA results followed by Tukey HSD) should be added to S7b.
- Supplemental Figure 2 - Please provide an gross (whole brain) histological image of the brain to compliment the high magnification image in S2b.
- Supplemental Figure 5 – This figure would be greatly improved by including de-identified information on TBI patients (collecting location; peripheral blood?, collection date post-TBI, severity & type of TBI, age/sex match with healthy subject?)

Are the claims appropriately discussed in the context of previous literature?

The discussion is well written.

Reviewer #3:

Remarks to the Author:

In this study, the Authors are investigating the underlying mechanisms for bone healing in patients and animals with concomitant traumatic brain injury (TBI). Their study focuses on deciphering the crosstalk between bone and brain in the pre-clinical and clinical settings of TBI, where they mainly attributed the bone healing effects to the exosomes released by injured neurons containing osteogenic miRNAs.

Major comment #1: The presence of fibronectin on exosomes identified as mechanism of action for targeting osteoprogenitors.

This comment refers to the statement made by the Authors on Page 12 "Moreover, we found that increased FN1 expression on exosomal surface contributed to the direction of TBI exosomes to bone". FN1 is an extracellular matrix protein known to have multiple binding partners, including integrins (also described by the Authors in Figure 8), which are molecules found on a variety of cell types, including the osteoprogenitors. Therefore, the high abundance of FN1 found on neuronal-derived exosomes may lead to an increase in binding capacity of the exosomes to their target cells. This assumption should be further discussed by the Authors when discussing their findings as they have not provided a detailed explanation as to how this would work (i.e. FN1 contributing to the direction of TBI exosomes). Overall, this statement is not well-supported by their results and may be overstated.

Major comment #2: Confirming the cell-of-origin releasing the exosomes. Further evidence regarding the cell-of-origin that is releasing the exosomes into the plasma of patients and rat model of TBI should be provided to support the claim that exosomes from injured neurons are being released into the plasma. EVs share surface molecules with the releasing cell, so neuronal markers should be validated for their presence on exosomes. A2M molecule was mentioned by the Authors on Page 10; however, additional supportive markers for neuronal origin should be added. Furthermore, mass spectrometry (MS) allows for the identification of candidate proteins and validation by orthogonal methods (ELISA, WB, FC, etc) is often used to validate the protein of interest identified (WB, FC for examples). A2M was identified as a candidate protein by MS, but not further validated.

Major comment #3: Confirming the cross-talk between the injured neurons and osteoprogenitor cells through exosomes. Have the Authors previously published in vitro data supporting the direct transfer of EVs from neurons to osteoprogenitor using an in vitro model such as Transwell assay? This assay allows to seed 2 cell types in separate chambers where exosomes transfer and functional effects (e.g. differentiation of osteoprogenitors) can be assessed. This would further strengthen their hypothesis of neuronal EV transfer to osteoprogenitor cells.

Major comment #4: This one relates to: Page 5 of 47 in section entitled "Plasma Exosomes from TBI Patients and Rats Contribute to TBI-Stimulated Osteogenesis". The Authors stated: "We found that plasma from patients or rats with TBI but not controls promoted osteoblastic differentiation of primary cultured rat bone marrow stem cell (BMSC) and osteoprogenitor cell line, MC3T3-E1, as manifested by an enhanced alkaline phosphatase staining and Runx2 expression in TBI plasma treated cells (Fig. 2a, b and Supplementary Fig. 5)."

- Healthy controls are mentioned in this section, but without any prior description in the Methods. The Authors should revise this section.

- Differentiation results shown in Figures 2a and 2b as well as in Suppl. Figure 5 for controls clearly showed osteoblastic differentiation from the addition of control plasma. By comparing the differentiation results from control to TBI plasma, it is true that the differentiation potential is enhanced, but it is incorrect to say that controls did not promote osteoblastic differentiation as the Authors stated.

- Results from qualitative assessment of Western Blot and chemical staining assays should also be quantified and analyzed for statistical significance.

Major comment #5: This one relates to the BMSC sample mentioned on Page 5 in section entitled "Plasma Exosomes from TBI Patients and Rats Contribute to TBI-Stimulated Osteogenesis".

The BMSC sample described on Page 5 in this section seems to refer to only 1 rat (i.e. 1 biological replicate) as mentioned in the Methods on Page 18 - "The BMSC were harvested from the femora and tibiae of SD rats. The cells from one rat were seeded onto [...] for 48 h". The authors should

clearly explain why they have only used 1 biological sample for all the testing of the BMSC response. Does this also apply to every experiment throughout the manuscript conducted using the BMSC such as the one mentioned on Page 7 where it is stated: " To determine the effects of these miRNAs on osteoblast formation, MC3T3-E1 cells and primary rat BMSCs were transfected with miRNA mimics or negative controls"?

Major comment #6: This one relates to the CD63-positive "particles" increased after TBI. On Page 9, the Authors state: "Importantly, CD63-positive particles increased dramatically after TBI, indicating that the production of exosomes was active in the damaged brain (Fig. 5b, c)". The Authors have not presented clear evidence that the CD63-positive signal refers to particles instead of cells (or both). CD63 is not exclusive to extracellular vesicles (EVs or exosomes). Cells are known to express CD63, including endothelial cells for example. Furthermore, the Authors need to provide further evidence to support this claim: "Interestingly, most of the CD63 was expressed in microtubule-associated protein 2 (MAP2)-positive, but not glial fibrillary acidic protein (GFAP)-positive cells, indicating that damaged neuronal cells, but not astrocytes secreted exosomes vigorously." No data are provided for the identification of the "damaged neuronal cells", as well as the secretion of exosomes by these cells. The Authors should clarify this.

Major comment #7: This one refers to the distribution of PKH67-exosomes evaluated by biophotonic imaging at 6 h after injections. The Authors should provide quantifiable data for the signal of all biophotonic imaging experiments. Statistical analyses should also be performed.

Major comment #8: This one refers to the Tandem Mass Tag (TMT) labeling and proteomics analyses.

- Details regarding the mass spectrometer (MS) they have used for analysis as well as details regarding the MS method of data analysis are missing. The Authors should provide this information.

- The proteomics experiment and the MS results have not yielded many protein candidates. This could be due to inadequate EV lysis efficiency as the few proteins identified by MS are of extracellular origins (e.g. extracellular matrix proteins, IgG, etc) or bound to membranes (e.g. Apolipoproteins). From the short list of proteins identified by MS (Suppl. Table) only a very few cargo proteins (intra) were identified. A low percentage of detergent (e.g. 0.1% SDS) would have yielded a better EV lysing process as well as enhanced protein solubility; hence, a higher yield of proteins would have been identified. The Authors should comment on the choice of their proteomics method and clarify their results.

- TMT labeling allows for the multiplexing of up to 16 samples (i.e. 16 different isotopic labels can be multiplexed without compromising on protein identification and quantitation). The Authors have used only 1 biological sample (i.e. 1 rat) in each group to conduct TMT proteomics. If the authors have pooled multiple biological samples for each group (i.e. tag 129 and tag 131), they should describe their approach. If they Authors have only used 1 biological sample per group, as currently stated, they should clarify why they have not used additional biological samples. Furthermore, at the validation stage of the MS data, the Authors should have included more biological samples for result validation as MS results require this kind of verification for the validity of the results.

Minor comment #1: The Supplementary Table 2 refers to "Clinical features of patients with fractures and concomitant TBI". On Page 7, the Suppl. Table 2 refers to the miRNA profile. Please correct.

Minor comment #2: I think that the authors meant to write "that levels of FOXO4 and CBL were decreased" and not "increased". To be found here: "We found that levels of FOXO4 and CBL were increased at the bone defect site of TBI-PTD rats as compared with that of PTD rats, which suggests that FOXO4 and CBL may be involved in TBI-stimulated bone formation (Fig. 4f, g)".

Minor comment #3: The article would benefit from having a Table where the samples used in the animal study would be described. For example, each sample would be identified with a unique identifier and a description of which one was used for which purpose should be included. This would bring clarity as to how many animals and which one was used for the different analyses/methods. Overall, the number of samples and which ones used throughout the

manuscript is not clear.

Minor comment #4: Referring to "Exosomes isolated from plasma and HT22 cells supernatant 24 h after OGD treatment were labeled with PKH67, 10 $\mu\text{g}/\text{mL}$ of monoclonal antibodies FN1-PE, and analyzed by nanoscale Apogee flow cytometer (FC)" – What is the total quantity of antibody used for the detection of FN1? Did the Authors do a titration experiment for confirming the use of 10 $\mu\text{g}/\text{mL}$?

Minor comment # 5: Figures 4h, i, j and k. Quantitation and statistical significance are required. Please add.

Minor comment #6: Figure 5: Most of the quantification data and statistical computations are missing. Please add.

Minor comment # 7: Figure 6: Most of the quantification data and statistical computations are missing (b, c, f, l). Please add.

We thank the reviewers for their time and thoughtful comments. We have extensively addressed and clarified the concerns of the reviewers, and we strongly believe that the revised manuscript has been improved as a result of the reviewers' comments. Below are point-by-point responses.

Reviewer #1 (Remarks to the Author):

Wei and Colleagues propose here to investigate the mechanism by which traumatic brain injury (TBI) accelerates bone healing. The authors describe that after TBI, neurons can produce exosomes that carry osteogenic miRNA, which are vehiculated to bone via the blood circulation, and target osteoprogenitors to promote bone formation and healing. Performing genomic and proteomic analyses, the authors found two miRNAs (miR-328a-3p and miR-150-5p) are markedly increased in plasma exosomes after TBI. These exosomes modulate FOXO4 and CBL, facilitating bone healing. They next propose that fibronectin expression on exosomal surface contributes to targeting osteoprogenitors in bone by TBI induced exosomes. These findings suggest a potential new strategy of bone-targeted drug therapy to treat delayed fractures and favor bone repair.

The overall findings presented in the manuscript are interesting and novel for the field. The authors propose an interesting model and their results may have potential important therapeutic applications. However, the authors provide a number of claims to support these data that are not well substantiated without revision. In conclusion, several major concerns need to be addressed to make this study more compelling and ultimately worthy of publication in Nature Communications.

We thank the reviewer for the careful reading and the positive comments that “the overall findings presented in the manuscript are interesting and novel for the field.” We also thank the reviewer for the constructive suggestions that have significantly improved the manuscript.

Major comments:

1- Critical measurements of bone analysis for their interpretations are missing such as the number of osteoblasts per bone perimeter, bone formation rate, osteoclast surface per bone surface and mineral apposition rate.

Thank you for the comment. We provide additional experiments to address this point. The numbers of osteoblasts and osteoclasts per bone perimeter were assessed. The number of osteoblasts per bone perimeter was increased in TBI rats (**new Fig. S3a**), while the number of osteoclasts per bone surface area was not changed (**new Fig. S3g**). The number of osteoblasts per bone perimeter was increased in Gel+Exo+miR-328a-3p group (**new Fig. S15a**), while the number of osteoclasts per bone surface area was not changed (**new Fig. S15h**). Accordingly, in vivo bone regeneration rate was also increased significantly after TBI (**new Fig. S3b-d**).

2- The authors claim that TBI-derived plasma promotes osteoblastic differentiation. However, based on the data presented in this manuscript, we cannot exclude that TBI-derived plasma could enhance osteoblast proliferation. This is an important question that the authors should address.

Thank you for the comment. We have provided additional experiments to address this point. Based on the request of the reviewer, CCK-8 assays were performed to evaluate the effect of TBI-derived plasma on osteoblast proliferation (**new Fig. S5b**). The results showed no significant difference of MC3T3-E1 cell proliferation between treatments with sham plasma or TBI-derived plasma.

3- There is a striking absence of WB quantifications in this manuscript. The authors need to include these quantifications and clearly state the number of experiments and samples used for each figure panel.

Thank you for this excellent suggestion. All the western blotting results were quantitated, and the number of experiments and samples used were added in each figure panel. We also redesigned the figures to show individual

data points and the biological replicate number.

4- The authors need to perform more convincing experiments for brain histological analyses. I) In Figure 5a, higher resolution images with clear markers of the hippocampal region of the brain are needed. It is unclear the audience is looking at the hippocampus ii) In Figure 5b, 6b and 6c, the pictures are not annotated. The authors need to precise what parts of the hippocampus are analyzed, and their differences within these different hippocampal area (DG, CA1, CA2, CA3). iii) The authors analyze the expression of miR-328a-3p and miR-150-5p in neurons and astrocytes. Microglia is an important source of exosome in the brain, so miRNAs expression should also be analyzed in this specific cell population. iv) The authors provided sufficient data that justify an exclusive focus on the hippocampal region of the brain as a major source of miRNA-enriched exosomes after TBI. However, the authors should provide additional data for miR-328a-3p and miR-150-5p expression in other brain regions after TBI and compare these information to the hippocampus.

We thank the reviewer for the careful reading and constructive suggestions. We provide both clarification and additional experiments to address these points.

i) Higher resolution images have been added (**new Fig. 5a**) so that the hippocampus can be seen clearly. We found that the miR-328a-3p expression level was higher of in the CA1 and DG regions of TBI rats than that of sham rats.

ii) In **new Figs. 5b, 5c, and 6d, and 6e**, the pictures were replaced with higher resolution images, and the CA1 regions were annotated with quantifications of CD63, FN1, and A2M mean signal intensities. Furthermore, immunocytofluorescence staining of MAP and GFAP of different hippocampal regions (CA2, CA3 and DG) have been provided (**Fig. S9, S10**).

iii) In additional experiments, real-time quantitative PCR was used to detect the profiles of miR-328a-3p and miR-150-5p in microglia cell BV2 exosomes 24 h after OGD treatment (**new Fig. S11a**). And the miR-328a-3p content did not change significantly while the content of miR-150-5p was increased in exosomes of murine microglial cell line BV2 after OGD treatment.

iv) Microscopic images with a larger field of view have been added (**new Fig. S8**) so that miR-328a-3p expression in other brain regions can be seen clearly. We found that miR-328a-3p expression was increased significantly and was higher in the hippocampus of TBI rats than that of other brain regions.

5- In Figure 5, the authors claim that exosomes and exosomal miRNAs derived from injured neurons after TBI could target bone and osteoprogenitors. Based on the experiment proposed here, relying on intravenous injections of PKH67-labelled exosomes, this is an overstated conclusion. To support this notion, we believe that the authors should perform additional experiments, such as confronting their results by repeating these analyses after ICV brain infusion of PKH67-labelled exosomes or primary neurons derived exosomes, and not after of intravenous injections. Thank you for your constructive suggestions. We have provided additional data to address this point. The organ distribution of the fluorescence signal in rats was detected 24 h after ICV brain infusion with PKH67-labeled exosomes (**new Fig. 5e**). The results showed that PKH67-labeled exosomes accumulated to higher levels in rat limb bones in the TBI-exo group, when compared with the other groups.

6- The authors need to provide the quantification of the exosome accumulation in Figure 6j and i.

Thank you for your suggestion. We have addressed this point using quantitation of exosome accumulation in the added **Fig. 6j and 6i**.

7- The authors choose to use HT22 cells as an in vitro model of hippocampal neurons. However, it is not clear whether these cells were differentiated into adult neurons. It would have been preferable to use primary

hippocampal neurons for more physiological relevance. There are no technical difficulties explaining this choice, it is relatively easy to produce primary hippocampal neuronal cultures with standard protocols. The authors should clearly argue on their choice of cell culture and soften their conclusions.

Thank you for your comments. To address this point, we have provided the results of additional experiments with primary hippocampal neuron cells (**new Fig. 6g-j**). The results were consistent with that of HT22 cells (**new Fig. S14c-f**). The western blotting and nanoscale flow cytometry results showed that FNI in exosomes isolated from primary culture of rat hippocampal neurons was increased after OGD treatment (**new Fig. 6g, 6h**). The immunocytofluorescence staining results showed that FNI in primary cultures of rat hippocampal neurons was increased after OGD treatment (**new Fig. 6i**). In addition, the uptake of exosomes isolated from primary hippocampal neurons with OGD treatment by BMSC and MC3T3-E1 cells was greater than that of the control (**new Fig. 6j**).

8- The authors only tested the therapeutic potential of miR-328a-3p. They should consider analyzing the possible effect of miR-150-5p or explain the rationale for focusing only on miR-328a-3p.

Thank you for your comment. Mir-328a-3p in plasma exosomes of TBI rat and patient was much more than that of miR-150-5p (**Fig 3b and 3g**). And mir-328a-3p in rat hippocampal neuron exosomes with OGD treatment increased much more than that of miR-150-5p (**Fig 5d**). In addition, mir-328a-3p was more effective than miR-150-5p on promoting osteogenesis (**Fig 3c-f**). So we chose miR-328a-3p for therapeutic potential analysis.

9- The authors should state more clearly how many independent experiments were performed and how many samples they used for each figure panel presented (for instance, in the figure legends) in the manuscript.

Thank you for your comment. We have again analyzed all data to show individual data points and the biological replicate numbers clearly in each figure panel in the revised manuscript.

10- Statistical outcomes are given in various figures only as p values but it's not always clear what test is being used. The authors perform ANOVAs for several of their experiments, but they did not mention if they followed this up with post hoc tests that correct for multiple comparisons (such as Tukey's).

We appreciate your suggestion. We have added statistical descriptions (one-way analysis of variance with Turkey's multiple comparisons tests were performed) and P-values (*P < 0.05, **P < 0.01, ***P < 0.001, and ****P < 0.0001) in the figure legends.

Minor comments:

- The authors used GW4869 as a specific inhibitor of exosome secretion. The authors should provide the quantification of the number of exosomes after injection of this inhibitor.

Thank you for the comment. We have provided additional experiments to address this point. The concentration of exosomes isolated from the plasma of rats was detected by NanoSight Analysis and western blotting (**new Fig. S7a, S7b**). The results showed that the purified exosomes from rat plasma decreased after GW4869 treatment.

- Figure 7: Authors should provide description of each treatment group.

Thank you for the comment. A description of each treatment group has been provided in the Fig. 7 legend.

- Supplemental Figure 5: The authors should provide quantifications of ALP staining and indicate how many experiments were performed.

Thank you for this excellent suggestion. We have provided quantitations of ALP staining in **Fig. S5a** and indicated

the number of experiments in the figure legend. It reads, “n = 3, Student’s unpaired t test. The quantification result were plotted as dot plots, showing the mean ± SE of three independent experiment s.”

- The following sentence is not correct: « In addition, the uptake of injured HT22 cell-derived exosomes by BMSC and MC3T3-E1 cells was far more than that by undamaged HT22 cells (Fig. 6i, j). ». The HT22 cells are not injured or undamaged. This sentence should be edited for readability.

Thank you for the comment. We have revised the sentence in the text. It reads “In addition, the uptake exosomes isolated from primary hippocampal neurons and HT22 cells with OGD treatment by BMSC and MC3T3-E1 cells were far more than that of controls.”

- The authors need to define the concept of “Hyperactive miRNA” in the manuscript.

We apologize for the confused expression. We have revised the sentence in the text on **Page 7**. It reads “We found that miR-328a-3p and miR-150-5p significantly promoted osteoblast differentiation.”

- The authors should fully explain the rationale of focusing on miRNA sequencing.

Thank you for the comment. We have fully explained the rationale of focusing on miRNA sequencing in the Results on **Page 7**. It reads “The miRNAs, which can be abundantly encapsulated in exosomes, are key regulators of bone remodeling in health and disease^{1,2}, we therefore evaluated the miRNA expression profiles in sham-Exo and TBI-Exo groups using miRNA sequencing.”

1. Xie, Y., Chen, Y.Y., Zhang, L.C., Ge, W. & Tang, P.F. The roles of bone-derived exosomes and exosomal microRNAs in regulating bone remodelling. *Journal of cellular and molecular medicine* 21, 1033-1041 (2017).

2. Gao, M.H., et al. Exosomes-the enigmatic regulators of bone homeostasis. *Bone Res* 6(2018).

- The authors need to provide the rationale for using Oxygen-glucose deprivation as a model in vitro.

Thank you for the comment. We have provided the rationale for using oxygen-glucose deprivation as a model *in vitro* the Results on **page 9**. It reads, “Primary traumatic injury generally initiates a cascade of secondary injury mechanisms such as edema, hemorrhage, and decreased cerebral blood flow resulting in ischemic injury to brain tissue¹. While ischemic injury is a secondary consequence of TBI, ischemia is the primary mechanism of injury in cases that disrupt arterial cerebral blood flow leading to a lack of oxygen and glucose in brain tissue and oxygen-glucose deprivation (OGD) is an in vitro model commonly used for TBI^{2,3}.”

1. Bramlett, H.M. & Dietrich, W.D. Pathophysiology of cerebral ischemia and brain trauma: Similarities and differences. *J Cerebr Blood F Met* 24, 133-150 (2004).

2. Salvador, E., Burek, M. & Forster, C.Y. An In Vitro Model of Traumatic Brain Injury. *Methods in molecular biology* (Clifton, N.J.) 1717, 219-227 (2018).

3. Wang, G.H., et al. Microglia/macrophage polarization dynamics in white matter after traumatic brain injury. *J Cerebr Blood F Met* 33, 1864-1874 (2013).

- A few typographical errors are present in the manuscript. Please correct.

We apologize for the typographical errors. We have corrected them in red in the revised manuscript.

Reviewer 2

What are the major claims of the paper?

Overall this is a high-quality research study that investigates whether exosomes collected from the circulation of patients following traumatic brain injury (TBI) contribute mechanistically to the observed clinical phenomenon of

accelerated fracture repair with concomitant TBI. The paper identified miR-328 and miR-150 in the exosomes isolated from TBI patient plasma and demonstrate that they inhibit FOXO4 and CBL, the negative regulators of osteogenesis, respectively. The paper also found that exosomes secreted after TBI have more fibronectin (FN1) on their membrane surface and suggest that FN1 selectively directs exosomes to bone and osteoprogenitors.

Are the claims novel? If not, please identify the major papers that compromise novelty

Yes, to my knowledge this is the first paper to investigate the role of exosomes within circulating plasma as a mechanism for brain to bone cross-talk and identify and underlying mechanism for regulation of enhanced bone regeneration from these exosomes. Importantly, this mechanism was successfully engineered into a therapy that shows translational promise for recapitulating TBI-induced accelerated fracture repair.

Will the paper be of interest to others in the field?

Yes, for bone researchers this study identifies novel regulators for bone homeostasis after injury and sheds light on brain-to-bone crosstalk. For researchers in other fields hoping to study exosomes this study and its methods will be of interest and a valuable reference.

Will the paper influence thinking in the field?

- Yes, this study should influence further research looking at other miRNAs that regulate bone homeostasis and provide new ideas supporting brain-to-bone crosstalk.

Are the claims convincing? If not, what further evidence is needed?

- Yes

Are there other experiments that would strengthen the paper further? How much would they improve it, and how difficult are they likely to be?

To improve the quality of the paper additional editing for English should be performed.

We thank the reviewer for the careful reading and the positive comments, such as “Overall this is a high-quality research study that investigates whether exosomes collected from the circulation of patients following traumatic brain injury (TBI) contribute mechanistically to the observed clinical phenomenon of accelerated fracture repair with concomitant TBI.” The editing for English has been completed in the revised manuscript by International Science Editing (ISE). We strongly believe that the revised manuscript has improved as a result of the reviewer’s comments.

In addition, the following listed concerns related to the experiments/figures/text should be addressed in a revised manuscript. These modifications are important to the paper but should require only minimal experimentation.

We thank the reviewer for the constructive suggestions that have significantly improved the manuscript.

- One limitation of this study is that the clinical scenario was a femoral shaft injury with IM rod stabilization that heals predominantly by endochondral ossification, where the preclinical study was a unicortical drill hole injury in the tibia that heals by intramembranous ossification. Since the majority of clinical fractures heal through secondary bone formation this discrepancy and potential translational consideration should be addressed in the discussion.

Thank you for your comments. We have addressed the limitation of monocortical defect model and the potential translational consideration in the discussion. It reads “The process of fracture healing is a complex process which requires many kinds of cells and factors. We studied bone repair with a monocortical defect model which have been

used in many previous research¹⁻⁶. Although this repair model recapitulates most steps of fracture healing, however, the majority of clinical fractures heal through secondary bone formation (endochondral ossification). Further studies using stabilized fracture model should be performed to verify its potential application”

1. Kim, J.B., et al. Bone regeneration is regulated by Wnt signaling. *J Bone Miner Res* 22, 1913-1923 (2007).
2. Hu, K. & Olsen, B.R. Osteoblast-derived VEGF regulates osteoblast differentiation and bone formation during bone repair. *J Clin Invest* 126, 509-526 (2016).
3. Schall, N., et al. Protein kinase G1 regulates bone regeneration and rescues diabetic fracture healing. *Jci Insight* 5(2020).
4. Zhao, S.-J., et al. MacrophageGIT1Contributes to Bone Regeneration by Regulating Inflammatory Responses in anERK/NRF2-Dependent Way. *J Bone Miner Res* (2020).
5. Zhao, S.-J., et al. Macrophage MSR1 promotes BMSC osteogenic differentiation and M2-like polarization by activating PI3K/AKT/GSK3 beta/beta-catenin pathway. *Theranostics* 10, 17-35 (2020).
6. Ono, T., et al. IL-17-producing gamma delta T cells enhance bone regeneration. *Nat Commun* 7(2016).

• All data figures should be re-designed to show individual data points as it is often unclear the biological replicate number (<https://journals.plos.org/plosbiology/article?id=10.1371/journal.pbio.1002128>).

This can be done easily with Prism Software

(<https://www.graphpad.com/support/faq/graph-tip-how-can-i-make-a-barcolumn-graph-that-also-shows-the-individual-data-points/>) or other graphing packages.

Thank you for your comments. We have redesigned all the figures to show individual data points and the biological replicate number, and have stated the number of experiments and samples in each figure legend in the revised manuscript.

• Statistics – Did wo-group analyses qualify for t-test? Mann-Whitney may be more appropriate given small sample size. Please justify or correct statistical testing. Please provide exact p- or F- values for statistical tests.

We appreciate your excellent suggestion. We have added better statistical descriptions and corrected statistical testing (one-way analysis of variance with Turkey’s multiple comparison tests or Student’s unpaired *t*-tests were performed) and we provided P-values (*P < 0.05, **p < 0.01, ***P < 0.001, ****P < 0.0001) with each statistical test.

• Sup Fig/Table 1 – Radiographic images show simple (not “severe” as stated in the methods) femoral shaft FX, please correct terminology or detail in supplemental patient data.

Can the supplemental patient data table 1 specify additional information regarding the relationship between the location of TBI and FX (ipsilateral and contralateral), the severity of TBI, and FX (AO classification)? This additional information would be beneficial since the phenomenon of TBI promoting FX healing is not always developed.

Thank you for the suggestions. We apologize for the confused expression, which has been revised with “All patients sustained an isolated shaft fracture of the femur.” We have corrected the terminology and provided additional information regarding the relationship between the location of TBI and FX, the severity of TBI (GCS score), and FX (AO/OTA fracture classification) in the supplemental patient data Table 1.

Fig 2 -

• The authors write, ‘We found that plasma from patients or rats with TBI but not controls promoted osteoblastic differentiation of primary cultured rat bone marrow stem cell (BMSC) and osteoprogenitor cell line’ in result section, but there is no result of whole patient plasma treatment (Figure 2a-d only show rat plasma and exosomes,

and Suppl. 5 shows only patient exosomes).

We apologize for the confusion in wording, which has been revised in **Supplementary Figure 5**. Plasma from TBI patients contributed to TBI-stimulated osteogenesis, which was related to the data shown in **Figure 2**. **Figure S5a** shows that TBI patient plasma treatment contributed to TBI-stimulated osteogenesis.

- Please expand methods with additional details related to the collection of the rat plasma/exosomes to include (a) collection date post-TBI, (b) were plasma/exosomes collected from a single rat or multiple pooled rats, (c) what was duration of plasma/exosome treatment on the cells.

Thank you for the suggestions. We have expanded the methods with additional details related to the collection of rat plasma/exosomes in **Supplemental Methods**. It reads, “Plasma samples. 24 h post TBI surgery, all rats were anaesthetized and peripheral blood was collected in a tube containing EDTA as an anticoagulant. The samples were stored at 4°C for a short time and immediately centrifuged at 4°C and 3000 rpm for 10 min. The supernatant was carefully collected and transferred to a new tube without disturbing the intermediate buffy coat layer. In total, 20 samples (10 from the TBI group and 10 from the sham group) were used to isolate plasma.”

“BMSCs and MC3T3-E1 cells (2×10^5 /well) were treated with the exosomes (10 μ g) or non-exosomal component (20 μ L) with conditioned media during differentiation.”

- How does 20 μ L plasma correlate to 10 μ g exosomes? Could a BCA be performed on non-exosomal component in figure 2? How was the concentration of exosomes (10 μ g) chosen?

Thank you for the comment. We apologize for the confusion. We performed CCK-8 assay to test the cytotoxicity of 20 μ L plasma and 10 μ g exosomes treated on BMSC and MC3T3-E 1. No significant cytotoxicity was observed for the concentration of exosomes (up to 10 μ g) and plasma (up to 20 μ L) in the 24-well plates. 20 μ L of plasma was not directly related to 10 μ g exosomes. For comparison, exosomes (2 μ L) and the non-exosomal component (20 μ L) isolated from equal amounts of plasma were used.

- Was efficacy of the GW4869 drug treatment and dosing confirmed by measuring exosomes in the blood of groups in 2f? Please show this data or provide a reference to confirm reduction/efficacy of the drug and there isn't some other cytotoxic effect?

Thank you for the comment. We provided additional experiments to address this point. The exosomes in plasma were detected by NanoSight Analysis and western blotting. The results showed that the exosomes decreased significantly after GW4869 treatment (**Fig. S7a, b**). Cytotoxicity was determined using the CCK-8 assay. The results showed no significant cytotoxicity for various concentrations of GW4869 (**Fig. S7c**). The CCK-8 assay has been described in **Supplemental Methods**.

- Why do PTD + TBI OCN levels not correlate when comparing figures 1g and 2k?

Thank you for the comment. We apologize for the confusion in data analysis. We used different statistical calculation methods before, which led to incorrect statistical points on the Y-axis. We have revised the statistical methods after standardization (**Fig. 1g**).

- Figure 3g please include additional methods and details related to collection of TBI patient plasma exosomes (collecting location - peripheral blood?, collection date post-TBI; how many patients concentrations and what was their severity & type of TBI, age/sex match with healthy subject?).

Thank you for the comments. Detailed methods related to collection of TBI patient plasma exosomes were added on Page 17 of the revised manuscript.

• Figure 4f – why would FOXO4 and CBL proteins be upregulated in PTD+TBI bone tissue (histology) if they are negative regulators of osteogenesis, more bone is forming and the exosomes from TBI are supposed to be downregulating these pathways – this data seems contradictory, please clarify.

We apologize for the incorrect description, which has been revised as “We found that levels of FOXO4 and CBL were **decreased** at the bone defect site of TBI-PTD rats as compared with that of PTD rats, which suggests that FOXO4 and CBL may be involved in TBI-stimulated bone formation.” As shown in Fig. 4f and 4g, FOXO4 and CBL proteins were decreased in PTD+TBI bone tissue.

Fig. 5

• Please correct the orientation of Fig. 5a brain images according to the standard: <http://atlas.brain-map.org/atlas?atlas=1&plate=100960260#atlas=1&plate=100960088&resolution=10.47&x=5247.9998779296875&y=4032.0000648498535&zoom=-3>

Thank you for the excellent suggestion. We have replaced brain images according to the standard (**Fig. 5a**).

• Fig. 5d - exosomal miRNA levels in damaged neuronal cells using OGD assay. Oxygen deprivation may be a better in vitro model of ischemic brain injury model (stroke) than TBI – can the authors please justify. and there is no in vivo expression data of miRNAs in the hippocampus after TBI?

Thank you for the comments. We have provided a rationale for using oxygen-glucose deprivation as a model in vitro in the Results section on Page 9. The revised text reads, “Primary traumatic injury generally initiates a cascade of secondary injury mechanisms such as edema, hemorrhage, and decreased cerebral blood flow resulting in ischemic injury to brain tissue¹. While ischemic injury is a secondary consequence of TBI, ischemia is the primary mechanism of injury in instances that disrupt arterial cerebral blood flow leading to a lack of oxygen and glucose in brain tissue and therefore oxygen-glucose deprivation (OGD) is an in vitro model commonly used for TBI²⁻³.”

The expression of miR-328a-3p in hippocampus of rats was shown to be increased significantly after TBI (**Fig. 5a** and **Fig. S8a**).

1. Bramlett, H.M. & Dietrich, W.D. Pathophysiology of cerebral ischemia and brain trauma: Similarities and differences. *J Cerebr Blood F Met* 24, 133-150 (2004).

2. Salvador, E., Burek, M. & Forster, C.Y. An In Vitro Model of Traumatic Brain Injury. *Methods in molecular biology* (Clifton, N.J.) 1717, 219-227 (2018).

3. Wang, G.H., et al. Microglia/macrophage polarization dynamics in white matter after traumatic brain injury. *J Cerebr Blood F Met* 33, 1864-1874 (2013).

• Fig 5e – it is not totally clear if there are increased exosomes following liver injury. Is there any semi-quantification or replicate information that can be included?

Thank you for the suggestions. The Nanosight analysis and western blot results were quantitated. We found that rat plasma exosomes increased after liver injury, while rat plasma exosomes increased much more after TBI (**Fig. S13a, b**).

• Fig5f – The authors should provide some quantification of PKH67-labeled exosome uptake, it is difficult from a 3 cell image to know how ubiquitous this is. Why did MLO-Y4 cells not stain with Phalloidin-TRITC

We appreciate your excellent suggestions. We provided additional experiments data to address this point, in which MLO-Y4 cells were stained with phalloidin-TRITC (**new Fig. 5f**). Moreover, quantification of PKH67-labeled

exosomes mean fluorescence intensity was added.

- Figure 5g – Please provide a low magnitude image to show where in the fracture these images are taken. Please resolve the apparent increased miR-328a-3p (Fig 5g) and the increased FOXO4 IHC (4f) they seem to conflict.

Thank you for the suggestions. We have provided a low magnification image to show the bone defects (**Fig. 5g**). We apologize for the incorrect description, which has been revised in the main text as, “We found that levels of FOXO4 and CBL were **decreased** at the bone defect site of TBI-PTD rats as compared with that of PTD rats, which suggested that FOXO4 and CBL may be involved in TBI-stimulated bone formation.” As shown in **Fig. 4f** and **4g**, FOXO4 and CBL proteins were decreased in rat PTD sites.”

Fig. 7

- Details regarding the hydrogel are very slim. What is the concentration of exosomes loaded into the hydrogel? How was the hydrogel placed in the fracture defect and what are the hydrogel degradation kinetics? What are the release kinetics of the exosomes from the hydrogel? (IMPORTANT!)

Thank you for the comment. We have improved the details regarding the hydrogel in the methods section related to the rat tibial bone defect and calvarial defect model on **page 19**. It now reads, “Each defect was cleaned and then treated with methacrylated glycol chitosan (MeGC) hydrogels containing various nanoparticles (200 µg/mL) or left empty. The hydrogel was formed by exposing 40 µL of the suspension under visible blue light (400–500 nm, 500–600 mW/cm²; Bisco Inc., Schaumburg, IL, USA) in the presence of a photoinitiator, riboflavin (final concentration 6 µM)”. The concentration of exosomes loaded into the hydrogel was 200 µg/mL.

We have provided the results of additional experiments to address this point. Release kinetics of exosomes from hydrogels is shown in Fig. S16a. There were approximately 5% of loaded exosomes released from hydrogels in the first 24 h, and exosomes showed sustained release for up to 2 weeks. We have supplemented the details about exosome release kinetics from the hydrogels in the revised main text on page 26. It now reads, “To examine exosome release from hydrogels, exosomes were loaded on the hydrogels at a concentration of 50 µg/hydrogel and incubated at 37°C in 1 mL PBS in 12-well plates. At each time point 1 mL of PBS was collected and replenished with equal amount of fresh PBS. The amount of exosomes was calculated by A-50 Micro-PLUS flow cytometry for a period of 14 days.”

- Fig 7e-f – Better statistical data should be given for 7f (ANOVA results followed by Tukey HSD). It appears (7e) that Gel+Exo+NC increases bone formation by microCT, this does not come through in the stats for 7f, but these stats do not appear complete.

We appreciate your excellent suggestions. We have performed analysis of variance tests for several experiments, but we did not mention it in the previous manuscript. We have performed and corrected the multiple comparisons in Fig. 7 (one-way analysis of variance with a Turkey’s multiple comparisons test was performed).

- Can IHC or gene expression analysis be performed to confirm the delivery of the Gel+Exo+MiR-328 lead to effective decrease in FOXO4. This would help confirm results of Sup Fig 7A-B. Better statistical description (ANOVA results followed by Tukey HSD) should be added to S7b.

Thank you for the comment. We have provided the results of additional experiments to address this point. The immunohistochemical staining of FOXO4 showed that delivery of the Gel+Exo+MiR-328 lead to an effective decrease in FOXO4 (**Fig. S15e**). A better statistical description has been added in the Fig. 7 legend in the revised manuscript.

- Supplemental Figure 2 - Please provide an gross (whole brain) histological image of the brain to compliment the high magnification image in S2b.

Thank you for the comment. We have provided a gross (whole brain) histological image of the brain to compliment the high magnification image (**Fig. S2b**).

- Supplemental Figure 5 – This figure would be greatly improved by including de-identified information on TBI patients (collecting location; peripheral blood?, collection date post-TBI, severity & type of TBI, age/sex match with healthy subject?)

Thank you for the comments. Detailed methods related to collecting TBI patient plasma exosomes were added on **page 17**. It now reads, “Plasma exosomes were collected from TBI patients (10 males) and health control (10 males) between 17 and 72 years of age admitted to the Third Affiliated Hospital of Southern Medical University. Venous blood samples were collected from all patients immediately after hospital admission (always within 8 h after trauma). Healthy matched individuals were selected among donated plasma samples as control. They were trauma-free and matched for age and sex status. No subjects had histories of neurological disease, psychiatric illness, head injury, stroke, nor learning disabilities.”

Are the claims appropriately discussed in the context of previous literature?

The discussion is well written.

Thank you for the comment.

Reviewer 3

In this study, the Authors are investigating the underlying mechanisms for bone healing in patients and animals with concomitant traumatic brain injury (TBI). Their study focuses on deciphering the crosstalk between bone and brain in the pre-clinical and clinical settings of TBI, where they mainly attributed the bone healing effects to the exosomes released by injured neurons containing osteogenic miRNAs.

We thank the reviewer for their careful reading. We also thank the reviewer for the constructive suggestions that have significantly improved the manuscript.

Major comment #1: The presence of fibronectin on exosomes identified as mechanism of action for targeting osteoprogenitors.

This comment refers to the statement made by the Authors on Page 12 “Moreover, we found that increased FN1 expression on exosomal surface contributed to the direction of TBI exosomes to bone”. FN1 is an extracellular matrix protein known to have multiple binding partners, including integrins (also described by the Authors in Figure 8), which are molecules found on a variety of cell types, including the osteoprogenitors. Therefore, the high abundance of FN1 found on neuronal-derived exosomes may lead to an increase in binding capacity of the exosomes to their target cells. This assumption should be further discussed by the Authors when discussing their findings as they have not provided a detailed explanation as to how this would work (i.e. FN1 contributing to the direction of TBI exosomes). Overall, this statement is not well-supported by their results and may be overstated.

Thank you for the comment. We have provided the results of additional experiments to address this point. Previous studies have found that extracellular matrix protein FN plays an important role in the targeting of exosomes¹⁻². One of the unique features of FN is its ability to bind a large number of cell adhesion receptors, growth factors, and extracellular matrix proteins. Microvesicles released by human cancer cells can be taken up by fibroblasts in an FN-dependent manner¹. Liver endothelial cells release SK1-containing exosomes, which engage with hepatic stellate cells via FN-integrin-dependent adhesion². All these studies, however, were carried out *in vitro*, and, therefore, the tissues or organs that the exosomes with FN will eventually reach *in vivo* are unknown. In this study, we found that following TBI, circulating exosomes and their cargo miRNAs were transferred selectively to peripheral organs and preferentially to the bone (Fig. S13c). In additional experiments, the organ distribution of fluorescence signals in rats were detected 24 h after intracerebroventricular (ICV) injection with PKH67-labelled exosomes (new Fig. 5e and S12), and the results showed that PKH67-labelled exosomes accumulated in rats limb bone in the TBI-exo group, which was much more than that in other groups. Importantly, these effects were completely reversed by an inhibitory peptide of FN1 GRGDNP (Gly-Arg-Gly-Asp-Asn-Pro) that inhibited cell attachment to FN1 (Fig. 6k, l).

1. Purushothaman, A., et al. Fibronectin on the Surface of Myeloma Cell-derived Exosomes Mediates Exosome-Cell Interactions. *J Biol Chem* 291, 1652-1663 (2016).

2. Wang, R.S., et al. Exosome Adherence and Internalization by Hepatic Stellate Cells Triggers Sphingosine 1-Phosphate-dependent Migration. *J Biol Chem* 290, 30684-U30651 (2015).

Major comment #2: Confirming the cell-of-origin releasing the exosomes. Further evidence regarding the cell-of-origin that is releasing the exosomes into the plasma of patients and rat model of TBI should be provided to support the claim that exosomes from injured neurons are being released into the plasma. EVs share surface molecules with the releasing cell, so neuronal markers should be validated for their presence on exosomes. A2M molecule was mentioned by the Authors on Page 10; however, additional supportive markers for neuronal origin should be added. Furthermore, mass spectrometry (MS) allows for the identification of candidate proteins and validation by orthogonal methods (ELISA, WB, FC, etc) is often used to validate the protein of interest identified

(WB, FC for examples). A2M was identified as a candidate protein by MS, but not further validated.

Thank you for the comment. We have provided the results of additional experiments to address this point. We performed proteomics analyses. The results showed that A2M, which has been reported to be a marker for neuronal injury¹⁻², increased 3.099 times after TBI treatment in rat plasma exosomes (**new Table. S3**), implying that these exosomes might originate from injured neurons. We have provided further validation by western blot analysis. The results showed that A2M was increased in exosomes isolated from TBI rat plasma, when compared with that of the sham group (**new Fig. S14a**). In addition, western blot results showed that MAP2 and UCHL1, two makers of neuron cells, were increased in exosomes isolated from TBI rat plasma (**new Fig. S14a**).

1. Miszczuk, D., Debski, K.J., Tanila, H., Lukasiuk, K. & Pitkanen, A. Traumatic Brain Injury Increases the Expression of Nos1, A beta Clearance, and Epileptogenesis in APP/PS1 Mouse Model of Alzheimer's Disease. *Mol Neurobiol* 53, 7010-7027 (2016).
2. Varma, V.R., et al. Alpha-2 macroglobulin in Alzheimer's disease: a marker of neuronal injury through the RCAN1 pathway. *Mol Psychiatr* 22, 13-23 (2017).

Major comment #3: Confirming the cross-talk between the injured neurons and osteoprogenitor cells through exosomes. Have the Authors previously published in vitro data supporting the direct transfer of EVs from neurons to osteoprogenitor using an in vitro model such as Transwell assay? This assay allows to seed 2 cell types in separate chambers where exosomes transfer and functional effects (e.g. differentiation of osteoprogenitors) can be assessed. This would further strengthen their hypothesis of neuronal EV transfer to osteoprogenitor cells.

We appreciate your suggestions. We have provided the results of additional experiments to address this point. We have performed a Transwell assays to analysis the cross-talk between the injured neurons and osteoprogenitor cells. The results of western blot of RUNX2 and ALP staining showed that the osteoblastic differentiation of BMSC and MC3T3-E1 cells increased after co-cultured with OGD treated primary hippocampal neurons (**new Fig. S11d, e**). We have improved the Methods section with co-culture experimental details added in **Supplemental Methods**. It now reads, "Co-culture experiments. The well inserts with a 0.4 mm pore size filter (BD Falcon) for 6-well plates were used following the manufacturer's instructions. Primary hippocampal neurons were seeded into the well inserts with DMEM/F12 medium or after OGD treatment. BMSC and MC3T3-E1 cells were seeded into the 6-well plates and induced into osteoblasts by addition of α -MEM. After differentiation, osteoblasts were washed with PBS, and then co-cultured for 7 d according to the experimental protocol. All co-culture experiments were done using α -MEM with exosome-free FBS (Life Technologies)."

Major comment #4: This one relates to: Page 5 of 47 in section entitled "Plasma Exosomes from TBI Patients and Rats Contribute to TBI-Stimulated Osteogenesis". The Authors stated: "We found that plasma from patients or rats with TBI but not controls promoted osteoblastic differentiation of primary cultured rat bone marrow stem cell (BMSC) and osteoprogenitor cell line, MC3T3-E1, as manifested by an enhanced alkaline phosphatase staining and Runx2 expression in TBI plasma treated cells (Fig. 2a, b and Supplementary Fig. 5)."

- Healthy controls are mentioned in this section, but without any prior description in the Methods. The Authors should revise this section.

Thank you for the comment. We have revised the healthy controls description in the Methods section on page 17. It now reads, "Healthy matched individuals were selected among donated plasma samples as controls. They were trauma-free and matched for age and sex status. No subjects had histories of neurological disease, psychiatric illness, head injury, stroke, or learning disabilities."

- Differentiation results shown in Figures 2a and 2b as well as in Suppl. Figure 5 for controls clearly showed osteoblastic differentiation from the addition of control plasma. By comparing the differentiation results from

control to TBI plasma, it is true that the differentiation potential is enhanced, but it is incorrect to say that controls did not promote osteoblastic differentiation as the Authors stated.

Thank you for the comment. We apologize for the confusion. We have clarified this point in the Results section on **page 5**, which now reads, “We found that plasma from TBI patients or rats promoted osteoblastic differentiation (**Fig. 2a, b** and Supplementary **Fig. 5a**)”.

- Results from qualitative assessment of Western Blot and chemical staining assays should also be quantified and analyzed for statistical significance.

Thank you for your suggestions. We have added the quantitations of western blotting and chemical staining assays results. We have also added a better statistical description (A one-way analysis of variance with Turkey’s multiple comparisons tests or Student’s unpaired t-tests were performed.) and P values (*P < 0.05[**P < 0.01[***P < 0.001[****P < 0.0001) were included in each figure legend.

Major comment #5: This one relates to the BMSC sample mentioned on Page 5 in section entitled “Plasma Exosomes from TBI Patients and Rats Contribute to TBI-Stimulated Osteogenesis”.

The BMSC sample described on Page 5 in this section seems to refer to only 1 rat (i.e. 1 biological replicate) as mentioned in the Methods on Page 18 – “The BMSC were harvested from the femora and tibiae of SD rats. The cells from one rat were seeded onto [...] for 48 h”. The authors should clearly explain why they have only used 1 biological sample for all the testing of the BMSC response. Does this also apply to every experiment throughout the manuscript conducted using the BMSC such as the one mentioned on Page 7 where it is stated: “ To determine the effects of these miRNAs on osteoblast formation, MC3T3-E1 cells and primary rat BMSCs were transfected with miRNA mimics or negative controls”?

We apologize for the confusion, and have revised this in the Methods section with details of cell line cultures described on page 20. It now reads, “The BMSCs were harvested from the femora and tibiae of SD rats. Femurs and tibiae were aseptically removed, and bone marrow was flushed with phosphate-buffered saline (PBS) via a sterile syringe. Bone marrow samples were centrifuged for 5 min at 1000 × g and then placed in minimum essential medium alpha (α-MEM; Gibco, Gaithersburg, MA, USA) and replenished with 10% fetal bovine serum (FBS, Gibco, Carlsbad, CA, USA) and 1% penicillin and streptomycin (Thermo), then incubated at 37°C in an incubator containing 5% CO₂ for 48 h. Subsequently, the non-adherent cells were discarded, whereas the adherent cells were allowed to grow to 80% confluency and these cells were defined as passage 1 cells (P1). P3 cells were used for all experiments.”

We have stated the number of experiments and samples in each figure legend. We have also redesigned all figures to show individual data points and the biological replicate numbers.

Major comment #6: This one relates to the CD63-positive “particles” increased after TBI. On Page 9, the Authors state: “Importantly, CD63-positive particles increased dramatically after TBI, indicating that the production of exosomes was active in the damaged brain (Fig. 5b, c)”. The Authors have not presented clear evidence that the CD63-positive signal refers to particles instead of cells (or both). CD63 is not exclusive to extracellular vesicles (EVs or exosomes). Cells are known to express CD63, including endothelial cells for example. Furthermore, the Authors need to provide further evidence to support this claim: “Interestingly, most of the CD63 was expressed in microtubule-associated protein 2 (MAP2)-positive, but not glial fibrillary acidic protein (GFAP)-positive cells, indicating that damaged neuronal cells, but not astrocytes secreted exosomes vigorously.” No data are provided for the identification of the “damaged neuronal cells”, as well as the secretion of exosomes by these cells. The Authors should clarify this.

Thank you for the comment. We agree with you that CD63 is not exclusive to extracellular vesicles (EVs or exosomes). We have replaced the sentence with “Importantly, immunofluorescence staining showed that expression of CD63 in rat brain was intensively induced by TBI (new Fig. 5b)”. Furthermore, primary hippocampal neurons were used for *in vitro* analysis in additional *in vivo* experiments. Primary cultured hippocampal neurons were subjected to OGD *in vitro* to simulate ischemia-like conditions for the identification of the “damaged neuronal cells”. NTA and Western blot analysis of the primary hippocampal neuron exosomes showed that the exosomes release was increased after OGD treatment (new Fig. S11b, c).

Major comment #7: This one refers to the distribution of PKH67-exosomes evaluated by biophotonic imaging at 6 h after injections. The Authors should provide quantifiable data for the signal of all biophotonic imaging experiments. Statistical analyses should also be performed.

Thank you for the comment. We have provided quantitative data for the signal of all biophotonic imaging experiments (Fig. 5e and Fig. S12). We have added a better statistical description (A one-way analysis of variance with Turkey’s multiple comparison tests or Student’s unpaired t-tests were performed.) and P-values (*P < 0.05; **P < 0.01; ***P < 0.001; ****P < 0.0001) were added to each figure legend.

Major comment #8: This one refers to the Tandem Mass Tag (TMT) labeling and proteomics analyses.

- Details regarding the mass spectrometer (MS) they have used for analysis as well as details regarding the MS method of data analysis are missing. The Authors should provide this information.

Thank you for the comment. We have provided the details regarding the mass spectrometer (MS) in **Supplemental Methods**.

It now reads, “Protein Extraction, Digestion, and TMT Labeling. The exosomal proteins of samples was suspended in lysis buffer (8M urea, 1% SDS with Protease Inhibitor Cocktail) and treated by ultrasound at 40 kHz, 40 W for 2 min, and incubated on ice for 60 min. After centrifugation at 12000 × g for 30min at 4°C, the protein concentration was determined using a BCA Protein Assay Kit (Thermo Scientific).

Proteins were digested as follows: A 100 µg of protein sample was taken and the volume was replenished to 90 µL with lysate. Tris(2-carboxyethyl) phosphine (TCEP) (final concentration of 10 mM) was added and the mixture was incubated at 37°C for 60 min. Then, 40 mM iodoacetamide was added to the mixture, and the mixture was incubated at room temperature in the dark for 40 min. Prechilled acetone was then added and incubated for 4 h at -20°C before centrifugation at 1000 × g for 20 min. All collected proteins were transferred to a new tube, followed by trypsin digestion with a substrate ratio of 1:50 (w/w), at 37°C overnight according to the manufacturer’s protocols. The digested peptides were labeled with a TMT Reagent Kit according to the manufacturer’s protocol (Thermo Fisher; Art. No.90111). After incubation for 2 h at room temperature, hydroxylamine was added to the reaction mixture for 15 min. Finally, all samples were pooled, desalted and vacuum-dried.

Reverse-phase Chromatography with High-pH Separation. The TMT-tagged peptides were mixed and reconstituted in UPLC loading buffer, followed by loading on an Acquity UPLC BEH C18 column (1.7 µm, 2.1 mm×150 mm; Waters, USA). The peptides were eluted at a flow rate of 200 µL/min with a linear gradient of 0–5% solvent B (80% acetonitrile (ACN) 20 mM NH₄HCO₂, pH 10) for 2 min, 5% solvent B for 15 min, 5–30% solvent B for 8 min, 30–36% solvent B for 3 min; 36–42% solvent B for 1 min; 42–100% solvent B for 1 min, and a hold for 8 min. A total of 20 fractions were collected based on peak type and time, and they were combined into 10 fractions per sample. Each of them was dried by vacuum centrifugation.

LC-MS/MS analysis. The labeled peptides were dissolved in solvent A (2% ACN with 0.1% formic acid), and then centrifuged (4°C, 12000 × g, 20 min), and the supernatant was transferred to the sample tube. The Nano LC-MS/MS was carried out using a Q-Exactive MS (Thermo Scientific) coupled online to the UPLC system

(Thermo Dionex). The peptide was loaded onto a C18 column (75 μm \times 25 cm; Thermo, USA) at a flow rate of 300 nL/min. Peptides were eluted from the column by a linear gradient from 5% solvent B (80% ACN with 0.1% formic acid) to 23% solvent B for 40 min; 29% solvent B for 10 min; 48% solvent B for 7 min; 100% solvent B for 1 min, and a hold for 8 min, after which the mobile phase was returned to 0% solvent B for 5 min. The 20 precursors with the most intense signals were selected for higher-energy collisional dissociation (HCD) fragmentation with a normalized collision energy, dynamically choosing the most abundant precursor ions from a scan (m/z 350–1300). HCD spectra were acquired in the Orbitrap with a 60,000 resolution at m/z 200 and the fixed first mass was set to m/z 100.

Database Search and Quantitative Proteomic Analysis. The RAW data files were analyzed using Proteome Discoverer (Thermo Scientific, Version 2.4) against a Uniprot Rattus norvegicus database (version 20200617, 35779 sequences). The MS/MS search criteria were as follows: Mass tolerance of 20 ppm for MS and 0.02 Da for MS/MS tolerance, trypsin as the enzyme with two missed cleavages allowed, carbamido methylation of cysteine and the TMT of N-terminus and lysine side chains of peptides as fixed modifications, and methionine oxidation as dynamic modifications, respectively. High confidence peptides were used for protein identifications by setting a target false discovery rate (FDR) threshold of 1% at the peptide level. Identified proteins, which had at least one unique peptide were used for protein identifications. The thresholds of fold change (> 1.2 or < 0.83) and P-value < 0.05 were used to identify differentially expressed proteins (DEPs). A total of 273 proteins expressed were identified as belonging to the proteome of *Solanum tuberosum*. Then, we found 36 upregulated and 68 downregulated proteins in group TBI compared with group sham.”

- The proteomics experiment and the MS results have not yielded many protein candidates. This could be due to inadequate EV lysis efficiency as the few proteins identified by MS are of extracellular origins (e.g. extracellular matrix proteins, IgG, etc) or bound to membranes (e.g. Apolipoproteins). From the short list of proteins identified by MS (Suppl. Table) only a very few cargo proteins (intra) were identified. A low percentage of detergent (e.g. 0.1% SDS) would have yielded a better EV lysing process as well as enhanced protein solubility; hence, a higher yield of proteins would have been identified. The Authors should comment on the choice of their proteomics method and clarify their results.

- TMT labeling allows for the multiplexing of up to 16 samples (i.e. 16 different isotopic labels can be multiplexed without compromising on protein identification and quantitation). The Authors have used only 1 biological sample (i.e. 1 rat) in each group to conduct TMT proteomics. If the authors have pooled multiple biological samples for each group (i.e. tag 129 and tag 131), they should describe their approach. If they Authors have only used 1 biological sample per group, as currently stated, they should clarify why they have not used additional biological samples. Furthermore, at the validation stage of the MS data, the Authors should have included more biological samples for result validation as MS results require this kind of verification for the validity of the results.

Thank you for the comment. We provide additional proteomics analysis to address this point. We apologize for the confusing information in the previous manuscript. The exosomal proteins of samples were suspended in lysis buffer (8M urea, 1% SDS with protease inhibitor cocktail) and treated by ultrasound at 40 kHz, 40 W for 2 min, then incubated on ice for 60 min. A total of 273 proteins expressed were identified as belonging to the proteome of *Solanum tuberosum* (**supplementary table 3**). Each group had four biological samples in this revised manuscript. The result showed that A2M and FN1 increased in rat plasma exosomes after TBI treatment (**new Table. S3**).

Minor comment #1: The Supplementary Table 2 refers to “Clinical features of patients with fractures and concomitant TBI”. On Page 7, the Suppl. Table 2 refers to the miRNA profile. Please correct.

Thank you for the comment. We have corrected the issue in that **Supplementary Table 1**, which refers to “Clinical

features of patients with fractures and concomitant TBI.” and **Suppl. Table 2** refers to the miRNA profile.

Minor comment #2: I think that the authors meant to write “that levels of FOXO4 and CBL were decreased” and not “increased”. To be found here: “We found that levels of FOXO4 and CBL were increased at the bone defect site of TBI-PTD rats as compared with that of PTD rats, which suggests that FOXO4 and CBL may be involved in TBI-stimulated bone formation (Fig. 4f, g)”.

We appreciate your excellent suggestions. We apologize for the confusion, which has been revised as “We found that levels of FOXO4 and CBL were **decreased** at the bone defect site of TBI-PTD rats as compared with that of PTD rats, which suggested that FOXO4 and CBL may be involved in TBI-stimulated bone formation.” As we showed in **Fig. 4f, g**, FOXO4 and CBL proteins were downregulated in PTD+TBI bone tissue.

Minor comment #3: The article would benefit from having a Table where the samples used in the animal study would be described. For example, each sample would be identified with a unique identifiers and a description of which one was used for which purpose should be included. This would bring clarity as to how many animals and which one was used for the different analyses/methods. Overall, the number of samples and which ones used throughout the manuscript is not clear.

We appreciate your suggestions. We have redesigned all figures to show individual data points as the biological replicate number and have stated the number of experiments and samples in each figure legend in the manuscript.

Minor comment #4: Referring to “Exosomes isolated from plasma and HT22 cells supernatant 24 h after OGD treatment were labeled with PKH67, 10 µg/mL of monoclonal antibodies FN1-PE, and analyzed by nanoscale Apogee flow cytometer (FC)” – What is the total quantity of antibody used for the detection of FN1? Did the Authors do a titration experiment for confirming the use of 10 µg/mL?

Thank you for the comment. Titration of all detecting antibodies was performed with pre-conjugated clones and dilutions were determined from the original concentration (µg/mL) as provided by the manufacturer (**Fig. S14b**). Dilutions of pre-conjugated antibodies were also performed using 0.20 µm filtered PBS. Ten µL of exosomes were incubated with 100 ng of FN1-PE in each experiment.

Minor comment # 5: Figures 4h, i, j and k. Quantitation and statistical significance are required. Please add.

Thank you for the comment. We have performed quantitation and statistical significance in **Fig. 4h, i, j, and k**.

Minor comment #6: Figure 5: Most of the quantification data and statistical computations are missing. Please add.

We apologize for the missing information. As the reviewer suggested, we have added quantitation data and statistical computations in **Fig. 5**.

Minor comment # 7: Figure 6: Most of the quantification data and statistical computations are missing (b, c, f, l). Please add.

We apologize for the missing information. As the reviewer suggested, quantitation data and statistical computations have been added in **Fig. 6**.

Reviewers' Comments:

Reviewer #1:

Remarks to the Author:

Overall, the authors thoroughly addressed all of our concerns about the original manuscript, and their responses to the other reviewers substantially strengthened the manuscript. Wei et al. now present a comprehensive set of findings that are of great interest to the field. Therefore, we recommend this manuscript for publication in Nature communications.

Reviewer #2:

Remarks to the Author:

This article represents a revision to the original manuscript following the comment of three Reviewers. Overall this is a high-quality research study that investigates whether exosomes collected from the circulation of patients following traumatic brain injury (TBI) contribute mechanistically to the observed clinical phenomenon of accelerated fracture repair with concomitant TBI. The paper identified miR-328 and miR-150 in the exosomes isolated from TBI patient plasma and demonstrate that they inhibit FOXO4 and CBL, the negative regulators of osteogenesis, respectively. The paper also found that exosomes secreted after TBI have more fibronectin (FN1) on their membrane surface and suggest that FN1 selectively directs exosomes to bone and osteoprogenitors.

The Authors have adequately and rigorously addressed the Reviewers concerns. It is recommended that this article is accepted pending minor editorial adjustments to the figures (see below).

With the additional figures a some of the text in the figures is not easily readable even at high magnification. Please check at editorial level: Fig 3A, Fig 5E, Fig 5F, Fig 6a/6D, Fig 7A

Reviewer #3:

Remarks to the Author:

The Authors have revised the manuscript to address the comments, which is satisfactory; however, there are remaining follow-up comments to address.

#1- Following up on the Major comment #1 (Re: The presence of fibronectin on exosomes identified as mechanism of action for targeting osteoprogenitors). The information provided by the Authors is appreciated; however, additional information is needed to fully address this point. The Authors stated that: "In additional experiments, the organ distribution of fluorescence signals in rats were detected 24 h after after intracerebroventricular (ICV) injection with PKH67-labelled exosomes (new Fig. 5e and S12), and the results showed that PKH67-labelled exosomes accumulated in rats limb bone in the TBI-exo group, which was much more than that in other groups. Importantly, these effects were completely reversed by an inhibitory peptide of FN1 GRGDNP (Gly-Arg-Gly-Asp-Asn-Pro) that inhibited cell attachment to FN1 (Fig. 6k, l)."

The remaining comments to address: The Authors should described in more detail their PKH67 exosome labelling protocol (referring to line 505 – missing procedure and concentration) as well as their FN1 peptide GRGDNP procedure in terms of concentration and route of injection. In addition, if the Authors have biophotonic images of the brain following the ICV injection of PKH67-labeled exosome it should be shown in their panel of organ as EVs are known to cross the BBB (Figure 5e).

#2 – Regarding "Major comment #3: Confirming the cross-talk between the injured neurons and osteoprogenitor cells through exosomes. Have the Authors previously published in vitro data supporting the direct transfer of EVs from neurons to osteoprogenitor using an in vitro model such as Transwell assay? This assay allows to seed 2 cell types in separate chambers where exosomes transfer and functional effects (e.g. differentiation of osteoprogenitors) can be assessed. This would further strengthen their hypothesis of neuronal EV transfer to osteoprogenitor cells." The Authors have provided additional data to address this, where they showed that the

osteoblastic differentiation of BMSC and MC3T3-E1 cells increased after co-cultured with OGD treated primary hippocampal neurons. What is currently missing to complete these results is the evidence of the effect of the OGD treatment on the treated primary hippocampal neurons. Do the Authors have assess the death of the neurons following OGD? This is necessary not only to assess whether the OGD treatment was successful, but also to which degree, which is important to understand the treatment outcome.

#3- This refers to the use of GW4869. Can the Authors comment of the specificity of GW4869 in its use to inhibit exosomes? How specific is it? Should they add a control for its diluent DMSO?

#4- Referring to Figure 1: The Authors should be more explicit regarding how the enrolled patients were analyzed in Figure 1, as they mentioned that they compared the recovery rate of 12 patients with femoral fracture and concomitant TBI and 12 patients with femoral fracture only. Only 4 patients per group are shown in Figure 1. The way the data is currently presented, the 12 patients have been assigned to 3 groups (week 2, week 4, week 6), meaning that each patients was not followed at the 3 different timepoints. Please explain.

Additional minor comments:

#1: The use of the term "Exosomes" (a specific populations of EVs) should be revised as per ISEV guidelines: <https://www.tandfonline.com/doi/full/10.1080/20013078.2018.1535750>.

Paper of reference: Clotilde Théry et al. Minimal information for studies of extracellular vesicles 2018 (MISEV2018): a position statement of the International Society for Extracellular Vesicles and update of the MISEV2014 guidelines.

From the ISEV guidelines: "ISEV endorses "extracellular vesicle" (EV) as the generic term for particles naturally released from the cell that are delimited by a lipid bilayer and cannot replicate, i.e. do not contain a functional nucleus. Since consensus has not yet emerged on specific markers of EV subtypes, such as endosome-origin "exosomes" and plasma membrane-derived "ectosomes" (microparticles/microvesicles) [3,4] assigning an EV to a particular biogenesis pathway remains extraordinarily difficult unless, e.g. the EV is caught in the act of release by live imaging techniques. Therefore, unless authors can establish specific markers of subcellular origin that are reliable within their experimental system(s), authors are urged to consider use of operational terms for EV subtypes that refer to a) physical characteristics of EVs, such as size ("small EVs" (sEVs) and "medium/large EVs" (m/IEVs), with ranges defined, for instance, respectively, < 100nm or < 200nm [small], or > 200nm [large and/or medium]) or density (low, middle, high, with each range defined); b) biochemical composition (CD63+/CD81+- EVs, Annexin A5-stained EVs, etc.); or c) descriptions of conditions or cell of origin (podocyte EVs, hypoxic EVs, large oncosomes, apoptotic bodies) in the place of terms such as exosome and microvesicle that are historically burdened by both manifold, contradictory definitions and inaccurate expectations of unique biogenesis."

#2- Line #241: Electroporation settings and procedure are missing.

#3- Line #342: "With" not "were" should be read.

#4- Line #349: How many Healthy matched individuals were included?

#5- Line #410: What is the concentration of calcein being used?

#6- Line #196: What does the 800 ug relate to (EVs)?

#7- Line #209: There is a typographical error: osteoprogenitors and not psteoprogenitors

#8- Line #141: There is a typographical error: healthy and not health controls

We thank the reviewers for their time and thoughtful comments. Below are point-by-point responses.

Reviewer #1 (Remarks to the Author):

Overall, the authors thoroughly addressed all of our concerns about the original manuscript, and their responses to the other reviewers substantially strengthened the manuscript. Wei et al. now present a comprehensive set of findings that are of great interest to the field. Therefore, we recommend this manuscript for publication in Nature communications.

We thank the reviewer for the careful reading and the positive comments: “Wei et al. now present a comprehensive set of findings that are of great interest to the field.”

Reviewer #2 (Remarks to the Author):

This article represents a revision to the original manuscript following the comment of three Reviewers. Overall this is a high-quality research study that investigates whether exosomes collected from the circulation of patients following traumatic brain injury (TBI) contribute mechanistically to the observed clinical phenomenon of accelerated fracture repair with concomitant TBI. The paper identified miR-328 and miR-150 in the exosomes isolated from TBI patient plasma and demonstrate that they inhibit FOXO4 and CBL, the negative regulators of osteogenesis, respectively. The paper also found that exosomes secreted after TBI have more fibronectin (FN1) on their membrane surface and suggest that FN1 selectively directs exosomes to bone and osteoprogenitors.

The Authors have adequately and rigorously addressed the Reviewers concerns. It is recommended that this article is accepted pending minor editorial adjustments to the figures (see below).

With the additional figures a some of the text in the figures is not easily readable even at high magnification. Please check at editorial level: Fig 3A, Fig 5E, Fig 5F, Fig 6a/6D, Fig 7A.

Thank you for the positive comments and suggestions. Fig. 3A, Fig. 5E, Fig. 5F, Fig. 6a/6D, and Fig. 7A have been revised in the manuscript. We appreciate the constructive suggestions, which have significantly improved the manuscript.

Reviewer #3 (Remarks to the Author):

The Authors have revised the manuscript to address the comments, which is satisfactory; however, there are remaining follow-up comments to address.

#1- Following up on the Major comment #1 (Re: The presence of fibronectin on exosomes identified as mechanism of action for targeting osteoprogenitors). The information provided by the Authors is appreciated; however, additional information is needed to fully address this point.

The Authors stated that: “In additional experiments, the organ distribution of fluorescence signals in rats were detected 24 h after after intracerebroventricular (ICV) injection with PKH67-labelled

exosomes (new Fig. 5e and S12), and the results showed that PKH67-labelled exosomes accumulated in rats limb bone in the TBI-exo group, which was much more than that in other groups. Importantly, these effects were completely reversed by an inhibitory peptide of FN1 GRGDNP (Gly-Arg-Gly-Asp-Asn-Pro) that inhibited cell attachment to FN1 (Fig. 6k, l).”

The remaining comments to address: The Authors should describe in more detail their PKH67 exosome labelling protocol (referring to line 505 – missing procedure and concentration) as well as their FN1 peptide GRGDNP procedure in terms of concentration and route of injection. In addition, if the Authors have biophotonic images of the brain following the ICV injection of PKH67-labeled exosome it should be shown in their panel of organ as EVs are known to cross the BBB (Figure 5e).

Thank you for the careful reading and suggestions. We apologize for the confused expression of peptide GRGDNP procedure. GRGDNP is a synthetic peptide, which competitively inhibits fibronectin-integrin binding containing RGD peptide sequences. The PKH67 exosome labeling protocol and peptide GRGDNP procedure have been added to the revised Supplementary Material. It now reads, “The isolated sEVs diluted in PBS were added to 0.5 mL of Diluent C. Two μ L of PKH67 dye was added and incubated for 4 min at room temperature. Two mL of 1% BSA/PBS was added to bind excess dye. The synthetic peptide, GRGDNP (300 μ g/mL) was incubated with labeled sEVs (50 μ g/mL) for 30 min.”

The procedure for GRGDNP intracerebroventricular injection has been added in the revised manuscript. It now reads, “For intracerebroventricular injection of sEVs, rats were anesthetized with 2.5% isoflurane and placed in a stereotaxic apparatus. A portion of the parietal skull was carefully removed, and a guide cannula was then inserted into the right lateral ventricle (coordinates from the bregma, -1.5 mm; medial/lateral, \pm 1.0 mm; dorsal/ventral, -3.2 mm) and fixed with dental cement. To verify the injected location, the guide cannula was connected to a polyethylene (PE) pipe that was filled with artificial cerebrospinal fluid. Then, the PE pipe was vertically inserted into the right lateral ventricle through the guide cannula. When the liquid level in the PE pipe was dropped-down, it implied that the PE pipe was successfully inserted into the lateral ventricle. Then, after the rat recovered from surgery, 10 μ L of either PKH-67-labeled sEVs (20 μ g/ μ L) or phosphate-buffered saline (PBS, 0.01 M) was micro-infused into the lateral ventricle using a 33-gauge infusion cannula connected to a microsyringe (Gao, Shanghai). The sEVs were injected with the use of an electric microinjection pump at a flow rate of 0.5 μ L/min. The cannula remained in situ for at least 5 min after infusion and was then slowly withdrawn.”

Biophotonic images of the brain following the ICV injection of PKH67-labeled sEVs have been added to Fig. 5e in the revised version.

#2 – Regarding “Major comment #3: Confirming the cross-talk between the injured neurons and osteoprogenitor cells through exosomes. Have the Authors previously published in vitro data supporting the direct transfer of EVs from neurons to osteoprogenitor using an in vitro model such as Transwell assay? This assay allows to seed 2 cell types in separate chambers where exosomes transfer and functional effects (e.g. differentiation of osteoprogenitors) can be assessed. This would further strengthen their hypothesis of neuronal EV transfer to osteoprogenitor cells.”

The Authors have provided additional data to address this, where they showed that the osteoblastic differentiation of BMSC and MC3T3-E1 cells increased after co-cultured with OGD treated primary hippocampal neurons. What is currently missing to complete these results is the evidence of the effect of the OGD treatment on the treated primary hippocampal neurons. Do the Authors have assess the death of the neurons following OGD? This is necessary not only to assess whether the OGD treatment was successful, but also to which degree, which is important to understand the treatment outcome.

Thank you for the comment. We provide additional experiments to address this point. CCK-8 assays were performed to assess the cell viability of the neurons following OGD (new Fig. S14c). The results showed that OGD treatment was successful because the viability of neurons following OGD was significantly decreased, which has also been reported in previous studies [1, 2].

#3- This refers to the use of GW4869. Can the Authors comment of the specificity of GW4869 in its use to inhibit exosomes? How specific is it? Should they add a control for its diluent DMSO?

Thank you for your comments. GW4869 is a cell-permeable, symmetrical dihydroimidazo amide compound that acts as a potent, specific, noncompetitive inhibitor of membrane neutral sphingomyelinase (nSMase) [3]. SMase is found in many types of compartments in all cells, including the Golgi apparatus, endosomes, and cell membranes, so its activity is not only linked to exosomes generation but also MV shedding. Although GW4869 has been extensively studied and reported to reduce sEV release *in vitro* and *in vivo* [4-8], it does not specifically inhibit exosomes. In this study, the concentration of sEVs isolated from the plasma of rats was detected by NanoSight analysis and western blotting (Fig. S7a, S7b). The results showed that sEVs in rat plasma decreased after GW4869 treatment.

We used DMSO saline as the control group, which has been used in previous studies [4, 9, 10]. We have revised the sentence for GW4869 administration in the manuscript. It now reads, "Micro-CT 3D structure image of proximal tibial defect healing at 7 and 14 days postsurgery with intraperitoneal injection of GW4869 or DMSO saline." "GW4869 was dissolved in dimethylsulfoxide (DMSO) at 8 mg/mL. The working solution was prepared in 0.9% normal saline, freshly made before use with a final concentration of 0.3 mg/mL (2.5 µg/g body weight), with 3.75% DMSO saline as a control. Rats were injected with the working solution twice a week."

#4- Referring to Figure 1: The Authors should be more explicit regarding how the enrolled patients were analyzed in Figure 1, as they mentioned that they compared the recovery rate of 12 patients with femoral fracture and concomitant TBI and 12 patients with femoral fracture only. Only 4 patients per group are shown in Figure 1. The way the data is currently presented, the 12 patients have been assigned to 3 groups (week 2, week 4, week 6), meaning that each patients was not followed at the 3 different timepoints. Please explain.

Thank you for the comment. Clinical evidence over the last five decades has shown that concomitant TBI accelerates bone healing [11-14]. In this manuscript, we collected clinical data from 12 patients to confirm this phenomenon. Unfortunately, patient compliance was not as previously expected. Thus, each patient was followed at multiple time points, and we assigned 12 patients to three groups (week 2, week 4, and week 6) to compare them to 12 patients with simple shaft fractures. We have revised the methods section. It now reads, "Among these 24 patients,

there were 12 patients with simple shaft fractures and 12 patients with shaft fractures and concomitant TBI who were assigned to three groups (week 2, week 4 and week 6, with four patients per group).”

Additional minor comments:

#1: The use of the term “Exosomes” (a specific populations of EVs) should be revised as per ISEV guidelines: <https://www.tandfonline.com/doi/full/10.1080/20013078.2018.1535750>.

Paper of reference: Clotilde Théry et al. Minimal information for studies of extracellular vesicles 2018 (MISEV2018): a position statement of the International Society for Extracellular Vesicles and update of the MISEV2014 guidelines.

From the ISEV guidelines: “ISEV endorses “extracellular vesicle” (EV) as the generic term for particles naturally released from the cell that are delimited by a lipid bilayer and cannot replicate, i.e. do not contain a functional nucleus. Since consensus has not yet emerged on specific markers of EV subtypes, such as endosome-origin “exosomes” and plasma membrane-derived “ectosomes” (microparticles/microvesicles) [3,4] assigning an EV to a particular biogenesis pathway remains extraordinarily difficult unless, e.g. the EV is caught in the act of release by live imaging techniques. Therefore, unless authors can establish specific markers of subcellular origin that are reliable within their experimental system(s), authors are urged to consider use of operational terms for EV subtypes that refer to a) physical characteristics of EVs, such as size (“small EVs” (sEVs) and “medium/large EVs” (m/IEVs), with ranges defined, for instance, respectively, < 100nm or <200nm [small], or > 200nm [large and/or medium]) or density (low, middle, high, with each range defined); b) biochemical composition (CD63+/CD81+ EVs, Annexin A5-stained EVs, etc.); or c) descriptions of conditions or cell of origin (podocyte EVs, hypoxic EVs, large oncosomes, apoptotic bodies) in the place of terms such as exosome and microvesicle that are historically burdened by both manifold, contradictory definitions and inaccurate expectations of unique biogenesis.”

Thank you for the careful reading and constructive suggestions. We have replaced all “exosome (exo)” with “small extracellular vesicle (sEV)” in the revised manuscript and Supplementary Material.

#2- Line #241: Electroporation settings and procedure are missing.

Thank you for the comment. Electroporation settings and procedures have been supplemented in the revised manuscript. It now reads, “To load the sEVs with miR-328a-3p/negative control mimics, 100 µg of purified sEVs and nanoparticle-miRNA complexes (final concentration of 100nM for miR-328a-3p/negative control mimics) were gently mixed in 100 µL of PBS at 4 °C. After electroporation at 200 V for 5ms in electroporation cuvettes, the mixture was incubated at 37 °C for 30 min.”

#3- Line #342: “With” not “were” should be read.

Thank you for the comment. We apologize for the confused expression. We have revised the sentences in the methods section. They now read, “From January 2016 to January 2019, 24 patients were admitted to the Third Affiliated Hospital of Southern Medical University. All patients were fully informed of the study, and they provided written consent prior to participation.

Among the 24 patients, there were 12 patients with simple shaft fractures and 12 patients with shaft fractures and concomitant TBIs who were assigned to three groups (week 2, week 4, and week 6, four patients per group).”

#4- Line #349: How many Healthy matched individuals were included?

Thank you for the comment. Healthy matched individuals have been described in the Methods section. Plasma exosomes were collected from TBI patients (10 males) and healthy control (10 males) between the ages of 17 and 72 years, admitted to the Third Affiliated Hospital of Southern Medical University.

#5- Line #410: What is the concentration of calcein being used?

Thank you for the comment. We have added the concentration of calcein being used. It now reads, “To investigate *in vivo* bone regeneration, rats were intraperitoneally injected with 15 mg/kg of calcein to mark the new bone 1 and 3 weeks after the surgery.”

#6- Line #196: What does the 800 ug relate to (EVs)?

Thank you for the comment. The protein concentration in exosomes was determined using the BCA protein assay kit. The amount of sEVs was represented by the amount of protein, which has been used in previous studies [15].

#7- Line #209: There is a typographical error: osteoprogenitors and not psteoprogenitors

Thank you for the comment. We have revised the word in the manuscript.

#8- Line #141: There is a typographical error: healthy and not health controls

Thank you for the careful reading. We have revised the sentence in the manuscript. It now reads, “Interestingly, miR-328a-3p and miR-150-5p were also markedly increased in plasma sEVs from patients with TBI, as compared to healthy controls.”

1. Song DD, Zhang TT, Chen JL, Xia YF, Qin ZH, Waeber C, Sheng R: **Sphingosine kinase 2 activates autophagy and protects neurons against ischemic injury through interaction with Bcl-2 via its putative BH3 domain.** *Cell Death & Disease* 2017, **8**.
2. Zhi J, Duan B, Pei JW, Wu SD, Wei JL: **Daphnetin protects hippocampal neurons from oxygen-glucose deprivation-induced injury.** *Journal of Cellular Biochemistry* 2019, **120**:4132-4139.
3. Catalano M, O'Driscoll L: **Inhibiting extracellular vesicles formation and release: a review of EV inhibitors.** *Journal of Extracellular Vesicles* 2020, **9**.
4. Iguchi Y, Eid L, Parent M, Soucy G, Bareil C, Riku Y, Kawai K, Takagi S, Yoshida M, Katsuno M, et al: **Exosome secretion is a key pathway for clearance of pathological TDP-43.** *Brain* 2016, **139**:3187-3201.
5. Richards KE, Zeleniak AE, Fishel ML, Wu J, Littlepage LE, Hill R: **Cancer-associated fibroblast exosomes regulate survival and proliferation of pancreatic cancer cells.** *Oncogene* 2017, **36**:1770-1778.
6. Xiao CC, Wang K, Xu YC, Hu HX, Zhang N, Wang YC, Zhong ZW, Zhao J, Li QJ, Zhu D, et al: **Transplanted Mesenchymal Stem Cells Reduce Autophagic Flux in Infarcted Hearts via the**

- Exosomal Transfer of miR-125b.** *Circulation Research* 2018, **123**:564-578.
7. Kosaka N, Iguchi H, Yoshioka Y, Takeshita F, Matsuki Y, Ochiya T: **Secretory Mechanisms and Intercellular Transfer of MicroRNAs in Living Cells.** *Journal of Biological Chemistry* 2010, **285**:17442-17452.
 8. Vora A, Zhou WS, Londono-Renteria B, Woodson M, Sherman MB, Colpitts TM, Neelakanta G, Sultana H: **Arthropod EVs mediate dengue virus transmission through interaction with a tetraspanin domain containing glycoprotein Tsp29Fb.** *Proceedings of the National Academy of Sciences of the United States of America* 2018, **115**:E6604-E6613.
 9. Guo BB, Bellingham SA, Hill AF: **Stimulating the Release of Exosomes Increases the Intercellular Transfer of Prions.** *Journal of Biological Chemistry* 2016, **291**:5128-5137.
 10. Kamekar S, LeBleu VS, Sugimoto H, Yang SJ, Ruivo CF, Melo SA, Lee JJ, Kalluri R: **Exosomes facilitate therapeutic targeting of oncogenic KRAS in pancreatic cancer.** *Nature* 2017, **546**:498-+.
 11. Boes M, Kain M, Kakar S, Nicholls F, Cullinane D, Gerstenfeld L, Einhorn TA, Torneta P: **Osteogenic effects of traumatic brain injury on experimental fracture-healing.** *Journal of Bone and Joint Surgery-American Volume* 2006, **88A**:738-743.
 12. Hofman M, Koopmans G, Kobbe P, Poeze M, Andruszkow H, Brink PRG, Pape HC: **Improved Fracture Healing in Patients with Concomitant Traumatic Brain Injury: Proven or Not?** *Mediators of Inflammation* 2015, **2015**.
 13. Perkins R, Skirving AP: **Callus formation and the rate of healing of femoral fractures in patients with head injuries.** *The Journal of bone and joint surgery British volume* 1987, **69**:521-524.
 14. Cadosch D, Gautschi OP, Thyer M, Song S, Skirving AP, Filgueira L, Zellweger R: **Humoral Factors Enhance Fracture-Healing and Callus Formation in Patients with Traumatic Brain Injury.** *Journal of Bone and Joint Surgery-American Volume* 2009, **91A**:282-288.
 15. Li DF, Liu J, Guo BS, Liang C, Dang L, Lu C, He XJ, Cheung HYS, Xu L, Lu CW, et al: **Osteoclast-derived exosomal miR-214-3p inhibits osteoblastic bone formation.** *Nature Communications* 2016, **7**.

Reviewers' Comments:

Reviewer #3:

Remarks to the Author:

The Authors have revised adequately the manuscript which now addressed all the Reviewers comments.